# Characterizing the tropospheric water vapor spatial variation and trend using 2007-2018 COSMIC radio occultation and ECMWF reanalysis data

Xi Shao[1], Shu-Peng Ho[2], Xin Jing[1], Xinjia Zhou[3], Yong Chen[2], Tung-Chang Liu[1], Bin Zhang[1,3], Jun Dong[1]

[1]Cooperative Institute for Satellite Earth System Studies (CISESS), Earth System Science Interdisciplinary Center, University of Maryland, College Park, MD, 20740, USA
[2]NOAA National Environmental Satellite, Data, and Information Service, Center for Satellite Applications and Research, College Park, MD, 20740, USA
[3]Global Science & Technology, Inc., 7855 Walker Drive, Suite 200, Greenbelt, MD 20770, USA

Correspondence to: Xi Shao (xshao@umd.edu)

**Abstract.** Atmospheric water vapor plays a crucial role in the global energy balance, hydrological cycle, and climate system. High-quality and consistent water vapor data from different sources are vital for weather prediction and climate research. This study assesses the consistency between Formosa Satellite Mission 3–Constellation Observing System for Meteorology, Ionosphere, and Climate (FORMOSAT-3/COSMIC) radio occultation (RO) and European Centre for Medium-Range Weather Forecasts (ECMWF) ReAnalysis Model 5 (ERA5) water vapor datasets. Comparisons are made across different atmospheric pressure levels (300 hPa, 500 hPa, and 850 hPa) from 2007 to 2018. Generally, the two datasets show good spatial and temporal agreements. COSMIC's global water vapor retrieval is slightly lower than ERA5's at 500 and 850 hPa, with distinct latitudinal differences between hemispheres. COSMIC exhibits global water vapor increasing trends of 3.47±1.77, 3.25±1.25, and 2.03±0.65%/Decade at 300 hPa, 500 hPa, and 850 hPa, respectively. Significant regional variability in water vapor trends, encompassing notable increasing and decreasing patterns, is observable in tropical and subtropical regions. At 500 and 850 hPa, strong water vapor increasing trends are noted in the equatorial Pacific Ocean and the Laccadive Sea, while decreasing trends are evident in the Indo-Pacific Ocean region and the Arabian Sea. Over land, substantial increasing trends at 850 hPa are observed in the southern United States, contrasting with decreasing trends in South Africa and Australia. The differences between the water vapor trends of COSMIC and ERA5 are primarily negative in the tropical regions at 850 hPa. However, the water vapor increasing trends at 850 hPa estimated from COSMIC are significantly higher than the ones derived from ERA5 data for two low-height stratocumulus cloud-rich ocean regions west of Africa and South America. These regions with notable water vapor trend differences are located in the Intertropical Convergence Zone (ITCZ) area with frequent occurrences of convection, such as deep clouds. The difference in characterizing water vapor distribution between RO and ERA5 in deep cloud regions may cause such trend differences. The assessment of spatiotemporal variability in RO-derived and reanalysis of atmospheric water vapor data helps assure the quality of these datasets for climate studies.

## 1. Introduction

Water vapor is one of the most important greenhouse gases in the atmosphere, which accounts for about 60% of the natural greenhouse effect (Kiehl and Trenberth, 1997; Wagner et al., 2006; Foster et al., 2007; Ahrens and Samson, 2011). Water vapor cycles with latent heat release through condensation and evaporation are closely linked to cloud formation, which alters atmospheric energy budgets. In addition, studies showed that water vapor amplifies global warming (Smith and Reynolds, 2005; Parker et al., 2007; Dai, 2006; Allan and Soden, 2008; Mieruch et al., 2008; Zhang et al., 2013). As the earth warms, the water vapor concentration in the lower troposphere increases with increasing temperature, increasing the evaporation rate and adding more atmospheric water vapor, which usually warms the atmosphere further. The water vapor's heat-trapping effect is crucial in climate change (Forster et al., 2007). Studies (i.e., Foster et al., 2007; Allan et al., 2010; Trenberth, 2011; Hegerl et al., 2015) show that water vapor has profound impacts on atmospheric temperature structure and hydrological cycle, which, in turn, increases the likelihood of extreme regional precipitation events, extreme weather conditions, and droughts.

Accurate atmospheric water vapor climate data records (CDRs) are critical for detecting climate change. Various studies have quantified the spatial and temporal variation and trend in atmospheric water vapor using two types of water vapor data: i) measurements or retrievals from sensor observations and ii) reanalysis data produced by assimilating various observations. The first data type includes both ground-based in situ and space-borne observations: long-term radiosonde measurements (Zhai and Eskridge, 1997; Ross and Elliott, 2001; Ho et al., 2010; Zhao et al., 2012; Zhang et al., 2018), weather station data (Dai, 2006), water vapor retrieved from ground-based Global Positioning System (GPS) station data (Kursinski et al., 1997; Bock et al., 2007; Nilsson and Elgered, 2008; Vey et al., 2010; Huang et al., 2013; Chen and Liu, 2016; Yuan et al., 2021), water vapor retrievals from space-borne radio occultation observations (Ho et al., 2009; Huang et al., 2013; Ho et al., 2018; Zhang et al. 2018; Andrisaniand and Vespe, 2020; Gleisner et al., 2022), visible spectral-range sensor observations (Mieruch et al. 2008; Grossi et al., 2015; Borger et al., 2021), microwave (Rosenkranz, 2001; Chen and Liu, 2016; Ho et al., 2018; Yadav et al., 2021), and infrared sounder observations (Susskind et al., 2003).

The second type of water vapor data is from the global atmospheric reanalysis products generated by the European Centre for Medium-Range Weather Forecasts (ECMWF) (Hersbach et al., 2020) and the National Centers for Environmental Prediction (NCEP) (Whitaker et al., 2008). These reanalysis data are constructed from assimilating in situ, and satellite observations through data assimilation (DA) systems blended with model outputs. These atmospheric reanalysis data have been used for investigating long-term atmospheric water vapor variability and trends (Bengtsson, 2004; Wagner et al., 2006; Adler et al., 2008; Ho et al., 2009; Dessler and Davis, 2010; Huang et al., 2013; Zhang et al., 2013; Chen and Liu, 2016; Xie et al., 2020; He et al., 2022) and climate change studies (Allan, 2002; Allan et al., 2014; Lu et al., 2015). However, the quality of the reanalysis data may be affected by i) discontinuity or changes of in situ data and satellite data, ii) the inadequate spatial and temporal coverage of the observations, iii) inadequate measurement bias corrections, iv) preliminary observation error

estimates, v) contaminations of ground-based and space-borne satellite observations due to clouds, and vi) potential/unknown model errors (Sherwood et al., 2010; Chen and Liu, 2016). The uncertainty of forecast and reanalysis data under cloudy conditions, especially over oceans, is still very large (Lonitz and Geer 2017).

Past climate modeling studies suggest that increasing surface temperature can result in an increasing trend in global water vapor (Held and Soden, 2000, 2006; Santer et al., 2006). Studies based on various types of observations and reanalysis data have shown an increasing water vapor trend over different periods ranging from several decades to the recent decade (Bengtsson, 2004; Wagner et al., 2006; Ho et al., 2009; Chen and Liu, 2016; Wang et al., 2017; Ho et al., 2018). However, these studies also showed substantial variation (with both increasing and decreasing) in regional water vapor trends (Ross and
Elliott, 2001; Dai, 2006; Mieruch et al., 2008, 2014; Zhang et al., 2018). This is mainly because regional water vapor concentration may change dramatically depending on multiple non-thermodynamic factors such as i) surface type, ii) long-range transport of air masses, and iii) water availability. As a result, the global surface temperature increase does not increase water vapor everywhere (Chou and Neelin, 2004; Wagner et al., 2006; Lu et al., 2015; Chen and Liu, 2016).

Many studies (i.e., Ho et al., 2009; Chen and Liu, 2016; Ho et al., 2018) have compared global reanalysis of water vapors with those derived from in situ and satellite sensors. For example, Chen and Liu (2016) evaluated the global precipitable water vapor (PWV) variability and trend from ECMWF and NCEP reanalysis results. They compared the water vapor reanalysis with 36-year (1979 to 2014) water vapor datasets collected from radiosonde, ground-based Global Navigation Satellite System (GNSS), and microwave satellite observations. All these datasets showed increasing PWV trends. The ERA-interim reanalysis
agrees with microwave satellite observations better than those from the NCEP reanalysis. The ERA-interim overestimates the PWV over the ocean for the period before 1992 compared to microwave satellite data. It is essential to continue comparing the differences and consistencies of atmospheric water vapor data's temporal and spatial variabilities from different sources and provide the climate community with high-quality water vapor data.

There is growing interest in comparing reanalysis data and all-weather water vapor profiles retrieved from GNSS radio occultation (RO) (Anthes et al., 2000; Kursinski et al., 2001; Ho et al., 2009, 2010). Complementing the measurements from microwave and infrared sounders, RO data can provide information on the temperature, water vapor, and pressure with high accuracy, precision, and vertical resolution. Because the quality of RO data does not change during the day or night and is not affected by clouds (Anthes et al. 2008; Ho et al. 2020a), the RO temperature and water vapor profiles co-located with reanalysis
data would help identify the variation of temperature and humidity under all-weather conditions over time. RO data has been used to evaluate biases and monitor calibration changes for microwave measurements (Iacovazzi et al., 2020; Shao et al., 2021a) and infrared sounders (Chen et al., 2022). Further, RO-derived water vapor profiles have been used to distinguish systematic water vapor biases in radiosondes (Ho et al., 2010; Sun et al., 2019; Ho et al., 2020a; Shao et al., 2021b).

In this paper, we characterize the water vapor data derived from Formosa Satellite Mission 3–Constellation Observing System for Meteorology, Ionosphere, and Climate (FORMOSAT-3/COSMIC) (hereafter COSMIC) and those from ECMWF Re-Analysis model 5 (ERA5). Launched in 2006, COSMIC was the first constellation of microsatellites carrying GPS RO receivers. COSMIC has demonstrated the value of RO data in the ionosphere for climate and meteorological research and operational weather forecasting (Ho et al., 2020a). This paper aims to characterize and compare the global, latitudinal, and regional variabilities of COSMIC and ERA5 water vapor distributions, seasonality, and long-term trends at selected pressure levels from 2007 to 2018. In addition, this paper identifies regions with notable increasing and decreasing water vapor trends, i.e., regions becoming moister or drier, and regions with significant water vapor trend differences between COSMIC and ERA5. Particular interest is also placed on comparing the COSMIC and ERA5 water vapor trends over the stratocumulus cloud-rich regions to investigate the impacts of stratocumulus clouds on near-surface water vapor data quality in ERA5.

This paper is organized as follows: Section 2 introduces the water vapor data from COSMIC RO retrieval and ERA5 reanalysis. Section 3 analyzes global and latitudinal variabilities of long-term (2007-2018) COSMIC and ERA5 water vapor data at three pressure levels, and their differences are quantified. In Section 4, the global and latitudinal water vapor trends derived from COSMIC and ERA5 are quantified and compared at different pressure levels. Section 5 examines the overall distribution of regional water vapor trends derived from the COSMIC and ERA5 time series and their differences. Furthermore, a few specific sites with frequent stratocumulus cloud coverage and large differences between COSMIC and ERA5 are selected to quantify the water vapor trend differences. Additionally, the seasonal variability of latitudinal water vapor distribution is summarized in Appendix A.1. Appendix A.2 and A.3 describe the estimation of the water vapor trend with sampling error removal and its associated uncertainties for a given region of interest (RoI). Appendix A.4 provides supplemental information on a few sites with notable increasing and decreasing water vapor trends. We present the conclusions and discussions in Section 6.

## 2. Datasets used for Spatial and Temporal Water Vapor Variability Analysis

### 2.1 ECMWF Reanalysis Data

This study used the ERA5 global atmospheric and climate reanalysis dataset (https://www.ecmwf.int/en/forecasts/dataset/ecmwf-reanalysis-v5). ERA5 is the fifth-generation ECMWF reanalysis dataset covering the past 4 to 7 decades. The ERA5 dataset is generated from the Four-Dimensional Variational (4DVAR) data assimilation system, which uses a fixed version of the ECMWF NWP system, i.e., Integrated Forecasting System (IFS) Cy41r2. The IFS-Cy41r2 system became operational in 2016 (Hersbach et al., 2020) and blends or assimilates meteorological observations (e.g., surface weather stations, ocean buoys, radiosonde stations, aircraft, and remote sensing satellites) with a previous forecast to obtain the best for both. These blended results serve as the initial conditions for the next forecast period. The ERA5 water vapor data are from the ground to ~0.1 hPa at 37 mandatory pressure levels. Our study used ERA5 global water vapor profiles from 2007 to 2018 in 6-hour increments. The ERA5 data were collected with a 0.25° spatially gridded

resolution, equivalent to a spatial resolution of ~25 km at the equator. Many studies have been conducted to validate the ERA5 atmospheric products using satellite measurements (Chen and Liu, 2016; Lei et al., 2020; Tang et al., 2021; Campos et al., 2022). Overall, the results of these studies show that ERA5 is in good agreement with satellite measurements (or retrieved

products). For example, Tang et al. (2021) compared the Atmospheric downward longwave radiation (DLR) from Clouds and Earth's Radiant Energy System (CERES) satellite retrievals and ERA5 data with observations at Baseline Surface Radiation Network (BSRN) stations over land surfaces. The ERA5 atmospheric reanalysis performed better than satellite retrievals in estimating DLR over the land surface. According to Chen and Liu (2016), the global water vapor trend over 1992–2014 from the data of the ECMWF reanalysis model agrees well with the microwave satellite data. These studies provide confidence in

the accuracy of the ERA5 products for comparison with COSMIC retrievals.

## 2.2 COSMIC WETPrf water vapor retrieval

The COSMIC RO receivers on the Low Earth Orbit (LEO) satellites measure the phase delay of radio waves, which are emitted from the GPS satellites and bent by atmospheric refraction. Profiles of atmospheric refractivity can be derived from the bending angles of radio wave trajectories when propagating through the ionosphere, stratosphere, and troposphere. From the retrievals

of RO limb-sounding observations, the bending angle and refractivity profiles from the excess phase data processed from the Doppler-shifted raw radio signals transmitted by GPS satellites are derived. Then, the One-Dimensional Variational (1DVAR) retrieval algorithm is applied to solve an under-determined problem: determine the atmospheric temperature and water vapor profiles from bending angle or refractivity data. The 1DVAR retrieval generally uses *a priori* state of the atmosphere, i.e., vertical background temperature and humidity profiles, and associated background and observation uncertainties/error

covariance matrices (ECM) to minimize a quadratic cost function.

In this paper, we analyze the 2007 to 2018 COSMIC wet profile data produced by the University Corporation for Atmospheric Research (UCAR) from COSMIC RO data, namely WETPrf (https://cdaac-www.cosmic.ucar.edu/cdaac/products.html). The WETPrf data from the COSMIC Data Analysis and Archive Center (CDAAC) consist of temperature, water vapor, and

pressure profiles with a high vertical resolution (100 m). UCAR WETPrf profile data contain the latitude and longitude of the RO perigee point, temperature, pressure, water vapor profile, and mean sea level height. COSMIC has provided more than seven million RO-sounding profiles over its lifetime. Many of the six COSMIC GPS receivers continued beyond their 2-year designed life and provided more than 1,000 occultation profiles per day through 2016. The COSMIC data decreased significantly in late 2019 and was decommissioned in May 2020.

The UCAR COSMIC WETPrf data was generated with the heritage 1DVAR algorithm at CDAAC to produce wet temperature and humidity profile data. In the 1DVAR algorithm for WETPrf, background profiles are taken from ERA-Interim gridded low-resolution data and interpolated to the time and location of RO measurements to separate the pressure, temperature, and moisture contributions to the refractivity. The constraint applied to WETPrf in the 1DVAR retrieval is very tight, such that

temperature and moisture profiles are reported only when the residual refractivity (i.e., the difference between the observed refractivity and simulated refractivity computed from the retrieved temperature and moisture profiles) are within the uncertainty of refractivity. This ensures that the information on refractivity measurements from RO is completely used in the 1DVAR (Ho et al., 2020a).

**2.3 Method of comparing COSMIC and ERA5 water vapor data**

In our analysis, COSMIC RO profiles with the 'Bad' flag have been filtered out. COSMIC RO and ERA5 water vapor profiles were paired through collocation before the analysis was performed. The ERA5 data have a global distribution over 0.25-degree latitude/longitude grids, vertically over 37 pressure layers, and at 6-hour intervals. Therefore, the ERA5 water vapor data at a given pressure level are interpolated at the latitude/longitude of the perigee point of the RO profile and at RO time to match the COSMIC RO measurement. For the RO data, the fine vertical resolution COSMIC RO water vapor profiles are interpolated

onto three pressure levels, e.g., 300, 500, and 850 hPa, selected to characterize water vapor variations at representative altitudes around 9 km, 5.5 km, and 1.5 km, respectively.

The pressure level at 850 hPa studied in this paper is close to the surface and within the boundary layer. Its water vapor can vary based on factors such as humidity levels near the surface, regional water vapor sources, and weather patterns. From

previous studies (Ho et al., 2009, 2020a; Shao et al., 2021a; Johnston et al., 2021) of comparing RO water vapor data with collocated reanalysis model data or radiosonde measurements, it was found that RO water vapor retrievals have a negative bias in the lower troposphere. The COSMIC water vapor retrieval is strongly affected by super-refraction at this pressure level in the moisture-rich regions (Ho et al., 2010). It is worth evaluating the relative biases and consistency in the trends on various spatial scales between COSMIC and ERA5 water vapor datasets at this 850 hPa pressure level.

The water vapor at 500 hPa can vary widely depending on local weather conditions and atmospheric patterns. Water vapor at 500 hPa is crucial for understanding the development of weather patterns, including mid-latitude cyclones, ridges, and troughs. This pressure level also contributes to the upper-level atmospheric circulation patterns through convection, which carries moist air upward from the lower troposphere and plays a role in redistributing heat and moisture. It was learned from the earlier

comparison of RO data with radiosonde measurements that starting from the pressure level at 500 hPa, the RO-water vapor retrieval uncertainty increases as altitude decreases. Therefore, we chose 500 hPa as the representative middle troposphere of interest to study in this paper.

The 300 hPa pressure level represents the water vapor layer with fewer horizontal variations at higher altitudes. Water vapor

in the upper troposphere plays a critical role in the Earth's radiative balance and climate system. It affects the absorption and emission of radiation, contributing to warming (absorbing and trapping infrared radiation, i.e., greenhouse effect) and cooling (emitting heat energy) effects. Johnston et al. (2021) showed large discrepancies in the ERA5 and MERRA2 reanalysis model

water vapor profiles compared to COSMIC-2 in the upper troposphere. There are large uncertainties for the reanalysis model to estimate the upper troposphere water vapor due to the combined effects of complex atmospheric dynamics (jet streams, convection, and mixing) at high altitudes, sparse observations and difficulties in validation, errors in extrapolating from lower altitude measurements, and accurate accounting of radiative effects at high altitudes. Therefore, we chose 300 hPa as the representative upper troposphere level to compare spatial and temporal variabilities of water vapor between COSMIC and ERA5.

## 2.4 Impact of ERA-Interim as *a priori* on COSMIC water vapor retrieval

The UCAR's 1DVAR retrieval algorithm for COSMIC WETPrf (water vapor and humidity) uses ERA-Interim profiles as the *a priori* input (Wee et al., 2022). In addition, the UCAR WETPrf water vapor/temperature retrieval also enforces a retrieval constraint to the residual refractivity. Such a constraint can determine the influence of ERA-Interim on the final water vapor retrieval at different pressure levels. On the other hand, the ERA5 provides a more comprehensive and reliable reanalysis by using improved weather forecast and data assimilation models with various ground, in-situ, and satellite measurements compared to ERA-Interim (Fujiwara et al., 2017; Hersbach et al., 2020). Figure 1 depicts the monthly (using January and July of 2007 as representative winter and summer months of the northern hemisphere) scatter plots of the collocated COSMIC global water vapor versus ERA5 and ERA-Interim water vapor data at three pressure levels. The linear regression statistics for COSMIC versus ERA5 and COSMIC versus ERA-Interim comparisons are also shown on the plots. All plots show that COSMIC versus ERA-Interim comparisons are more scattered than the COSMIC versus ERA5 comparison. Quantitatively, the correlation coefficients between COSMIC and ERA5 are around 0.96, while the correlation coefficient between COSMIC and ERA-Interim varies from 0.88 to 0.93. The linear fitting coefficients, i.e., slopes, of COSMIC versus ERA5 fittings are closer to 1 than COSMIC versus ERA-Interim fitting in all panels of Figure 1. In terms of the linear fitting root-mean-square-error (RMSE) residuals, the RMSEs of COSMIC versus ERA5 fitting are lower than the COSMIC versus ERA-Interim fitting by 24% to 47% over the two selected months (January and July of 2007) and three pressure levels. These analysis results indicate that the COSMIC water vapor retrievals are more consistent with ERA5 than ERA-Interim. It suggests that the information on COSMIC 1DVAR retrievals is mainly from the COSMIC refractivity instead of the ERA-Interim. We also inspected the comparison of COSMIC versus ERA5 or ERA-Interim for other months (not shown here), and the conclusion that COSMIC water vapor data is more consistent with ERA-5 than ERA-Interim holds for these months as well.

The comparisons between COSMIC and ERA5 water vapor (Fig. 1) suggest overall consistencies over the two selected months and at three pressure levels, which requires further quantitative analysis of the variabilities. In the following sections, we analyze the collocated COSMIC and ERA5 water vapor at three pressure levels to study their spatial (Section 3) and trend (Sections 4 and 5) variabilities (the seasonal trend is provided in Appendix A.1).

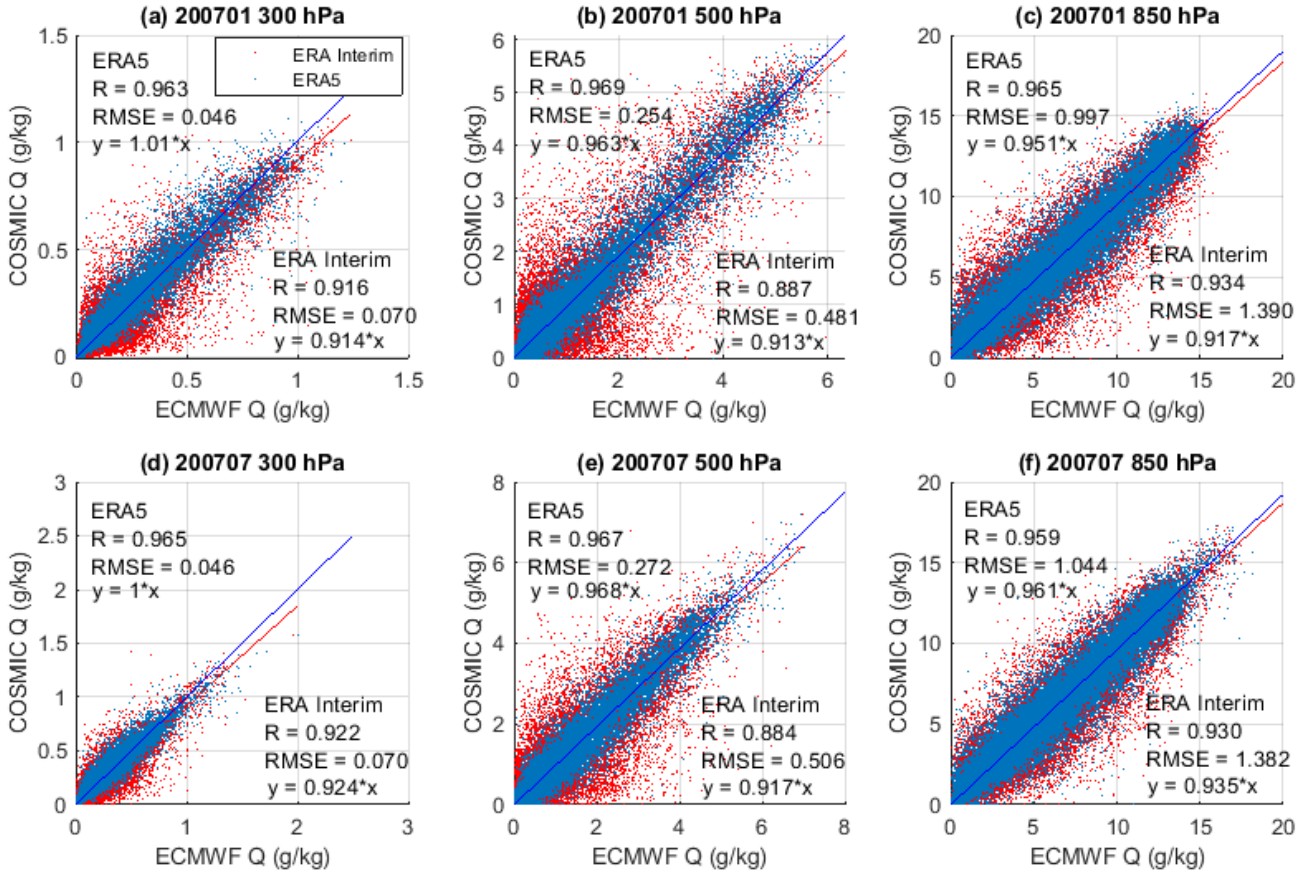

**Figure 1: Scatter plots of collocated COSMIC water vapor retrieval versus ERA5 and ERA-Interim water vapor data for two months (a, b, c: 2007/01; d, e, f: 2007/07) at three pressure levels: (a, d) 350 hPa, (b, e) 500 hPa, and (c, f) 850 hPa. The correlation coefficient (*R*), linear fitting coefficient, and RMSE of the fitting residual for COSMIC water vapor retrieval versus ERA5 and ERA-Interim comparisons are listed in each panel.**

**3.  Comparison of Spatial Variability of Water Vapor between COSMIC and ERA5**

**3.1 Global distribution of COSMIC and ERA5 water vapor**

To inter-compare the spatial variability of the water vapor data between COSMIC and ERA5 (interpolated onto COSMIC locations and times), the collocated global humidity data over 12 years (2007-2018) are grouped into 10°x10° latitude/longitude grids and spatial/time-averaged at three selected pressure levels, e.g., 300, 500, and 850 hPa.

Figure 2 compares time-averaged global water vapor distribution maps over three pressure levels between COSMIC (left column) and ERA5 (right column). The overall global distribution of water vapor of COSMIC and ERA5 at three pressure

levels is generally consistent. At all three pressure levels, the global water vapor distribution exhibits high concentration in the low latitude tropical regions, decreases rapidly toward the polar region, and is low in some high terrain regions such as the

245 Tibetan Plateau. In the low latitude tropical region, i.e., latitudes between -20 and 20 degrees, increased water vapor concentrations occur in the East Indian and West Pacific Ocean regions and over the Amazon rainforest regions in South America at these three pressure levels. It is noted that COSMIC bending angles are assimilated into ERA5, which significantly improves the upper-troposphere and lower-stratosphere temperatures (Hersbach et al. 2020). However, the COSMIC 1DVAR retrieval has more independence from its *a priori* (ERA-Interim) for water vapor within the lower/middle troposphere. Primary

water vapor information is retrieved from the RO observations at these altitudes, which our study is focused on. The evaluations of global and latitude-dependent water vapor differences between COSMIC and ERA5 in the following sections would help understand the extent and regional dependence of the assimilation of COSMIC RO water vapor data in ERA5.

To quantitatively evaluate the consistency between COSMIC and ERA5 water vapor ($Q$) data, the relative

biases ($(Q_{COSMIC} - Q_{ERA5})/Q_{ERA5}$ (%)) between COSMIC and ERA5 are calculated with the 12-year collocated COSMIC and ERA5 global water vapor data. The mean differences between COSMIC and ERA5 global water vapor are 5.67±34.30%, -1.86±30.09%, and -2.30±21.21% for pressure levels at 300, 500, and 850 hPa, respectively. This suggests that at 500 and 850 hPa, COSMIC water vapor retrieval is lower than ERA5 water vapor data. This is consistent with the negative moisture biases below 5 km for the RO retrievals compared to the collocated radiosonde data (Ho et al., 2009, 2020a; Shao et al., 2021b). Such

near-surface moisture biases may come from the 1DVAR RO retrieval when the super-refraction with a sharp refractivity gradient occurs in the moisture-rich low-tropospheric RO profiles (Ho et al., 2020b; Shao et al., 2021a,b). At 300 hPa, the COSMIC water vapor concentration is about 5.67% higher than ERA5. Since the water vapor concentration at 300 hPa is very low, its contribution to the total precipitable water would be minimal. The main cause that at 300 hPa, water vapor from COSMIC is higher than from ERA5 stems from the distinctive cloud-penetration capability of the RO signal. In contrast, there

are uncertainties in the water vapor from the reanalysis data over the cloud-free scenes since these scenes can be over thin or cirrus clouds due to the difficulty in the data assimilation system over these types of clouds. The water vapor concentration derived from COSMIC is expected to be higher than ERA5 at 300 hPa when the thin or cirrus clouds are present. Our evaluation of water vapor at 300 hPa indicates that the difference between RO and ERA5 about 5.7% is likely due to the uncertainty in classifying cloud-free scenes in the data assimilation and in the RO retrieval system. Such assessment is consistent with the

water vapor biases between COSMIC-2 and ERA5 presented in Johnston et al., 2021.

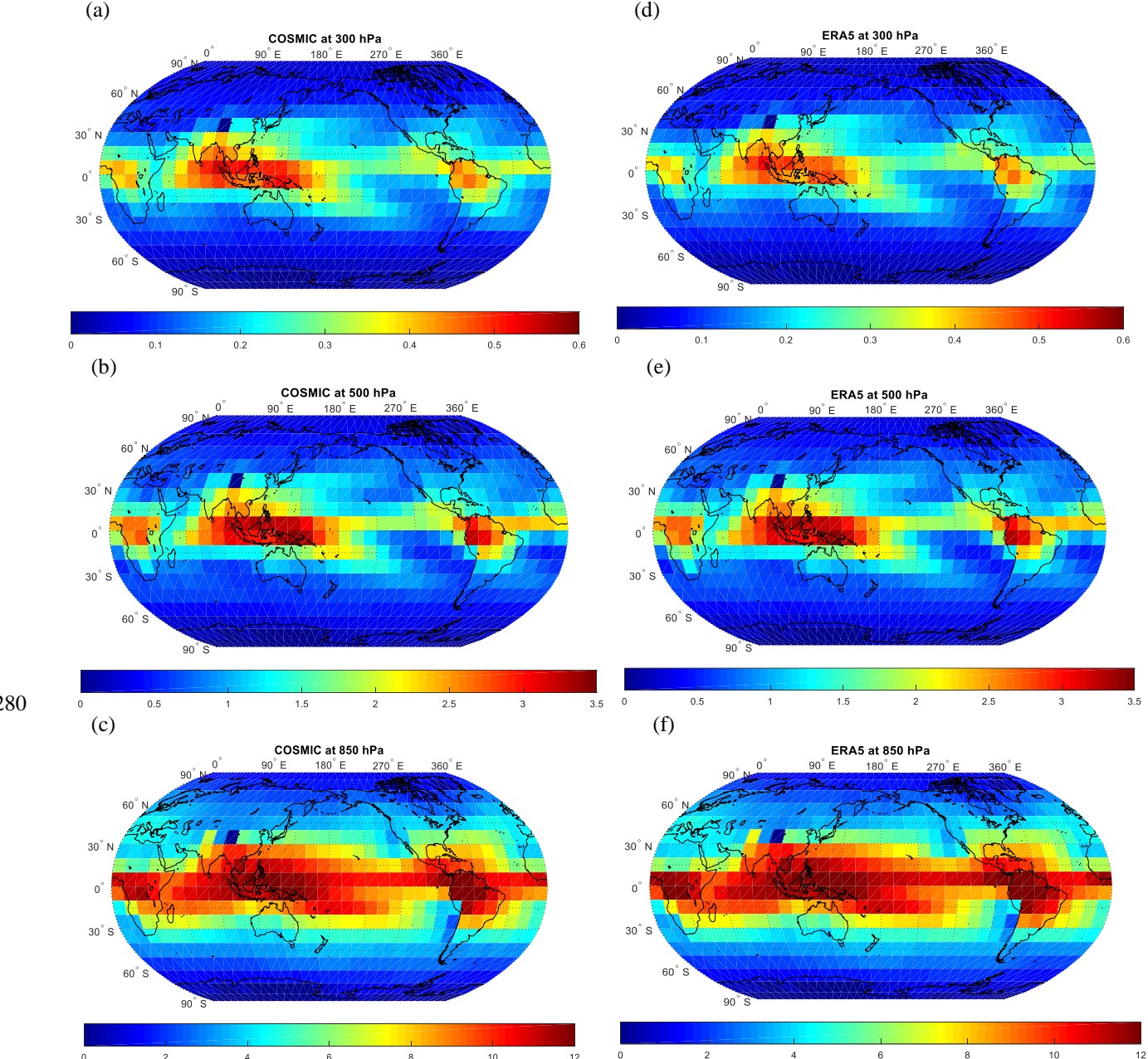

**Figure 2: Comparison of the global distribution of 10×10-degree grid-averaged water vapor (g/kg) data between COSMIC retrievals (a, b, c) at 300, 500, 850 hPa, and ERA5 data (d, e, f) at 300, 500, 850 hPa, respectively.**

We also notice the significant uncertainties in estimating upper troposphere water vapor in the reanalysis model. Johnston et al. (2021) analyzed COSMIC-2 and reanalysis (ERA5 and MERRA2) water vapor differences in different latitude zones. It was shown that the UCAR COSMIC-2 water vapor retrieval is consistently lower than both ERA5 and MERRA2 water vapor

data in the lower troposphere (below 2 km). However, COSMIC-2 water vapor retrieval data is higher than ERA5 data and lower than MERRA2 data at altitudes above 5 km. The magnitude of the COSMIC-2 vs. ERA5 water vapor difference is smaller than that of COSMIC-2 vs. MERRA2 above 5 km. The opposite sign and large magnitude of the ERA5 and MERRA2 model water vapor differences relative to COSMIC-2 in the upper troposphere suggest the large uncertainties in calculating water vapor in the reanalysis model over this altitude region. There are ongoing efforts to quantify the ERA5 biases in the upper troposphere through comparison with other measurements, such as using multi-campaign datasets on research aircraft (Krüger et al., 2022). However, the results are inconclusive due to the comparison's limited regional, height, and temporal coverage. In this regard, the comparisons presented in this paper help assess the biases in the reanalysis model. Further comparisons with collocated radiosonde measurements can also help assess the biases in ERA5 in the upper troposphere.

### 3.2 Latitude-dependence of COSMIC and ERA5 water vapor distribution

The comparisons of the latitudinal dependence of water vapor distribution between COSMIC and ERA5 at three pressure levels are shown in Fig. 3. Eight latitudinal bins from -80 to 80 degrees with 20-degree bin width are used to group COSMIC and ERA5 water vapor data. The 20-degree wide latitude bins over northern and southern hemispheres are selected to characterize water vapor latitude-dependence in different reprehensive latitudinal zones such as 0°-20° for tropical, 20°-40° for sub-tropical, 40°-60° for mid-latitude, and 60°-80° for high-latitude regions. The regions with latitudes above 80 degrees were not selected due to much less data coverage from COSMIC. The collocated COSMIC and ERA5 water vapor data over all months in 12 years (2007-2018) have been used to calculate the mean water vapor over these latitude bins, as shown in Fig. 3. Figures 3a, 3d, and 3g show the side-by-side comparison of COSMIC and ERA5 water vapor data averaged over 20-degree latitude bins at the three selected pressure levels (300, 500 and 850 hPa), respectively. The panels in the middle and right columns of Fig. 3 show the latitude-dependence of the COSMIC minus ERA5 water vapor mean difference ($\Delta Q_{COSMIC-ERA5} = Q_{COSMIC} - Q_{ERA5}$) and relative difference ($\Delta Q_{COSMIC-ERA5}(\%) = (Q_{COSMIC} - Q_{ERA5})/ Q_{ERA5} \times 100$).

In general, COSMIC and ERA5 water vapor data (Fig. 3) show that latitudinal water vapor distribution peaks in the -20 to 20-degree equatorial latitude zones and rapidly decreases toward the polar region at all three pressure levels. There is an asymmetry in the latitude-dependent distribution of water vapor between the northern and southern hemispheres. For example, the northern hemisphere's 0 to 20-degree equator latitude bin has the highest water vapor compared with all other latitude bins, including the southern -20 to 0-degree latitude bin for all three pressure levels. The decrease of water vapor from the low-latitude tropics to the polar region in the southern hemisphere is more rapid than in the northern hemisphere, which results in a higher water vapor concentration in the north latitudinal bins than those corresponding latitudinal bins in the southern hemisphere.

Feulner et al. (2013) showed the asymmetric distribution of annually and zonally averaged surface air temperatures between the northern and southern hemispheres, with the mean surface air temperature in the Northern hemisphere being 1–2°C warmer

than in the southern hemisphere. The close relationship between temperature and the capacity of the atmosphere to hold water vapor is governed by the Clausius-Clapeyron equation (Held and Soden, 2006). The equation states that for every 1-degree Celsius increase in temperature, the saturation vapor pressure increases by about 7%. As temperature increases, this will lead to the potential for more water vapor to be held in the air. In other words, warmer air has a higher capacity to hold water vapor. This relationship is crucial for understanding how temperature changes can impact atmospheric humidity. The observed and modeled evidences presented by Wentz and Schabel (2000), Trenberth et al. (2005), Held and Soden (2006), and Allan et al. (2014), supports the notion that higher atmospheric water vapor contents are, in general, associated with higher temperatures.

Since the warmer temperature is closely coupled with a higher water vapor evaporation rate, our findings of moister high-latitude zones in the northern hemisphere are consistent with the interhemispheric temperature difference observed in Feulner et al. (2013). Furthermore, Feulner et al. (2013) examined climatological data, Earth's energy budget, and model simulations for factors that could lead to interhemispheric temperature differences. The study of Feulner et al. (2013) compared various factors, including seasonal differences in solar radiation, the tropical land area difference, the difference in albedo and temperature between Antarctic and Arctic polar regions, as well as cross-equatorial ocean heat transport from the southern hemisphere to the northern hemisphere. It was shown by Feulner et al. (2013) that for the preindustrial climate, the northward meridional heat transport by ocean circulation, with an additional contribution from the albedo differences between the northern and southern polar regions, are the dominant factors for the interhemispheric temperature difference. As greenhouse gas emissions continued to rise throughout the industrial era, interhemispheric temperature disparities became larger. This is attributed to the intensified warming of land areas compared to oceans and the significant reduction of Arctic sea ice and snow cover in the northern hemisphere. These factors, including cross-equatorial ocean heat transport, albedo difference in polar regions, intensified warming of land areas, and reduction of Arctic ice/snow cover, which affect interhemispheric temperature difference, can also be the primary driving factors of the interhemispheric water vapor difference.

The comparisons between COSMIC and ERA5 water vapor at three pressure levels shown in the middle and right columns of Fig. 3 show some latitude-dependent differences. At the 300 hPa pressure layer, the mean difference and relative difference $\Delta Q_{COSMIC-ERA5}$ (%) are all positive (Fig. 3b and 3c), i.e., $Q_{COSMIC}$ being higher than $Q_{ERA5}$. The peak relative differences (~7-8%) occur in the two equatorial latitude bins (-20 to 0-degree and 0 to 20-degree bins). The percent difference values range from 2% to 8% over the eight latitudinal bins. This suggests that the 5.67% bias in the global $Q_{COSMIC}$ versus $Q_{ERA5}$ comparison mainly comes from the water vapor difference near the equator.

At the 500 hPa pressure level, the $\Delta Q_{COSMIC-ERA5}$ (Fig. 3e) are negative for all the latitude bins, with the amplitude of the water vapor difference being low in the equatorial latitude bins, which is different from those at 300 hPa (Fig. 3b) and 850 hPa (Fig. 3h). At this pressure layer, the mean $Q_{COSMIC}$ is entirely consistent with the mean $Q_{ERA5}$, i.e., $\Delta Q_{COSMIC-ERA5}$ (%) is

within -0.5% as shown in Fig. 3f, in the -20 to 20-degree latitude bins around the equator. Away from the equator, the percent
difference $\Delta Q_{COSMIC-ERA5}(\%)$ increases to around -3%.

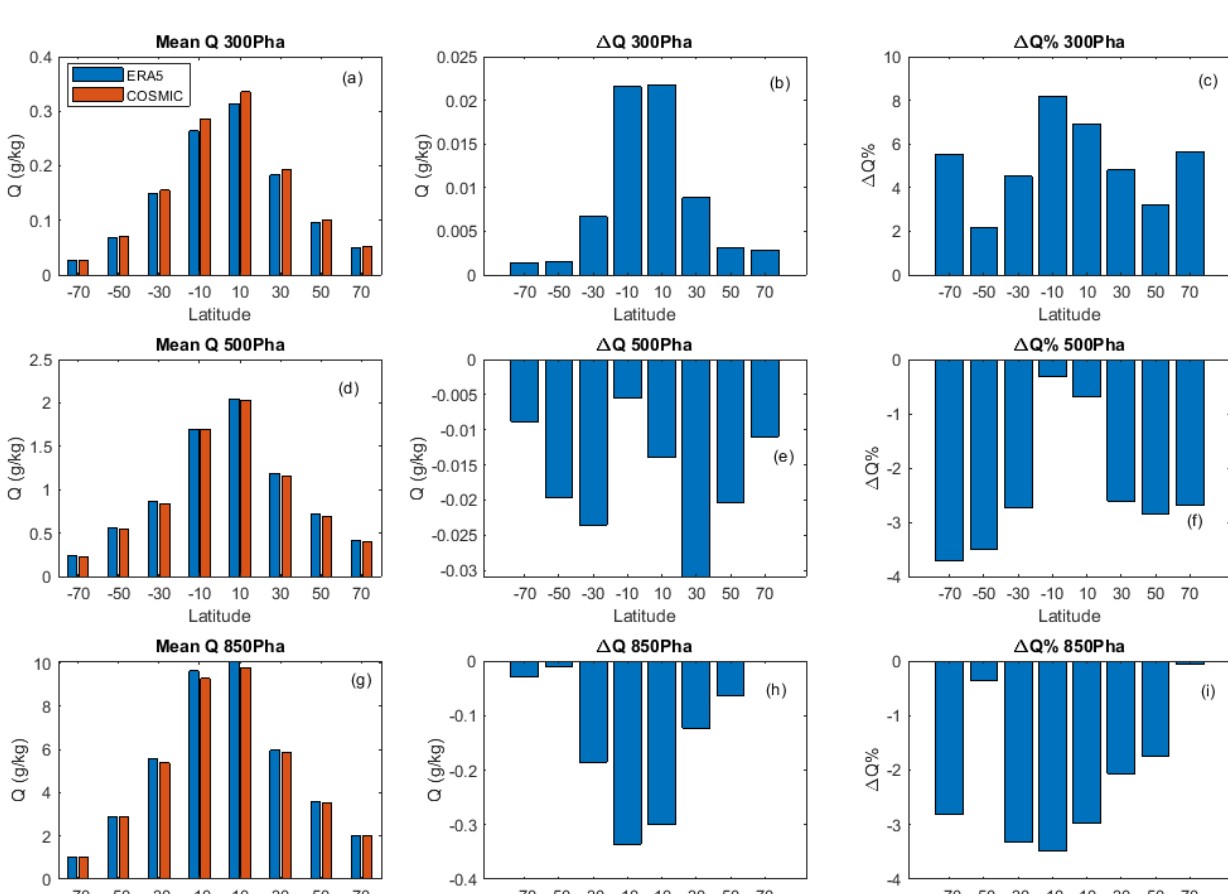

**Figure 3: (a, d, g) Comparison of bin-mean water vapor between COSMIC retrieved data and ERA5 data at three pressure levels. Panels (b, e, h) and (c, f, i) show the value-difference and percent-difference (COSMIC minus ERA5) of latitude bin-mean water vapor data between COSMIC retrieved data and ERA5 data, respectively. The Top, middle, and bottom rows show the comparisons**
**at 300, 500, and 850 hPa, respectively. In all bar-chart panels, the bar centers on the x-axis are placed at the centers of the 20-degree latitudinal bins. For this figure, collocated COSMIC and ERA5 water vapor data for all months of the considered 12-year period (2007-2018) have been used to calculate the mean water vapor in the corresponding latitude bins.**

At the 850 hPa near-surface level, a consistent latitudinal pattern is evident (Fig. 3h and 3j), characterized by negative biases
in $\Delta Q_{COSMIC-ERA5}$ across all eight latitude bins under investigation. From Fig. 3h, it can be seen that the amplitudes of negative
$\Delta Q_{COSMIC-ERA5}$ are dominantly distributed over the -40 to 40-degree latitude zone while peaking at the -20 to 20-degree equator
zone, which agrees with the occurrence of negative water vapor bias in the COSMIC 1DVAR retrieval due to super refraction
in the near-surface moisture-rich low latitude regions (Ho et al., 2010). From Fig. 3i it can be seen that $\Delta Q_{COSMIC-ERA5}(\%)$ of
all latitude bins have negative differences around -2% to -3% except for two latitude bins (-60 to -40 degree and 60 to 80
degree) which have smaller negative $\Delta Q_{COSMIC-ERA5}(\%)$ near zero.

## 4. COSMIC and ERA5 Water Vapor Time Series Analysis and Trend Comparison

With six satellites, COSMIC occultations generally have uniform spatial and temporal distributions. However, because the daily sample number of COSMIC occultations decreased dramatically after 2010 (see Fig. A.4e in Appendix), we need to remove the COSMIC sampling uncertainty for the trend calculation. A detailed description of the method to remove sampling uncertainty, i.e., sampling error removal and calculating trends from water vapor time series data, can be found in Appendix A.2 and is not further described here. This section compares the water vapor trends derived from the COSMIC and ERA5 time series data after removing sampling error and deseasonalization. This section calculates and compares the global and latitude-dependent water vapor trends from the collocated COSMIC and ERA5 data from 2007 to 2018 at three pressure levels (300, 500, and 850 hPa).

### 4.1 Comparison of global COSMIC and ERA5 water vapor trends

Figure 4a shows the time series of global mean COSMIC and ERA5 water vapor at three pressure levels. At 300 hPa, COSMIC water vapor data is consistently higher than ERA5 data. At 500 and 850 hPa, the COSMIC water vapor data is slightly lower than the ERA5 data. These differences between COSMIC and ERA5 are consistent with the bias analysis in Section 3.1. Figure 4a shows that although the COSMIC and EAR5 time series are different, their trends are pretty close (Figure 4b), which will be further quantified after the time series data are deseasonalized.

It can be seen in Fig. 4a that there were two abnormal water vapor increases around 2010 and 2015-2016 in both the COSMIC and ERA5 time series at all three pressure levels. The abnormal increases in water vapor around 2010 and 2015-2016 were also observed in the long-term total precipitable water monitoring (Mears et al., 2022), which used multiple-RO sensors and radiosonde data to construct the time series data. These abnormal water vapor increases were attributed to El Niño, i.e., the warm phase of the El Niño Southern Oscillation (ENSO). These warm events can enhance surface evaporation, increase tropospheric water vapor, and warm the entire tropical troposphere (e.g., Zveryaev and Allan 2005; Trenberth et al. 2005). The recent 2015-2016 El Niño event broke warming records in the central Pacific according to Niño3.4 (sea surface temperature (SST) anomalies averaged over the equatorial region (Latitude: -5º to 5º; Longitude: -150º to 160º) of the Pacific Ocean) and Niño4 indices (SST anomalies over the region (Latitude: -5º to 5º; Longitude: -150º to 160º)). The 2015-2016 El Niño event was among the most significant events recorded in this century. During the El Niño event from April 2015 to May 2016, the equatorial Pacific Ocean waters stayed warm for a whole year, reaching peak temperatures in November 2015 (https://www.ecmwf.int/en/newsletter/151/meteorology/2015-2016-el-nino-and-beyond). The long period of warm Pacific Ocean temperature significantly impacted the global weather patterns and diminished the seasonal cycles. This also caused anomalies in the seasonal variation of the 2015-2016 global atmospheric water concentration through the coupling between ocean and atmosphere over the equatorial Pacific Ocean and the atmospheric winds (Fig. 4a).

To quantitatively evaluate the trend of global water vapor, Figure 4b shows the time series of sampling error-removed and deseasonalized monthly-mean global water vapor of COSMIC and ERA5 at three pressure levels. The slope values, i.e., long-term trends, are derived with linear regression and listed in Table 1 in both units of g/kg/decade ($D_Q$) and %/decade ($ND_Q$). In
calculating the percent/decade trend, *i.e.*, normalized trend ($ND_Q$), the long-term averaged global mean water vapor (g/kg) at a given pressure level has been used to normalize the trend with the unit g/kg/decade.

COSMIC and ERA5 water vapor trend data (Fig. 4) show that the global water vapor trends at three pressure levels are all positive, suggesting the increase of global water vapor concentration during the period from 2007 to 2018, i.e., becoming
globally moister, over time at these pressure levels. Many earlier studies have reported a rise of global atmospheric water vapor in different periods, e.g., over the period 1979-2001 with ERA-40 reanalysis (Bengtsson 2004), over the period 1976-2004 using global meteorological data measured by weather stations and marine ships (Dai 2006), and over 1996-2002 with Global Ozone Monitoring Experiment (GOME) data (Wagner et al. 2006). In Chen and Liu (2016), five global PWV data sets, e.g., ECMWF and NCEP reanalysis data, radiosonde, ground GPS stations, and microwave satellite measurements, over the period
2000-2014, were used to derive the trend, and all show positive global PWV trend. Allan et al. (2022) studied the global-scale changes in water vapor and responses to surface temperature variability since 1979 using coupled and atmosphere-only CMIP6 climate model simulations. In the water vapor trend estimation over the 1988 to 2014 period, Allan et al. (2022) showed a positive increase of global water vapor at the near-surface, at 400 hPa and Column Integrated Water Vapor from an ensemble of climate model simulations with the CMIP6 historical and Atmospheric Model Intercomparison Project (AMIP) experiments.
The period of COSMIC RO data studied in this paper (2007 to 2018) partially overlaps with the simulations of Allan et al. (2022). The increasing trend in the global atmospheric water vapor concentration at the three pressure levels considered in our trend analysis is generally consistent with the results from Allan et al. (2022). It was suggested that an increasing trend in water vapor could be the response to the surface temperature increase (Held and Soden, 2006; Santer et al., 2006; Zhang et al., 2013).

Table 1 shows that the increasing trends of global water vapor vary from ~2 to ~4 %/Decade from the analysis of both COSMIC and ERA5 data at the three pressure levels. It was also shown by Allan et al. (2022) that in the ensembled historical experimental model simulations, the water vapor increases by 1.53 and 3.52 %/Decade at the surface and at 400 hPa, respectively. Our study shows that the increasing global water vapor trend estimated for the COSMIC data over the period 2007-2018 are 2.03±0.65, 3.25±1.25, 3.47±1.47 %/Decade at 850, 500, and 300 hPa, respectively, which is in general
agreement with the results from in Allan et al. (2022), considering that the two work cover two distinct periods with 8 overlapping years. In Allan et al. (2022), there is an increase in water vapor trend from the surface to 400 hPa by ~2 %/Decade. Our work shows an increase of water vapor trend by 1.44 %/ Decade when pressure level varies from the near-surface (at 850 hPa) to 300 hPa, which is generally consistent.

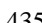

Figure 4: (a) Monthly-mean time series of COSMIC and ERA5 global mean water vapor data at three pressure levels (solid lines) and linear trend (dashed lines). (b) Time series of sampling error-removed and deseasonalized monthly-mean COSMIC and ERA5 global water vapor data (solid lines) and linear trend (dashed lines). In all panels, red and blue lines are time series (solid lines) and trends (dashed lines) of ERA5 and COSMIC water vapor data, respectively.

**Table 1: Comparison of the global water vapor trends (slope ± 95% Confidence Interval) derived from COSMIC and ERA5 data.**

| Pressure Level | COSMIC $Q$ Trend ($D_{Q,COSMIC}$, g/kg/Decade) | Normalized COSMIC $Q$ Trend ($ND_{Q,COSMIC}$, %/Decade) | ERA5 $Q$ Trend ($D_{Q,ERA5}$, g/kg/Decade) | Normalized ERA5 $Q$ Trend ($ND_{Q,ERA5}$, %/Decade) |
|---|---|---|---|---|
| 300 hPa | $0.0047 \pm 0.0024$ | $3.47 \pm 1.77$ | $0.0046 \pm 0.0022$ | $3.58 \pm 1.71$ |
| 500 hPa | $0.0275 \pm 0.0106$ | $3.25 \pm 1.25$ | $0.0355 \pm 0.0107$ | $4.12 \pm 1.24$ |
| 850 hPa | $0.0912 \pm 0.0293$ | $2.03 \pm 0.65$ | $0.1302 \pm 0.0311$ | $2.83 \pm 0.68$ |

The increasing trend values at 300 hPa derived from COSMIC and ERA5 global water vapor data are consistent. At 500 hPa and 850 hPa, the $ND_{Q,ERA5}$ are higher than COSMIC trends by 0.87%/Decade and 0.8%/Decade, respectively, which suggests that ERA5 may over-estimate the increase of water vapor during 2007 to 2018. Chen and Liu (2016) showed that the increasing PWV trend from 2000 to 2014 derived from ECMWF data is ~0.37%/Decade larger than the PWV trend derived from the
ground GPS station data. The difference between $ND_{Q,ERA5}$ and $ND_{Q,COSMIC}$ from our analysis at 500 hPa and 850 hPa are about 0.5%/Decade higher than the differences between the trends of ECMWF and ground GPS station PWV data studied by Chen and Liu (2016).

Using the trend results from COSMIC data, we can also see that water vapor trends increase with lower pressure levels. Table
1 shows that the increasing trend at 850 hPa from COSMIC data ($ND_{Q,COSMIC}$) is lower by 1.44 and 1.22 %/Decade than at 300 and 500 hPa, respectively.

### 4.2 Comparison of COSMIC and ERA5 latitudinal water vapor trends

To further understand the latitudinal distribution of the water vapor trends, we calculate the slopes of the linear fit for COSMIC ($D_{Q,COSMIC}$) and ERA5 ($D_{Q,ERA5}$) at eight 20° latitudinal bins distributed from -80° to 80°. The latitudinal bins above 80° in the
northern and southern polar regions are excluded from this analysis due to too few COSMIC RO observations. Figure 5 compares slope values of the linear fit of water vapor between COSMIC and ERA5 over eight latitude bins at three pressure levels. The first column of Fig. 5 shows the water vapor trends ($D_Q$) of unit g/kg/Decade. To account for the latitudinal variation of water vapor, the middle column of Fig. 5 shows the water vapor trends ($ND_Q$) normalized by the corresponding long-term latitude-bin-averaged water vapor mean and expressed with the unit of %/Decade. The third column of Fig. 5 shows the
latitude-dependent water vapor trend difference ($ND_{Q,COSMIC}$ - $ND_{Q,ERA5}$, %/Decade) between COSMIC and ERA5. Table 2 lists the water vapor trend values of COSMIC and ERA5 for eight latitude bins and at three pressure levels.

From Fig. 5, the latitude-mean water vapor trends are mostly positive (increasing), and their magnitudes vary with latitude bins substantially at three pressure levels. The only latitude bin with a small negative water vapor trend with large uncertainty
is in the -80° to -60° southern high latitude bin at 500 hPa. From the global surface temperature trend analysis by Gu and Adler,

2022, there is a mixture of a weak decreasing trend in the surface temperature in the Southern Ocean around the Antarctic and an increasing trend over the Antarctic in the -80° to -60° southern latitude bin. However, the uncertainties of estimating the temperature and water vapor trends in this latitude zone are large.

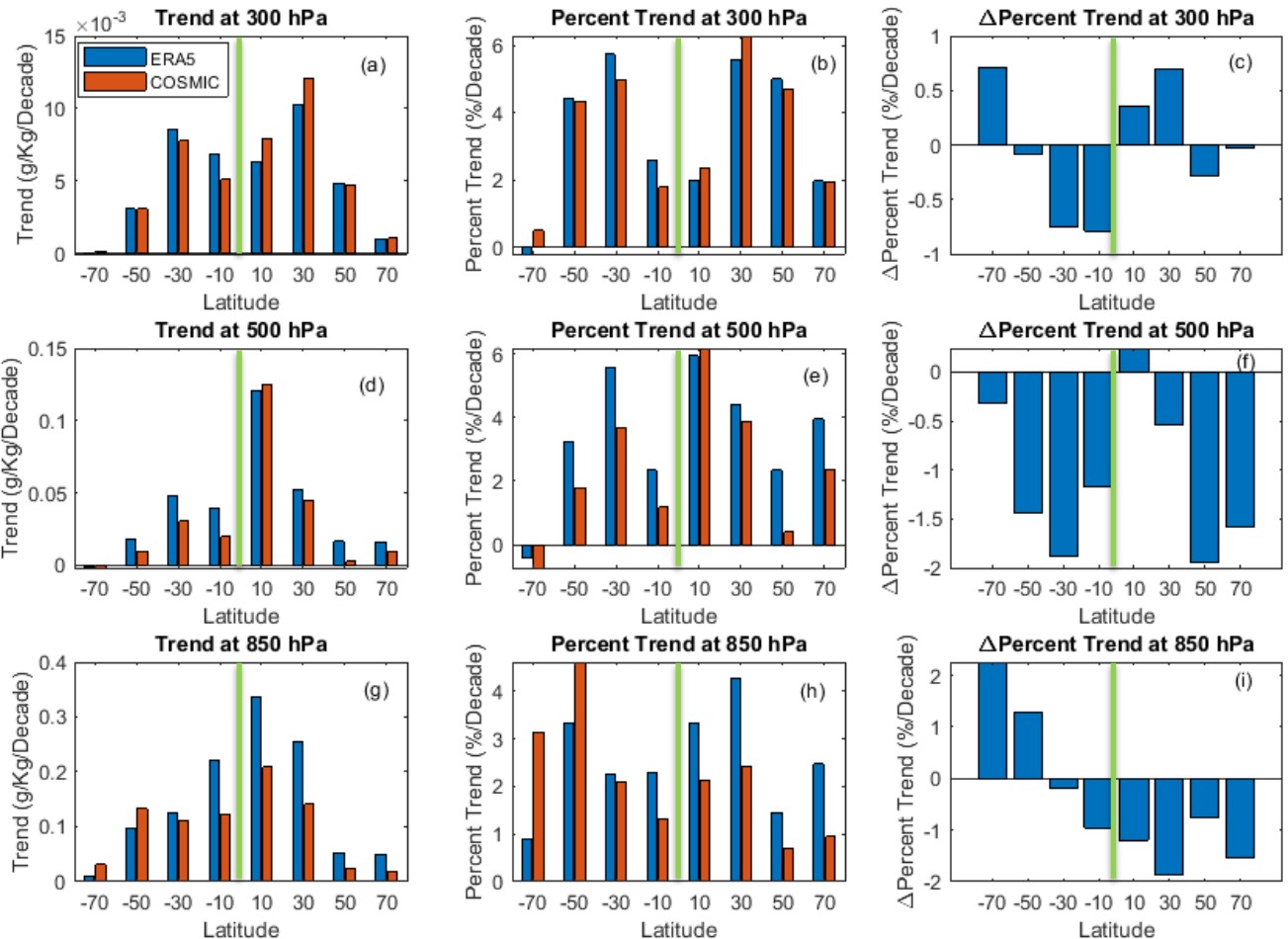

**Figure 5: (a, d, g) Comparison of the latitude bin-mean water vapor trends (g/kg/Decade) between COSMIC and ERA5 data at 300, 500, and 850 hPa, respectively. (b, e, h) Comparison of normalized latitude bin-mean water vapor trends (%/Decade) between COSMIC and ERA5 data at 300, 500, and 850 hPa, respectively. (c, f, i) The difference (COSMIC minus ERA5) of normalized latitude bin-mean water vapor trend (%/Decade) between COSMIC and ERA5 data at 300, 500, and 850 hPa, respectively. The $x$ values on the horizontal axis represent the centers of the 20° latitude bins. The green line in each panel separates the southern (to its left) and northern (to its right) hemispheres.**

At 300 hPa, the differences in water vapor trends (Fig. 5c) between COSMIC ($ND_{Q,COSMIC}$) and ERA5 ($ND_{Q,ERA5}$) consist of positive and negative values and with magnitudes being less than 0.8 %/Decade over the eight latitude bins. In other words, the COSMIC and ERA5 water vapor trends are consistent within 0.8 %/Decade over all eight latitude bins. In Fig. 5b, the

trends of water vapor change in the four latitude bins over the -60º to -20º and 20º to 60º zones are in the range of 4 to ~6

%/Decade, which is higher than the water vapor trends (1.79 to 2.58 %/Decade) of the two equatorial latitude bins (0º to 20º

and -20 º to 0º). The southern -80º to -60º latitude bin has the lowest water vapor trends (both $\left|ND_{Q,ERA5}\right|$ and $\left|ND_{Q,COSMIC}\right|$ <

0.6%/Decade) at 300 hPa among the eight latitude bins studied in this paper.

**Table 2: Latitude bin-mean water vapor trends (g/kg/Decade and %/Decade) and 95% confidence interval estimated from COSMIC and ERA5 data at 300, 500, and 850 hPa**

| Latitude Bin | At 300 hPa | | At 500 hPa | | At 850 hPa | |
|---|---|---|---|---|---|---|
| | $(D_{Q,COSMIC},$ $D_{Q,ERA5})$ (g/kg/Decade) | $(ND_{Q,COSMIC},$ $ND_{Q,ERA5})$ (%/Decade) | $(D_{Q,COSMIC},$ $D_{Q,ERA5})$ (g/kg/Decade) | $(ND_{Q,COSMIC},$ $ND_{Q,ERA5})$ (%/Decade) | $(D_{Q,COSMIC},$ $D_{Q,ERA5})$ (g/kg/Decade) | $(ND_{Q,COSMIC},$ $ND_{Q,ERA5})$ (%/Decade) |
| -80º to -60 º | 0.0001±0.0016, -0.00005±0.0016 | 0.52±5.96, -0.19±6.13 | -0.00±0.01, -0.00±0.01 | -0.72±6.48, -0.41±6.14 | 0.03±0.04, 0.01±0.04 | 3.14±3.87, 0.88±3.61 |
| -60º to -40 º | 0.0031±0.0039, 0.0031±0.004 | 4.34±5.58, 4.43±5.72 | 0.01±0.02, 0.02±0.02 | 1.80±3.74, 3.25±3.66 | 0.13±0.06, 0.10±0.06 | 4.61±2.00, 3.34±1.94 |
| -40º to -20 º | 0.008±0.0065, 0.0085±0.0064 | 4.98±4.16, 5.74±4.30 | 0.03±0.03, 0.05±0.03 | 3.67±3.85, 5.55±3.74 | 0.11±0.08, 0.13±0.08 | 2.09±1.50, 2.27±1.41 |
| -20º to -0 º | 0.0051±0.0098, 0.0068±0.0091 | 1.79±3.42, 2.58±3.44 | 0.02±0.06, 0.04±0.06 | 1.17±3.52, 2.35±3.50 | 0.12±0.12, 0.22±0.13 | 1.33±1.31, 2.29±1.32 |
| 0º to 20 º | 0.0079±0.01, 0.0063±0.0094 | 2.36±3.04, 2.00±2.98 | 0.13±0.05, 0.12±0.06 | 6.17±2.71, 5.93±2.71 | 0.21±0.10, 0.34±0.10 | 2.14±1.06, 3.34±1.02 |
| 20º to 40 º | 0.012±0.007, 0.01±0.007 | 6.29±3.56, 5.59±3.61 | 0.04±0.03, 0.05±0.03 | 3.88±2.95, 4.41±2.80 | 0.14±0.09, 0.25±0.09 | 2.41±1.61, 4.27±1.56 |
| 40º to 60 º | 0.0047±0.0044, 0.0048±0.0044 | 4.72±4.40, 5.01±4.48 | 0.00±0.02, 0.02±0.02 | 0.40±3.14, 2.35±3.17 | 0.02±0.08, 0.05±0.08 | 0.69±2.23, 1.46±2.27 |
| 60º to 80 º | 0.001±0.0031, 0.001±0.003 | 1.94±5.99, 1.98±6.32 | 0.01±0.02, 0.02±0.02 | 2.37±5.38, 3.95±5.25 | 0.02±0.07, 0.05±0.07 | 0.94±3.49, 2.48±3.44 |

At 500 hPa, both $D_{Q,COSMIC}$ and $D_{Q,ERA5}$ are the highest (~0.13 g/kg/Decade) in the 0 º to 20º latitude bin (Fig. 5d). Regarding

the normalized trends of the unit %/Decade, the $ND_{Q,COSMIC}$ and $ND_{Q,ERA5}$ (%/Decade) are all positive except in the -80º to -

60º latitude bin. Over the latitude bins in the -60 º to 80º latitude zone, the values of $ND_{Q,ERA5}$ vary between 2.35 and 5.93

%/Decade while values of $ND_{Q,COSMIC}$ vary between 0.4 and 6.17 %/Decade. The water vapor trends of $ND_{Q,COSMIC}$ and

$ND_{Q,ERA5}$ in the -80º to -60º latitude bin are both quite stable with a weak negative trend of -0.72 %/Decade. Figure 5f shows

the difference between $ND_{Q,COSMIC}$ and $ND_{Q,ERA5}$ are all negative (-2 to -0.3 %/Decade) except for one small positive

difference (0.24%/Decade) at the 0º to 20º latitude bin. The smaller global water vapor trend from COSMIC at 500 hPa

compared to the trend from ERA5, as shown in Table 1, mainly comes from the latitude bins with negative $ND_{Q,COSMIC}$ -

$ND_{Q,ERA5}$ (Fig. 5f). This analysis indicates that at 500 hPa, both ERA5 and COSMIC water vapor data confirm the increasing

trends in all the latitude zones from -60º to 80º, and the trends estimated from COSMIC water vapor data are lower than those from ERA5 in most latitude bins except the 0 º to 20º equatorial bin.

At 850 hPa, the water vapor trends are all positive from the COSMIC and ERA5 data analysis over eight latitude bins at three pressure levels (Figure 5g and Table 2). Regarding the absolute water vapor trend, i.e., of unit g/kg/Decade, the water vapor growth peaks in the 0º to 20º bin and decreases as the latitude increases toward higher latitudes. The overall magnitudes of water vapor trends are larger than 0.1 g/kg/Decade from ERA5 and COSMIC data estimated for all latitude bins in the -40º to 40º latitude zone. The $D_{Q,ERA5}$ is larger by 0.1 to 0.13 g/kg/Decade than $D_{Q,COSMIC}$ in all of the latitude bins from -20º to 40º.

The normalized water vapor trends in Fig. 5h and Table 2 show that both $ND_{Q,COSMIC}$ and $ND_{Q,ERA5}$ have substantial variabilities (between 0.69 to 4.61 %/Decade) among all of the latitude bins. Figure 5i shows that $ND_{Q,COSMIC}$ is lower than $ND_{Q,ERA5}$ over all the latitude bins from -40º to 80º and $ND_{Q,COSMIC}$ is larger than $ND_{Q,ERA5}$ over all the latitude bins from -80º to -40º. The magnitudes of the difference ($ND_{Q,COSMIC}$- $ND_{Q,ERA5}$) in all the latitude bins from -60º to 80º are less than 2%/Decade. This indicates that the relatively lower global water vapor trends estimated from COSMIC data compared to

ERA5 data at the 850 hPa level (as presented in Table 1) are mainly due to the lower values of COSMIC trends within the middle and low latitude bins.

## 5. Regional Comparisons of COSMIC and ERA5 Water Vapor Trends

### 5.1 Global map of the 10º×10º COSMIC and ERA5 water vapor trends

To quantify and compare the global distribution of the regional water vapor trends derived from COSMIC and ERA5 data, we
grouped the collocated global water vapor data over 12 years (2007-2018) into 10º×10º latitude/longitude grids. We followed the procedure of estimating the water vapor trend outlined in Appendix A.2 to calculate the trends ($D_{Q,COSMIC}$, $ND_{Q,COSMIC}$, $D_{Q,ERA5}$, $ND_{Q,ERA5}$) for the globally distributed 10º×10º RoIs. When the grid size is limited to 10º×10º, there are missing monthly data for specific RoI due to the limited orbital coverage of COSMIC. Figure 6 shows the percentage of missing monthly data distribution over the 2007 to 2018 period in the global 10º×10º grids. The grids with no missing monthly data during this
period are shown as white blanks. The grids with substantial missing monthly data are mostly found over northern and southern polar regions with latitudes greater than 70 degrees. Missing COSMIC RO data is prominent over the regions covering the Tibetan Plateau, specifically at pressure levels of 500 and 850 hPa. The absence of RO data in these regions can be attributed to the lower atmospheric pressure prevailing over areas at an average altitude of around 4 km. Our 10º×10º RoI-based trend analysis excludes the grids with more than 1.5% missing monthly data at 850 hPa. In other words, grids with > 2-month
missing monthly data are excluded from the trend calculation. The effects of sampling error removal on regional water vapor trend analysis uncertainty are discussed in Appendix A.3.

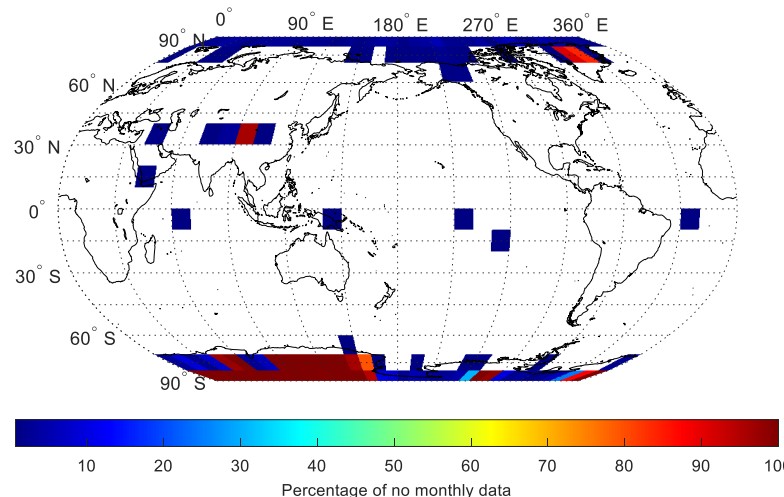

**Figure 6: The percentage of missing monthly data over the 2007 to 2018 interval on the global 10°×10° grids. The percentage of missing data is shown as color-coded. Grids with complete monthly data and without gaps, i.e., covering all months, are represented as white blank spaces.**

Figure 7 shows the global distribution of COSMIC and ERA5 water vapor trends ($D_{Q,COSMIC}$ and $D_{Q,ERA5}$) and their difference ($D_{Q,COSMIC} - D_{Q,ERA5}$) at 500 and 850 hPa. The distributions of COSMIC and ERA5 water vapor trends at 300 hPa have smaller regional variations. They are not shown in Fig. 7. In Section 4, Figure 4 and Figure 5 suggest that the global water vapor trends are increasing. The latitude-bin-based water vapor trends are increasing in low and middle latitudes at all three pressure levels we studied. Figure 7a-d shows that both COSMIC and ERA5 data indicate substantial regional variabilities in the global distribution of the water vapor trends. The magnitude of water vapor trends peaks near the equator and decreases as it approaches the polar regions, where the atmosphere is drier.

Near the equator, at 500 hPa and 850 hPa, both $D_{Q,COSMIC}$ and $D_{Q,ERA5}$ are strongly positive, i.e., becoming moister over time, around 180° to 240° longitude and 10° to 20° latitude range in the equatorial Pacific Ocean. This region in the Pacific Ocean with a strong positive water vapor trend is encased at the west side by two regions with negative water vapor trends located around (Latitude: 20°; Longitude: 130°) and (Latitude: -10°; Longitude: 130°) which are on the northern and southern side, respectively. These two regions are located between the western Pacific and the eastern Indian Ocean, where sizeable regional moisture flux convergence occurs (Fig. 2). Such a pattern of strong increasing water vapor trend in the equatorial Pacific Ocean and decreasing water vapor trend near the region between the west Pacific and the east Indian Ocean are more prominent at 500 hPa than at 850 hPa. At 500 hPa, the negative water vapor trends are extended to northern Australia and southern Asia, covering the Indo-Pacific warm pool region.

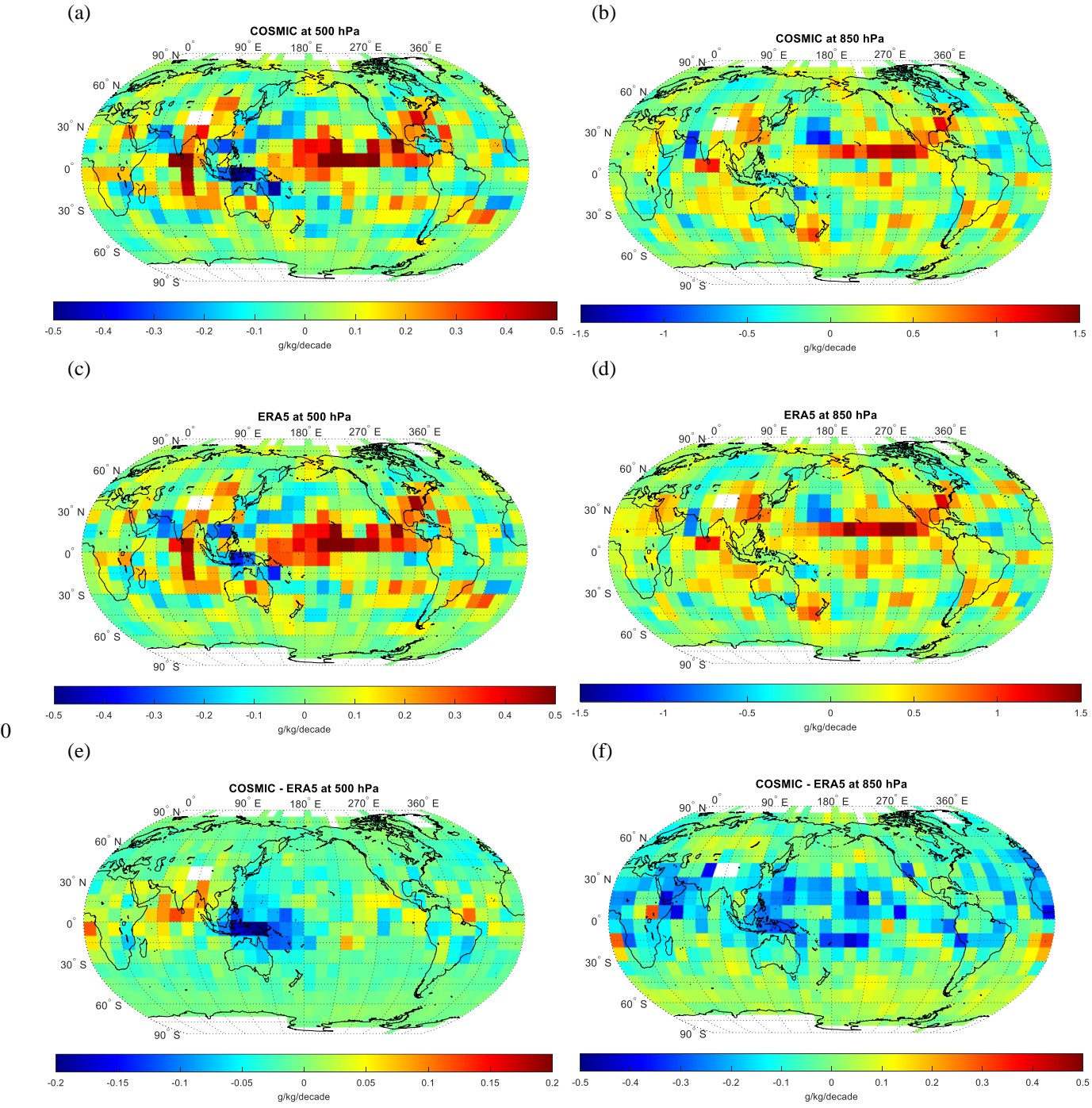

**Figure 7: (a, b) and (c, d) The global distribution of water vapor trends (g/kg/Decade) in 10º × 10º grids derived from long-term COSMIC (a, b) and ERA5 (c, d) data, respectively; (e, f) The global distribution of the water vapor trend difference (g/kg/decade) between COSMIC and ERA5 (COSMIC minus ERA5). The left and right columns are derived with water vapor data at 500 and 850 hPa, respectively.**

Sea surface temperature has been increasing in the western Pacific during recent decades (e.g., Gu and Adler 2022). There is
a high correspondence with regard to the trends in sea surface temperature and tropospheric water vapor in the western Pacific
during the recent decades (e.g., Gu and Adler 2013). It was shown by Chen and Liu (2016) that the moderate increase in
surface temperature over the Pacific Ocean could cause the PWV to increase in the equatorial region of the Pacific Ocean and
decrease in this Indo-Pacific warm pool region, which is what we observe here. Further quantitative analysis of trends at
selected locations in the Pacific Ocean (Site # 6 in Fig. 8) and the Indo-Pacific warm pool region (Site #4 in Fig. 8) will be
performed in the following sections.

In the Indian Ocean, the region in the Laccadive Sea near the northern edge of the Indian Ocean (Latitude: 0º to 10º; Longitude:
70º to 90º) has strong increasing water vapor trends at 850 hPa. At 500 hPa, the region with strong positive water vapor trends
expands to a larger region (Latitude: -20º to 10º; Longitude: 80º to 90º). This region is affected by the monsoon climate over
the south of the Himalayas. The monsoon climate influences water vapor variability and trends through moisture transport (An
et al., 2015; Turner and Annamalai, 2012). The variability in water vapor trends in a region experiencing a monsoon climate
is closely tied to the alternating wet and dry phases. Factors such as the strength and duration of the monsoon, the temperature
of the ocean waters, and atmospheric circulation patterns all play a role in determining the extent of moisture transport and its
impact on water vapor levels. Changes in sea surface temperatures due to global warming can affect the intensity and timing
of monsoon patterns, leading to shifts in moisture transport and potentially altering the variability of water vapor content in
the affected regions. Indian Ocean is an essential part of the coupled Indian monsoon system because it feeds the moist
convection over both land and ocean. It is shown that the Indian Ocean has been warming up in recent decades (Gu and Adler,
2022). The warming up of the Indian Ocean can be the main driver for this region's positive water vapor trend (Latitude: 0º to
10º; Longitude: 70º to 90º). The region near the Gulf of Oman in the Arabian Sea (Latitude: 10º to 30º; Longitude: 60º to 70º)
has strong decreasing water vapor trends at 850 hPa. At 500 hPa, this region with negative water vapor trends expands to the
area with Latitude: 10º to 30º; Longitude: 50º to 80º and covers the northern coast. The variability of the water vapor trends in
this region may arise from the moisture transport influenced by the monsoon climate.

Over the land, a significantly increasing water vapor trend at 850 hPa can be observed around the region in the eastern United
States (Latitude: 30º to 40º; Longitude: 270º to 280º) and over the region near southeastern China (Latitude: 20º to 40º;
Longitude: 110º to 130º).

Figures 7e and 7f show the spatial distribution of the $D_{Q,COSMIC}$ - $D_{Q,ERA5}$. i.e., the water vapor trend differences between
COSMIC and ERA5, at 500 and 850 hPa, respectively. At 500 hPa, the negative differences ($D_{Q,COSMIC} < D_{Q,ERA5}$) are
primarily distributed in the regional box (latitude: -10º to 10º; longitude: 120º to 170º) where the Indo-Pacific Ocean region is
located, and the decreasing water vapor trends are observed by both COSMIC and ERA5. The difference is positive at 500
hPa, i.e., $D_{Q,COSMIC} > D_{Q,ERA5}$, in the northern Indian Ocean and near its north coast. At 850 hPa, the difference is primarily

negative, with the COSMIC trend being lower than ERA5 in tropical areas. Such dominantly negative differences between $D_{Q,COSMIC}$ and $D_{Q,ERA5}$ in tropical regions (30°S to 30°N) at 850 hPa determines the lower global and low-latitude $D_{Q,COSMIC}$

in comparison with $D_{Q,ERA5}$ as shown in Table 1, 2 and Fig. 5.

In the following sections, we selected a few representative 10°×10° grids (sites) to quantitatively characterize the spatial variability of COSMIC and ERA5 water vapor trends. The center locations of these selected 10°×10° grids are shown in Fig. 8. These sites include stratocumulus cloud-rich sites (Sites #1-3 discussed in Section 5.2) and sites with a notable difference

between ERA5 and COSMIC trends (Sites #4-7 discussed in Section 5.3), which can help quantitatively understand the regional difference of water vapor trends between COSMIC and ERA5. To quantitatively characterize the large regional variabilities, i.e., mixed with strong increasing and decreasing, of water vapor trends shown in Fig. 7a-d, we also identified and analyzed several sites with notable increasing (moister, Sites #8-#12) and decreasing (drier, Sites #13-#17) water vapor trends. The analysis results for these sites are presented in Appendix A.4.

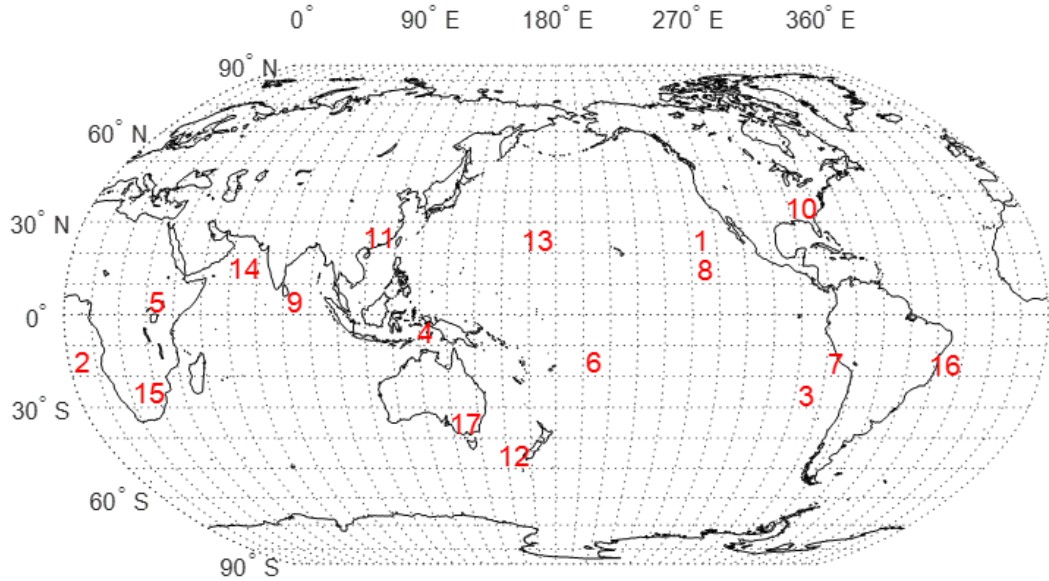

**Figure 8: Center locations of selected sites for regional analysis of water vapor trends.**

## 5.2 Water vapor trends over stratocumulus cloud-rich regions

The first set of sites (Sites #1-3 in Fig. 8) we selected is over stratocumulus cloud-rich regions. These three sites are selected

according to the stratocumulus cloud-rich regions identified by Wood et al. 2011; Wood, 2012; Ho et al., 2015. Stratocumulus clouds are typically shallow and occur at low altitudes (below 2 km) due to weak convective currents with drier and stable air above, preventing continued vertical development. Stratocumulus clouds usually occur over subtropical and polar oceans. Over

regions with frequent stratocumulus clouds, it is challenging to accurately estimate water vapor at low altitudes in the ECMWF assimilation (Lonitz and Geer 2017).

On the other hand, the RO signal can penetrate the cloud layer because the wavelengths for L1 and L2 frequency of RO signals are around 19 cm and 24.2 cm, respectively, which are much larger than the size of cloud water droplets and ice crystals (Kursinski et al., 1997). The water vapor retrieval from RO data is not affected by the cloud. This study helps to determine whether there are differences between COSMIC and ERA5 water vapor trends over these stratocumulus cloud-rich regions and

630 quantify the difference. Therefore, we compare the near-surface water vapor trend of ERA5 and COSMIC at 850 hPa over three stratocumulus cloud-rich regions. Table 3 lists the water vapor trends at 500 and 850 hPa over three sites in the ocean: West of the Baja coast (#1), West of Africa (#2), and West of South America (#3), derived from COSMIC and ERA5 data.

**Table 3: Water vapor trends over three selected stratocumulus cloud-rich sites.**

| | Center (Lat., Long.) Region | At 500 hPa | | | At 850 hPa | | |
|---|---|---|---|---|---|---|---|
| | | $(\overline{Q_{COSMIC}, Q_{ERA5}})$ (g/Kg) | $(D_{Q,COSMIC}, D_{Q,ERA5})$ (g/kg/Decade) | $(ND_{Q,COSMIC}, ND_{Q,ERA5})$ (%/Decade) | $(\overline{Q_{COSMIC}, Q_{ERA5}})$ (g/Kg) | $(D_{Q,COSMIC}, D_{Q,ERA5})$ (g/kg/Decade) | $(ND_{Q,COSMIC}, ND_{Q,ERA5})$ (%/Decade) |
| Site #1 | (25º, 235º) West of Baja coast | 0.77±0.28, 0.82±0.52 | 0.12±0.24, 0.10±0.24 | 15.19±30.63, 12.37±28.79 | 4.12±1.42, 3.83±2.26 | 0.61±0.84, 0.53±0.85 | 14.76±20.34, 13.92±22.10 |
| Site #2 | (-15º, 5º) West of Africa | 0.83±0.49, 0.84±0.74 | 0.09±0.28, 0.03±0.27 | 10.97±33.23, 3.94±32.44 | 4.38±1.55, 4.74±2.13 | 0.36±0.72, 0.07±0.71 | 8.13±16.49, 1.51±14.94 |
| Site #3 | (-25º, 275º) West of South America | 0.49±0.15, 0.52±0.32 | 0.21±0.15, 0.22±0.15 | 42.66±29.67, 42.64±28.13 | 3.91±1.10, 3.91±1.60 | 0.16±0.63, 0.06±0.64 | 4.02±16.06, 1.56±16.24 |

At 850 hPa, COSMIC and ERA5 data show that these three sites have comparable mean water vapor (around 4 g/kg). At 500 hPa, site #3 has a lower mean water vapor than the other two sites. These three sites have positive water vapor slopes at 500 and 850 hPa. At 850 hPa, Site #1 has the strongest increasing trend of water vapor and $ND_{Q,COSMIC}$ (14.76 %/Decade) is comparable to $ND_{Q,ERA5}$ (13.92 %/Decade). For Site #2 and #3, there are significant differences between the trends estimated

with COSMIC and ERA5 data at 850 hPa. For example, the increasing trend estimated from COSMIC ($ND_{Q,COSMIC}$) is about 6.62 %/Decade higher than $ND_{Q,ERA5}$ for Site #2 (Table 3). For Site #3, the $ND_{Q,COSMIC}$ is higher than $ND_{Q,ERA5}$ by 2.46 %/Decade. This analysis indicates that for two of the three selected sites around the stratocumulus cloud-rich regions, the estimated water vapor trends from COSMIC at 850 hPa can be significantly higher than those estimated from ERA5 data. The possible cause of smaller trends from ERA5 water vapor data over stratocumulus cloud-rich regions could be the difficulty in

accurately estimating water vapor at low altitudes from ERA5 reanalysis data compared with COSMIC RO measurements that are unaffected by stratocumulus clouds (Lonitz and Geer, 2017).

**5.3 Sites with a notable water vapor trend difference between ERA5 and COSMIC**

Comparing regional water vapor trends between COSMIC and ERA5 data and quantifying their differences contribute to validating both datasets. In particular, it can identify regions where the reanalysis model could exhibit constraints. In this section, we select a few sites with a notable trend difference between COSMIC and ERA5 to quantitatively understand the magnitude of the differences and the distribution of these sites. To identify these sites, we first searched the $10°\times10°$ global map of the water vapor trend difference between COSMIC and ERA5 (Fig. 7e and 7f shown in Section 5.1). We identified the regions with the largest positive or negative water vapor trend difference between COSMIC and ERA5. Within these regions, we selected one representative $10°\times10°$ grid in each region as the site of interest. The estimated water vapor trends for COSMIC and ERA5 over these sites, with notable trend differences are listed and compared in Table 4. Sites #4 and #6 are over the ocean, and sites #5 and #7 are over land. Sites #4, #5, and #6 are moisture-rich. Site #4 is located in the Indo-Pacific Ocean region, which suggests large uncertainty in the characterization of $D_{Q,ERA5}$ in this region. Site #5 is among the few sites (see Fig. 7f in Section 5.1) with $D_{Q,COSMIC}$ larger than $D_{Q,ERA5}$ (by 0.29 g/kg/Decade) at 850 hPa. Site #6 is the typical low-latitude site with $D_{Q,COSMIC}$ being less than $D_{Q,ERA5}$. For Site #7 in Peru, COSMIC shows a much steeper decreasing trend, lower by -8.34 %/Decade, than ERA5 at 850 hPa. This $10°\times10°$ grid of Site #7 is mixed with the Andes Mountains on the east portion of the grid and the Pacific Ocean on the west. There are no 850 hPa RO data over the Andes Mountains (over 6 km in altitude) area. The RO water vapor trend data mainly come from the Pacific Ocean in the $10°\times10°$ grid of Site #7. The COSMIC water vapor trend indicates that Site #7 has decreased near-surface water vapor from the period 2007 to 2018, while ERA5 data suggests no significant long-term change in the amount of water vapor. From the linear trend study of global surface temperature during 1998-2020 by Gu and Adler, 2022, there is a trend of decreasing ocean surface temperature (~-0.1 K/Decade) near Site #7, which matches the decrease of water vapor observed by COSMIC. Site #7 is situated in close proximity to Site #3 and falls within an area where there is a frequent presence of low-height stratocumulus clouds (Wood, 2012), which makes it more challenging to accurately estimate water trends from ERA5 data than from COSMIC data.

The dominantly negative trend differences between $D_{Q,COSMIC}$ and $D_{Q,ERA5}$ at low-latitude regions at 850 hPa (see Figure 7f in Section 5.1), and the notable large trend difference between COSMIC and ERA5 over Sites #4-#7 are all concentrated within the northern and southern boundaries of the Intertropical Convergence Zone (ITCZ). The ITCZ encircles Earth near the thermal equator, where trade winds converge between the northeast (in the northern hemisphere) and the southeast (in the southern hemisphere). The specific position of ITCZ varies seasonally. The ITCZ has concentrated deep clouds spanning nearly the entire circumference of the equatorial regions, one of the most prominent atmospheric circulation features. Johnston et al. (2021) investigated the distribution and variability of COSMIC-2 water vapor by comparing it to collocated ERA5 and MERRA-2 reanalysis profiles in the tropical and subtropical regions. It was found by Johnston et al. (2021) that the largest moisture differences and weakest correlations were typically observed in regions that experience frequent convection, such as along the ITCZ, over the Indo-Pacific warm pool, or in central Africa. These locations match what we found in our study. Our

explanation for such difference is that for regions with frequent atmospheric circulation, such as deep clouds, the RO retrievals may characterize water vapor distribution and occurrence better than ERA5 due to the cloud-penetrating ability of GPS signal and higher height-resolution in RO data to resolve sharp moisture gradient better.

**Table 4: Water vapor trends over selected sites with notable COSMIC and ERA5 trend differences.**

| | Center (Lat., Long.) Region | At 500 hPa | | | At 850 hPa | | |
|---|---|---|---|---|---|---|---|
| | | $(\overline{Q_{COSMIC}}, \overline{Q_{ERA5}})$ (g/Kg) | $(D_{Q,COSMIC}, D_{Q,ERA5})$ (g/kg/Decade) | $(ND_{Q,COSMIC}, ND_{Q,ERA5})$ (%/Decade) | $(\overline{Q_{COSMIC}}, \overline{Q_{ERA5}})$ (g/Kg) | $(D_{Q,COSMIC}, D_{Q,ERA5})$ (g/kg/Decade) | $(ND_{Q,COSMIC}, ND_{Q,ERA5})$ (%/Decade) |
| Site #4 | (-5º, 135º) Arafura Sea | 3.24±0.63, 3.33±0.82 | -0.42±0.34, -0.25±0.33 | -13.09±10.44, -7.39±9.81 | 11.44±1.14, 11.88±0.99 | -0.27±0.37, 0.03±0.35 | -2.40±3.27, 0.26±2.98 |
| Site #5 | (5º, 35º) South Sudan | 2.41±0.83, 2.29±0.85 | 0.00±0.30, -0.01±0.29 | 0.13±12.33, -0.53±12.49 | 10.24±1.63, 10.71±1.72 | 0.32±0.45, 0.03±0.42 | 3.17±4.40, 0.24±3.93 |
| Site #6 | (-15º, 195 º) South Pacific Ocean | 1.98±0.78, 1.97±1.08 | -0.02±0.39, -0.01±0.39 | -0.77±19.94, -0.58±19.96 | 10.37±1.19, 10.96±1.27 | 0.04±0.44, 0.46±0.44 | 0.35±4.21, 4.18±4.01 |
| Site #7 | (-15º, 285 º) Peru | 1.24±0.55, 1.78±0.75 | -0.01±0.28, 0.03±0.27 | -1.07±22.55, 1.42±15.13 | 4.05±2.15, 6.11±1.81 | -0.34±0.42, -0.01±0.34 | -8.51±10.38, -0.17±5.63 |

## 6. Discussion and Conclusion

This paper evaluates the spatiotemporal consistency and difference between UCAR COSMIC (WETPrf) and ECMWF's ERA5 global reanalysis of water vapor data from 2007 to 2018. The analysis of temporal variability focuses on the long-term trends and seasonal variability of COSMIC and ERA5 water vapor data. Spatial variabilities of global, latitudinal, and regional distribution of COSMIC and ERA5 mean water vapor and trends at three pressure levels (300, 500, and 850 hPa) are analyzed and quantitatively compared. These two water vapor datasets generally show good agreements in spatiotemporal distributions and trends.

The key comparison results of time-averaged water vapor between COSMIC and ERA5 can be summarized as follows:

i) There have been coordinated efforts from Stratosphere–troposphere Processes And their Role in Climate (SPARC) Reanalysis Inter-comparison Project (S-RIP) to compare reanalysis datasets such as ERA5 and ERA-Interim using a variety of key diagnostics. The SPARC S-RIP confirmed the significant improvements of the latest version of reanalyses in ERA5 compared to ERA-interim (Fujiwara et al., 2017). Our study shows that COSMIC water vapor retrievals are more consistent with ERA5 reanalysis data than ERA-Interim. This suggests that although UCAR COSMIC 1DVAR retrieval used ERA-Interim as the background model (see Section 2.2), the impacts from ERA-Interim in the UCAR 1DVAR retrieval processing are minimum.

ii) At 300, 500, and 850 hPa, the differences between COSMIC water vapor retrievals and water vapor from ERA5 over the globe are 5.67±34.30%, -1.86±30.09%, and -2.30±21.21%, respectively. Ho et al. (2010) and Shao et al.

(2021b) showed systematic negative water vapor biases below 5 km for RO retrievals compared to radiosonde data. Such negative water vapor biases can be traced to the negative RO bending angle biases compared to the reanalysis model (Ho et al., 2020a). The negative water vapor biases below 5 km, *e.g.*, at 500 and 850 hPa, as studied here, are mainly due to the underestimation of water vapor in RO retrieval in the presence of atmospheric super-refraction or ducting in the moisture-rich low-troposphere (Sokolovskiy, 2003; Ao et al., 2003; Xie et al., 2006; Ao, 2007). Super-refraction occurs when the vertical atmospheric refractivity gradient exceeds a critical refraction threshold, *i.e.,* in the presence of a sharp change in refractivity. Such sharp change often exists around the planetary boundary layer, where sharp vertical gradients in moisture and temperature inversion are frequently observed. To address the negative moisture biases in RO retrieval and account for super-refraction or ducting, there are efforts to improve the 1DVAR retrieval algorithm by incorporating the reconstruction method introduced by Xie et al. (2010). Our study shows that the negative water vapor biases at 850 hPa are dominantly in the -40º to 40º (tropical and sub-tropical) moisture-rich regions. This study shows that the global (Fig. 4 and Table 1) water vapor trends are generally consistent with ERA5 at 500 and 850 hPa, although the negative water vapor biases are present at these two pressure levels.

iii) Latitude-dependence study shows the asymmetry in the latitudinal distribution of water vapor between the northern and southern hemispheres. There was a more rapid decrease of water vapor from the low-latitude tropical to the polar region in the southern hemisphere than in the northern hemisphere. The inter-hemispheric water vapor difference can be traced to the inter-hemispheric difference in temperature (Feulner et al., 2013).

The key findings from the trends estimates for the period from 2007 to 2018 COSMIC and ERA5 water vapor data at global, latitudinal, and regional (10 by the 10-degree grid) levels are summarized as follows:

i) The anomalous water vapor increase around 2015-2016 is identifiable in the COSMIC and ERA5 time series of water vapor data at all three pressure levels and was attributed to an El Niño event from April 2015 to May 2016.

ii) COSMIC and ERA5 global water vapor shows increasing trends at three pressure levels. The positive global water vapor trends from COSMIC data are 3.47±1.77, 3.25±1.25, and 2.03±0.65 %/Decade at 300, 500, and 850 hPa, respectively. The positive global water vapor trends can be the response to the global surface temperature increase (Held and Soden, 2006; Santer et al., 2006; Zhang et al., 2013; Chen and Liu, 2016; Ho et al., 2018; Allan et al., 2022).

iii) The latitude-mean water vapor trends are mostly positive (increasing) except in the southern -80º to -60º latitude zone and show substantial variability (between 0.4 to ~6 %/Decade) with latitude bins. The trend difference between COSMIC and ERA5 is less than 2 %/Decade for most latitude bins at three pressure levels.

iv) The regional distribution of water vapor trends in the tropical and subtropical regions has large local variabilities and is mixed with substantial increasing and decreasing trends. The regions in the equatorial Pacific Ocean with strong

increasing water vapor trends are identified. Negative (decreasing) water vapor trends, i.e., becoming drier, are observed near the Indo-Pacific Ocean region at 500 and 850 hPa.

    v) The assessment of regional water vapor trend variability and consistency between COSMIC and ERA5 indicates

a) A significant difference in the water vapor trends was estimated between COSMIC and ERA5 data at 850 hPa over two stratocumulus cloud-rich ocean sites. The possible cause of smaller trends from ERA5 water vapor data over stratocumulus cloud-rich regions could be the difficulty in accurately estimating water vapor at low altitudes in ERA5 reanalysis data (Lonitz and Geer 2017) compared with COSMIC RO measurements that are unaffected by stratocumulus cloud.

b) Over land, significantly increasing water vapor trends at 850 hPa can be observed around the region in the southern United States (Latitude: 35º, Longitude: 275º) and the region near south-eastern China (Latitude: 25º, Longitude: 115º). Two sites in southern Africa and Australia have long-term negative water vapor trends at 850 hPa, which can cause a regional long-term drier atmosphere and intensified droughts. The site in Australia has huge negative trends (< -10%/Decade at 850 hPa) (becoming drier) from both COSMIC and
ERA5 water vapor trends, which is consistent with Dai (2006) and Zhang et al. (2018).

        c) The differences between the water vapor trends of COSMIC and ERA5 are primarily negative in the tropical regions at 850 hPa. At 500 hPa, the negative differences are mainly distributed in the Indo-Pacific Ocean region. In contrast, the positive difference is located near the northern coast of the  Indian Ocean.

From our analysis, the regions with notable trend differences between COSMIC and ERA5 are mostly distributed within the northern and southern boundary of the ITCZ area, over the Indo-Pacific warm pool or central Africa. These regions experience frequent convection, such as deep convective clouds. Because of the cloud-penetration property of GNSS signal and higher height-resolution of RO retrieval, the height and temporal distribution of water vapor can be better characterized in RO retrievals than ERA5 in the presence of convection, such as deep clouds. The better representation of water vapor in RO data
may cause the difference in water vapor trend estimation between COSMIC and ERA5 over these regions, which will need further studies with other long-term water vapor data. In particular, comparing long-term ground-based GNSS and GPS data (Mears et al., 2017) and radiosonde data (Patel and Kuttippurath, 2022) can help address the biases and trend differences between RO and the reanalysis model over land.

In analyzing long-term water vapor trends from RO data, it is important to remove sampling errors to correct the biases due to RO data's limited time and location coverage. The sampling error removal accounts for the difference between the orbital-specific distribution of COSMIC RO measurements and uniformly distributed global ERA5 data. After applying sampling error removal, our estimations indicate a reduction in uncertainty by approximately 4.8 times at 500 hPa and 3.1 times at 850 hPa. This magnitude of uncertainty reduction is close to that shown by Gleisner et al. (2020). Our study also shows that the
COSMIC water vapor retrievals are more consistent with ERA5 than ERA-Interim model data and confirms that ERA5 has

significantly improved quality than ERA-Interim. This paper's overall global water vapor trends are close to the trend results from Allan et al. (2022). We postulate that using other global reanalysis models, such as NCEP and MERRA-2, may have compatible global trends but differ in regional trends from our results, which will need further evaluation.

This paper compares twelve years of COSMIC data from 2007-2018 with ERA5 reanalysis data. As the follow-on mission of COSMIC, the COSMIC-2 constellation with six satellites has produced RO data since 2019 (Ho et al., 2020b; Ho et al., 2022). In addition, commercial RO sensors such as Spire and GeoOptics (Chen et al., 2021) and the upcoming RO sensors onboard MetOp Second Generation and other RO missions continue to augment RO data's temporal and spatial coverage. These growing RO datasets combined with the historical multiple RO mission data will provide the opportunity to establish consistent

long-term CDR-grade global temperature, water vapor, and derived climatology data products. It is important to emphasize that consistently processed temperature and water vapor data with the same excess phase to bending angle and 1DVAR retrieval models is critical to establish such long-term CDR-grade datasets from multiple RO mission data.

### Appendix

### A.1. Seasonal variability of COSMIC and ERA5 water vapor distribution

To understand the seasonal variability of water vapor at different pressure levels, we show the annual variation of mean water vapor over 12 months in eight latitudinal bins (20-degree bins from -80 to 80 degrees in latitude) in Fig. A.1a, c, e and A.2a, c, e at 300, 500, and 850 hPa pressure levels for the southern and northern hemisphere, respectively. Each month's 12-year (2007 to 2018) water vapor data in each latitude bin have been averaged for COSMIC and ERA5. Figures A.1a, c, e, and A.2a, c, e show that the water vapor is high (wet) in the summer and low (dry) in the winter for the corresponding hemisphere at all

three pressure levels. The latitudinal and seasonal variability of water vapor differences between COSMIC and ERA5 are further quantified as the relative difference ($\Delta Q_{COSMIC-ERA5}(\%)$) in Fig. A.1b, d, f and A.2b, d, f for the southern and northern hemisphere, respectively. Figures A.1a,c,e, and A.2a,c,e show the overall agreement in seasonal variability between COSMIC and ERA5 at three pressure levels over the northern and southern hemispheres. We can use COSMIC data as a reference to evaluate the overall seasonal variability in different latitude zones. We extracted the summer maximum ($Q_{max,COSMIC}$) and

winter minimum ($Q_{min,COSMIC}$) monthly mean COSMIC water vapor from Fig. A.1 and A.2.

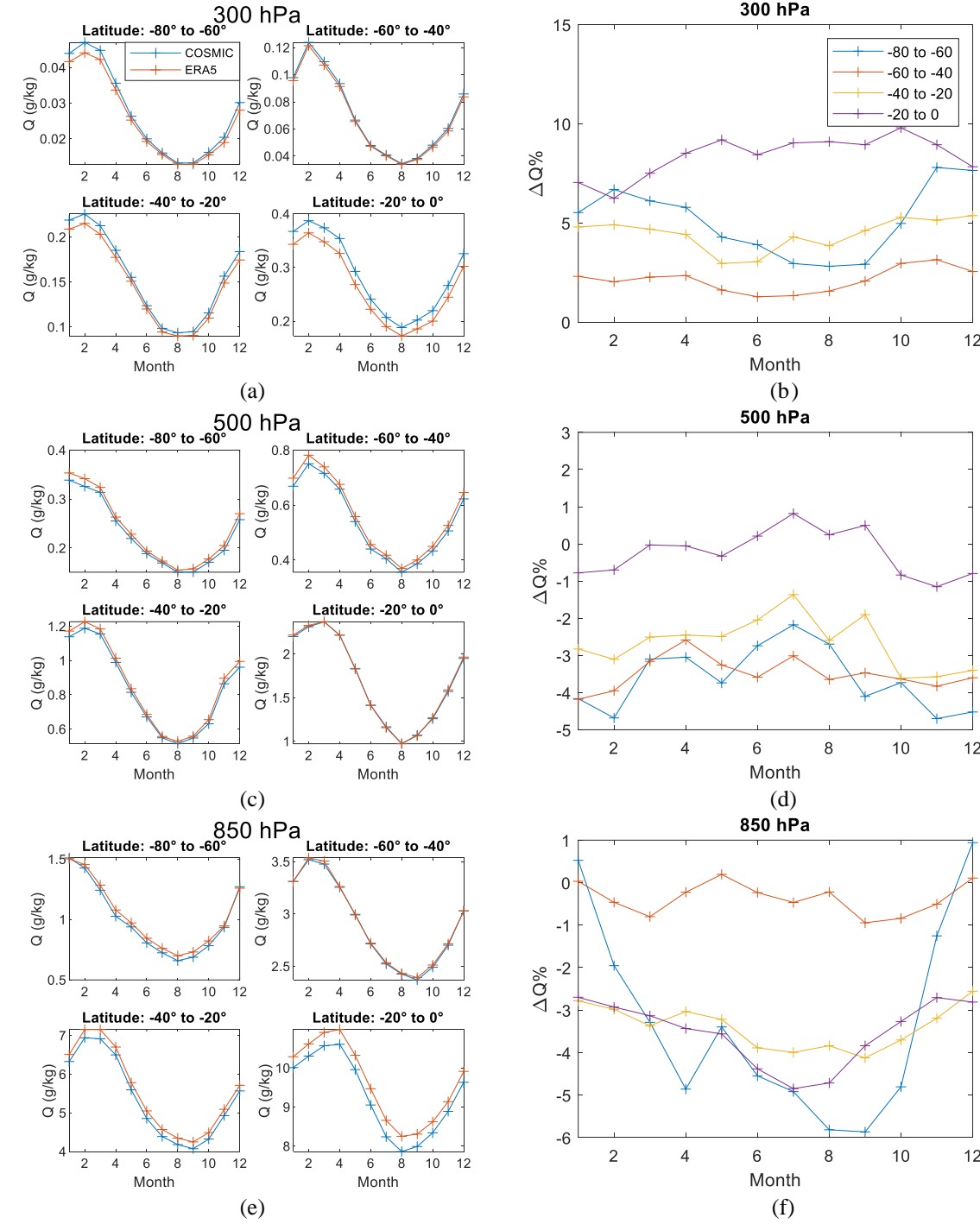

**Figure A.1: (a, c, e) Comparison of seasonal variability (over 12 months) between COSMIC and ERA5 water vapor data in 4 southern hemisphere latitude bins at 300, 500, and 850 hPa, respectively. (b, d, f) Seasonal variation of the percent difference between COSMIC and ERA5 water vapor data grouped in 4 southern hemisphere latitude bins at 300, 500, and 850 hPa, respectively.**

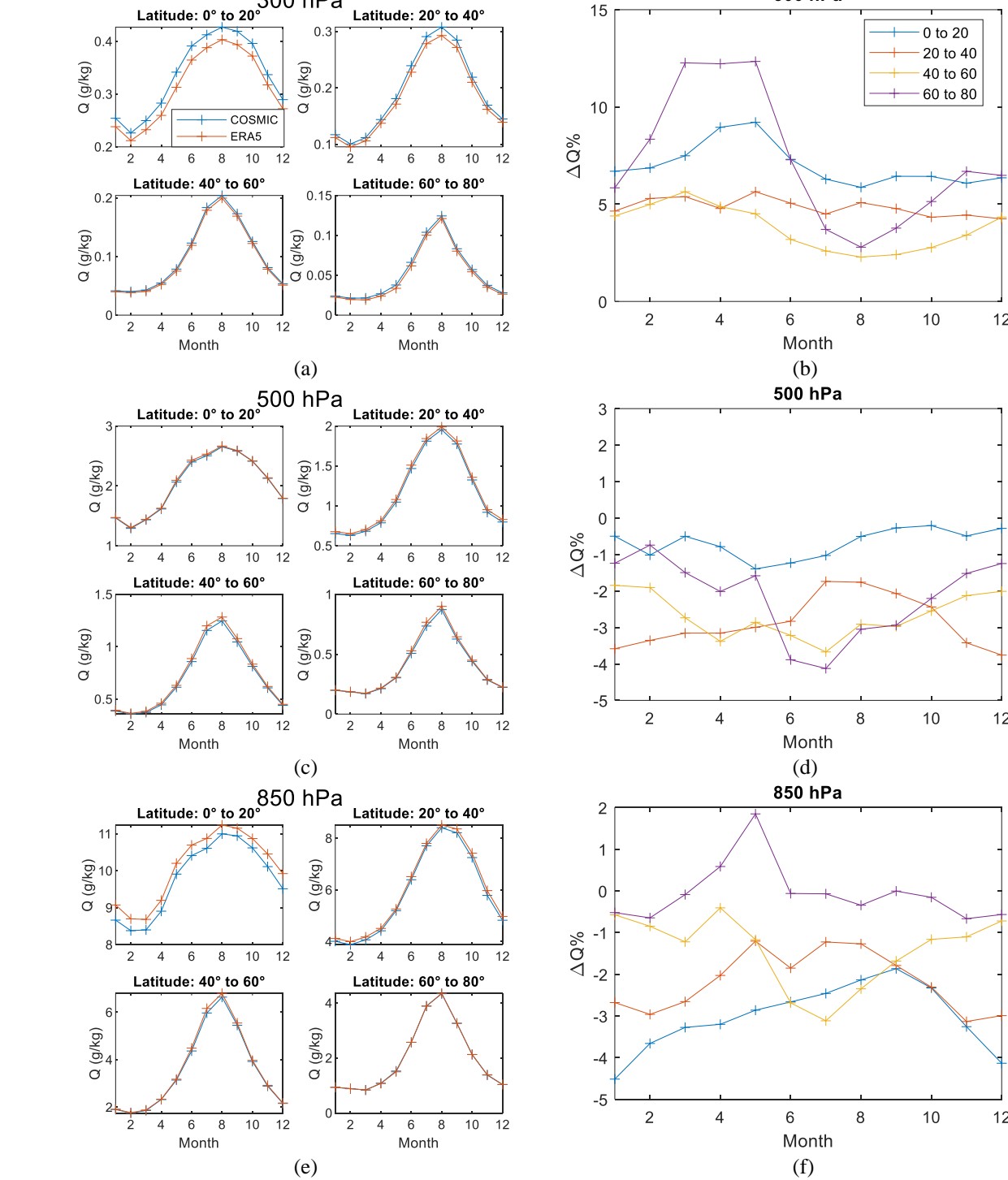

(a)                  (b)

(c)                (d)

(e)                  (f)

**Figure A.2 Same as Fig. A.1, but over the northern hemisphere.**

In Fig. A.3, we show the summer maximum and winter minimum monthly mean COSMIC water vapor and the annual water vapor variation magnitude defined as $\Delta Q_{max-min,COSMIC} = Q_{max,COSMIC} - Q_{min,COSMIC}$ at three pressure levels. Over all three pressure levels, the two low latitude bins (-20º to 0º and 0º to 20º) both have comparable $Q_{max,COSMIC}$, $Q_{min,COSMIC}$, and $\Delta Q_{max-min,COSMIC}$, which suggest that the mixture of water vapor in these two southern and northern latitude zones is quite efficient at all three pressure levels. As approaching higher latitudes in bins with |latitude| > 20º, the southern hemisphere atmosphere is generally drier than the matching latitude zones in the northern hemisphere at all three pressure levels. Figure A.3 also shows that the seasonal water vapor variabilities, i.e., $\Delta Q_{max-min,COSMIC}$, are more significant in the northern hemisphere than in the southern hemisphere for latitude zones above 20 degrees at all three pressure levels.

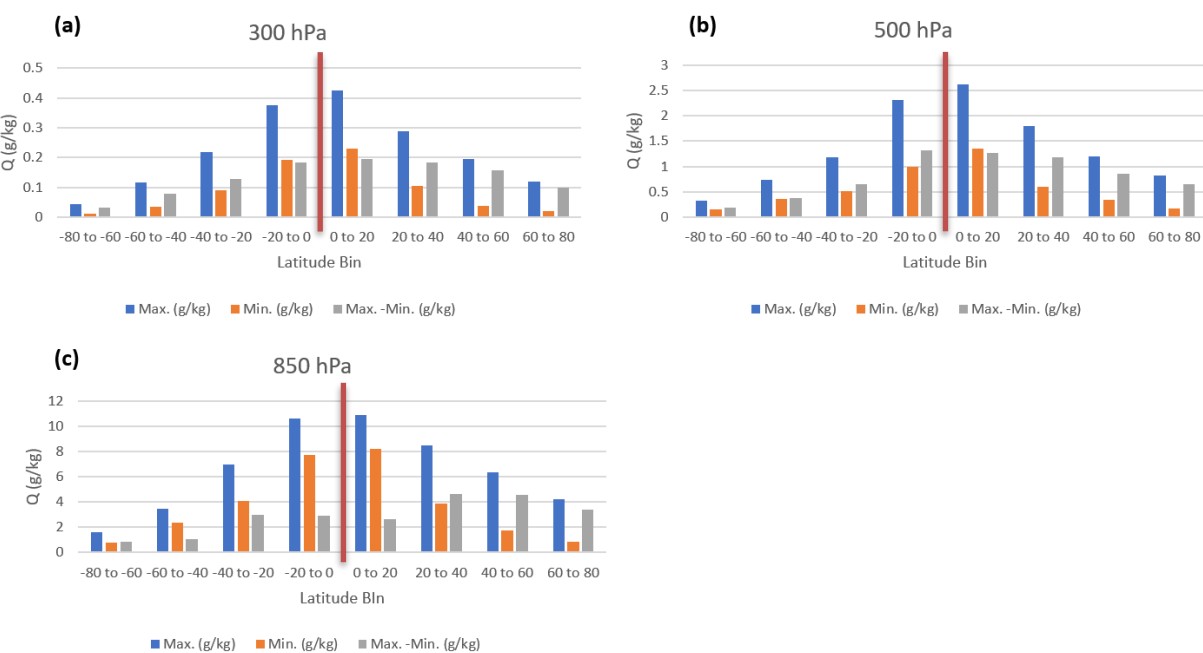

**Figure A.3: Maximum, minimum, and maximum-minimum annual monthly water vapor statistics at three pressure levels (a) 300, (b) 500, and (c) 850 hPa from COSMIC retrievals. The vertical red line in each panel separates the latitude bins in the southern (to its left) and northern (to its right) hemispheres.**

Next, we quantify the difference in the seasonal variability between the COSMIC and ERA5 water vapor data. As shown in Fig. A.1b and A.2b, at 300 hPa, $\Delta Q_{COSMIC-ERA5}(\%)$ are all positive, i.e., $Q_{COSMIC} > Q_{ERA5}$, with values ranging from 1% to 12.5% over twelve months and both hemispheres. The most substantial seasonal variability in the peak-to-valley value of annual $\Delta Q_{COSMIC-ERA5}(\%)$ occurs in the northern 60º to 80º latitude bin with seasonal variation around ~8% from March to August. The high (60º to 80º) and low (0º to 20º) latitude bins in both northern and southern hemispheres all have a peak-to-valley value of annual $\Delta Q_{COSMIC-ERA5}(\%)$ higher than 4%. For middle (20º to 60º) latitude bins in northern and southern

hemispheres, the magnitude of seasonal variation of $\Delta Q_{COSMIC-ERA5}(\%)$ is less than 2%. The latitudinal variability of $\Delta Q_{COSMIC-ERA5}(\%)$ agrees with the mean latitudinal values shown in Fig. 3c.

At 500 hPa, Fig. A.1d and A.2d show that $\Delta Q_{COSMIC-ERA5}(\%)$ are negative over twelve months for all latitude bins except the -20° to 0° latitude bin which has $\Delta Q_{COSMIC-ERA5}(\%)$ varying from -1% to 1%. The overall peak-to-valley seasonal variabilities of $\Delta Q_{COSMIC-ERA5}(\%)$ are in the range of 1% to 3%, with the most significant seasonal variability (~3%) in the 60° to 80° high latitude bin. Such magnitudes of seasonal variability of $\Delta Q_{COSMIC-ERA5}(\%)$ at 500 hPa are much smaller than those at 300 hPa, which suggests that using $\Delta Q_{COSMIC-ERA5}(\%)$ as the metrics, the water vapor of COSMIC retrieval is more consistent with ERA5 at 500 hPa than at 300 hPa. The latitudinal variability of $\Delta Q_{COSMIC-ERA5}(\%)$ at 500 hPa is consistent with the mean latitudinal values shown in Fig. 3f.

At 850 hPa, Fig. A.1f and A.2f show that $\Delta Q_{COSMIC-ERA5}(\%)$ are dominantly negative over twelve months for all latitude bins except one bin in latitude 60° to 80° which has $\Delta Q_{COSMIC-ERA5}(\%)$ varying from -0.7% to 1.2%. The seasonal variabilities (peak to valley variation of annual $\Delta Q_{COSMIC-ERA5}(\%)$) are weak ($< 2.5\%$) for all of the latitude bins except the southern high latitude bin in -80° to -60° which has the most significant seasonal variability ~ 6%. The latitudinal variability of $\Delta Q_{COSMIC-ERA5}(\%)$ at 850 hPa agrees with Fig. 3i.

## A.2. Method of removing the COSMIC sampling errors for water vapor time series analysis

The steps of calculating the COSMIC sampling error and reconstructing the water vapor time series for trend analysis are detailed below.

1) For a RoI such as the global, latitudinal bins, or a 10°× 10° latitude/longitude grid, the collocated water vapor data from COSMIC and ERA5 in that region are accumulated for each month. For COSMIC WETPrf data, the location of the RO profile is used to determine whether the RO data is in the RoI. For a given pressure layer, interpolation over the RO profile pressure levels was carried out for COSMIC water vapor data to derive the water vapor at the specific pressure. The ERA5 data are distributed globally on 0.2-degree latitude/longitude grids, 37 pressure layers, and 6-hour intervals. Therefore, we interpolate ERA5 data over latitude/longitude and time at the given pressure level that matches the COSMIC RO observation. With the accumulated monthly COSMIC or ERA5 water vapor data for a given RoI, the monthly mean values at a given pressure level are calculated to form the long-term time series of monthly mean water vapor ($\overline{Q_{COSMIC\_Sample}}$) for the RoI. Figure A.4a shows an example of the long-term time series of COSMIC ($\overline{Q_{COSMIC\_Sample}}$) and ERA5 ($\overline{Q_{ERA5\_Sample}}$) water vapor variation at 850 hPa pressure level for the 0°-20° latitude bin RoI in the northern hemisphere.

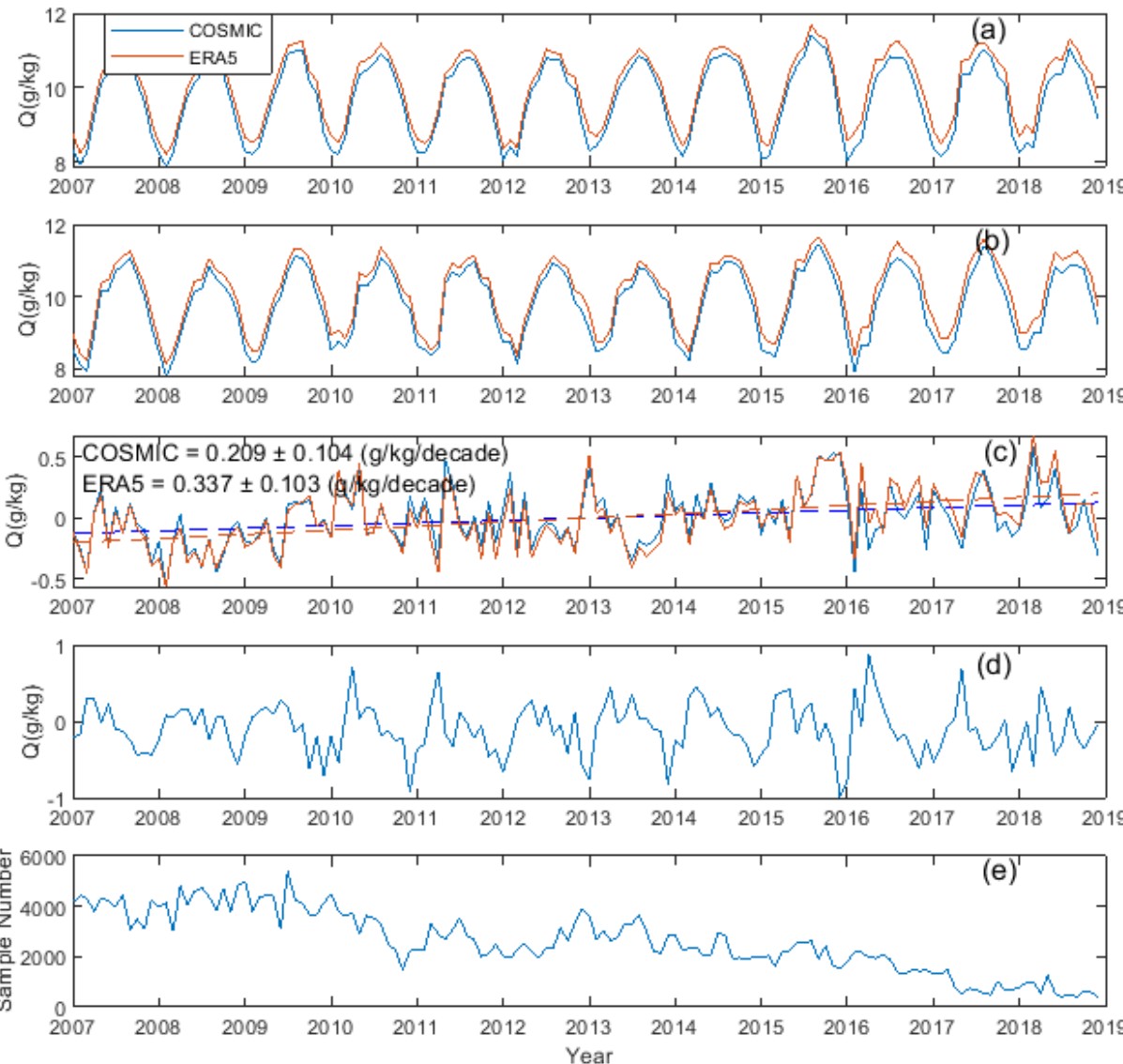

**Figure A.4 Steps to derive the long-term water vapor trend for a given RoI at pressure level = 850 hPa. (a) The time series of the monthly mean of collocated COSMIC and ERA5 water vapor data in the 0°-20° latitude bin over the northern hemisphere. (b) Time series of COSMIC and ERA5 water vapor data after sampling error removal. (c) The deseasonalized monthly-mean COSMIC and ERA5 water vapor data time series over the 0°-20° latitude bin. Dashed lines are the trends derived from linear regression. In (a-c), red and blue lines are time series of ERA5 and COSMIC water vapor data or trends, respectively. (d) Time series of COSMIC water vapor sampling error $Q_{SE}$ calculated with Eq. (A.1). (e) The sample number of COSMIC observations time series fall into the 0° -20° latitude bin.**

2)    Fig. A.4e shows the monthly sample number of COSMIC RO data that fall into the 0˚-20˚ latitude bin RoI and has substantial variations over the lifetime of COSMIC when the number of available RO sensors in the COSMIC constellation varies over time. Particularly, there was a continuous decrease in the sample number after the middle of 2013. There are six small satellites (C1E1 to C1E6) in the COSMIC-1 constellation. The service interval and performance of these six satellites vary over time. C1E3 is the first satellite that stopped producing data in mid-2010.
C1E2, C1E3, and C1 E4 ended their operations over the time interval from 2015 to 2017. C1E1 and C1E6 continued in operation until the middle of 2019 and early 2020, respectively. Due to the varying performances and availabilities of C1E1 to C1E6, the time series of the combined valid profile numbers from these six satellites thus show the pattern shown in Fig. A.4e.

To account for the impacts of the limited and varying sample number on the trend analysis, we need to apply sampling error removal to COSMIC data. The sampling errors are the difference between the sample-mean and cell-mean, which can be estimated using monthly ERA5 data from 2007 to 2018. Eq. (A.1) illustrates the calculation of the sampling error ($Q_{SE}$).

$$Q_{SE} = \overline{Q_{ERA5\_Sample}} - \overline{Q_{ERA5\_RoI}}, \qquad\qquad (A.1)$$

where $\overline{Q_{ERA5\_Sample}}$ is the monthly mean of the interpolated water vapor profiles from ERA5 that match the COSMIC RO observations in the RoI at a given pressure level; $\overline{Q_{ERA5\_RoI}}$ is the monthly spatial and temporal mean of the ERA5 water vapor in the RoI at the same pressure level. The sampling error removal is carried out by subtracting monthly
$Q_{SE}$ from the COSMIC monthly water vapor data using Eq. (A.2)

$$Q_{COSMIC\_SER} = \overline{Q_{COSMIC\_Sample}} - Q_{SE}, \qquad\qquad (A.2)$$

where $Q_{COSMIC\_SER}$ is the COSMIC water vapor data after sampling error removal. For ERA5 data, the application of
sampling error $Q_{SE}$ removal to $\overline{Q_{ERA5\_sample}}$ essentially recovers $\overline{Q_{ERA5\_RoI}}$. The time series of $Q_{COSMIC\_SER}$ are unaffected by the limited and varying sample number of COSMIC RO observations. They are used to construct monthly-mean climatology (MMC) water vapor data records and characterize the long-term trend of water vapor variation for a given RoI. Fig. A.4b compares the time series of $Q_{COSMIC\_SER}$ and $\overline{Q_{ERA5\_RoI}}$ for the 0˚-20˚ latitude bin RoI at 850 hPa.

Fig. A.4d shows the time series of COSMIC sampling error $Q_{SE}$ in the 0˚-20˚ latitude bin. Similar to the COSMIC sampling error data shown in Gleisner et al. (2020) and Shen et al. (2021), there are seasonal oscillations (around 0

g/kg) in the time series of water vapor sampling error shown in Fig. A.4d, which is mainly due to the difference between the orbital-specific distribution of COSMIC RO observations (Ho et al., 2020a) and uniformly-distributed global ERA5 data. The non-uniform local time and latitude distribution of COSMIC-1 profiles coupled with the annual variation of the Sun's declination contribute to the seasonal oscillation in the sampling error time series. As the monthly sample number of COSMIC RO data decreases after 2010 (Fig. A.4e), $Q_{SE}$ appears to have increased amplitudes. Over the interval after the middle of 2017, when the sample number of COSMIC decreases more significantly, $Q_{SE}$ appears to have more rapid oscillations.

3) As shown in Fig. A.4b, there are substantial seasonal oscillations in the monthly-mean water vapor data time series after the sampling error removal. To calculate the long-term trend from the time series data, the monthly mean water vapor data must be deseasonalized to filter out the annual oscillation. This step is carried out by grouping the monthly-mean water vapor data of the same month over the 2007-2018 period and calculating the mean as climate monthly mean. In this way, we have twelve monthly climate water vapor means that can characterize the annual water vapor variation. The long-term water vapor time series is then deseasonalized by subtracting the corresponding climate monthly mean at each data point. Figure A.4c shows an example of the time series of the deseasonalized water vapor for COSMIC and ERA5 at 850 hPa pressure level in the 0˚ -20˚ latitude bin RoI.

4) Linear regression has been carried out with the deseasonalized time series of water vapor to calculate the slope, i.e., the trend $D_Q$ (g/kg/decade), of the water vapor variation. The example in Fig. A.4c shows the linear fitting curves as dashed red and blue lines for ERA5 and COSMIC data, respectively. The values and 95% confidence interval of the ERA5 and COSMIC water vapor trends are also listed in the figure.

## A.3. Effects of sampling error removal on the uncertainty of the regional water vapor trend analysis

Figure A.5a and b show the global ($10^o \times 10^o$) distribution of trends derived from the sampling error $Q_{SE}$ time series at 500 and 850 hPa, respectively. The grids with > 1.5% missing monthly data over the 2007 to 2018 interval are marked as white blanks in Fig. A.5a and A.5b. It can be seen that the sampling error removal does introduce corrections to the regional trends of COSMIC water vapor data. To further evaluate the impacts of the sampling error removal on the uncertainty of the water vapor trend analysis using long-term COSMIC water vapor data, we calculated the histogram distribution of the relative water vapor trend difference between the COSMIC and ERA5 data, i.e., $\Delta ND_Q = ND_{Q,COSMIC} - ND_{Q,ERA5}$, from the global ($10^o \times 10^o$) distribution. In particular, COSMIC water vapor data without and with sampling error removal are used to calculate $\Delta ND_{Q,without\ SER}$ and $\Delta ND_{Q,with\ SER}$, respectively. Fig. A.6a and b show the histogram distribution and Gaussian-fit of $\Delta ND_{Q,without\ SER}$ and $\Delta ND_{Q,with\ SER}$ at 500 and 850 hPa, respectively. Gleisner et al. (2020) showed that removing sampling error could help reduce the uncertainty to about 1/3 in analyzing multiple RO data products processed by RO Meteorology Satellite Application Facility (ROM SAF). From our analysis, the Full-Width-Half-Maximum (FWHM) histogram distribution in Fig. A.6a and b has been reduced from 28.1%/Decade and 25.6%/Decade to 5.8%/Decade and 8.2%/Decade at 500 and 850

940 hPa, respectively, after applying the sampling error removal to COSMIC data. This is about a 4.8 and 3.1-time reduction in uncertainty at 500 and 850 hPa, respectively, which is quite close to the ~3 times of uncertainty reduction shown in Gleisner et al. (2020). We note that the ERA5 trend is used as the reference in the uncertainty analysis. On the other hand, the remaining differences between $ND_{Q,COSMIC}$ with sampling error removal and $ND_{Q,ERA5}$ can be partly due to better cloud-penetration characteristics of COSMIC RO observations over regions with frequent clouds. Therefore, our analysis of the impacts of

945 sampling error removal on trend uncertainty provides an upper-bound estimation.

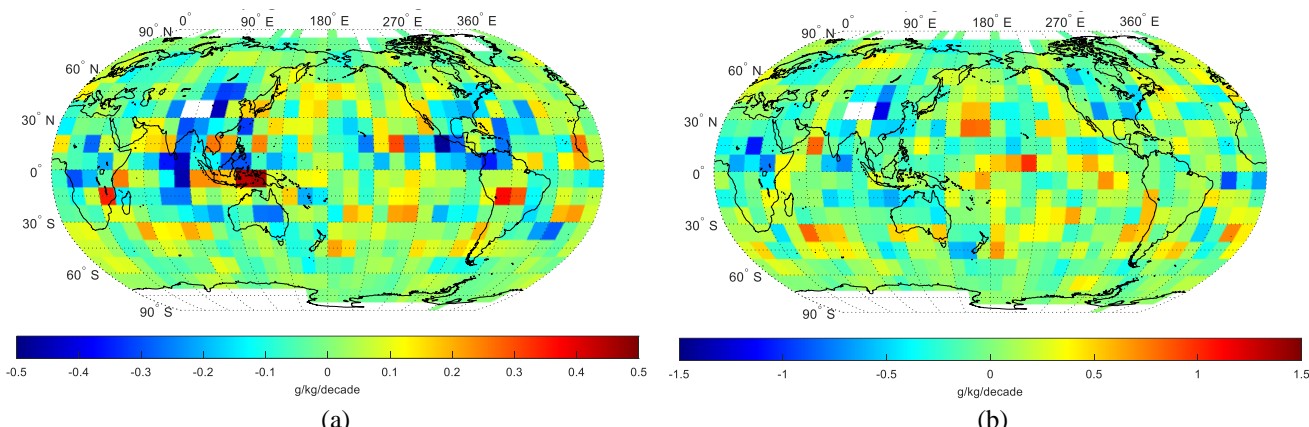

(a)                                                                                      (b)

**Figure A.5: (a) and (b) The distribution of trends of sampling error $Q_{SE}$ time series at 500 and 850 hPa, respectively. The white blanks in (a) and (b) are grids with > 1.5% missing monthly data over the 2007 to 2018 interval.**

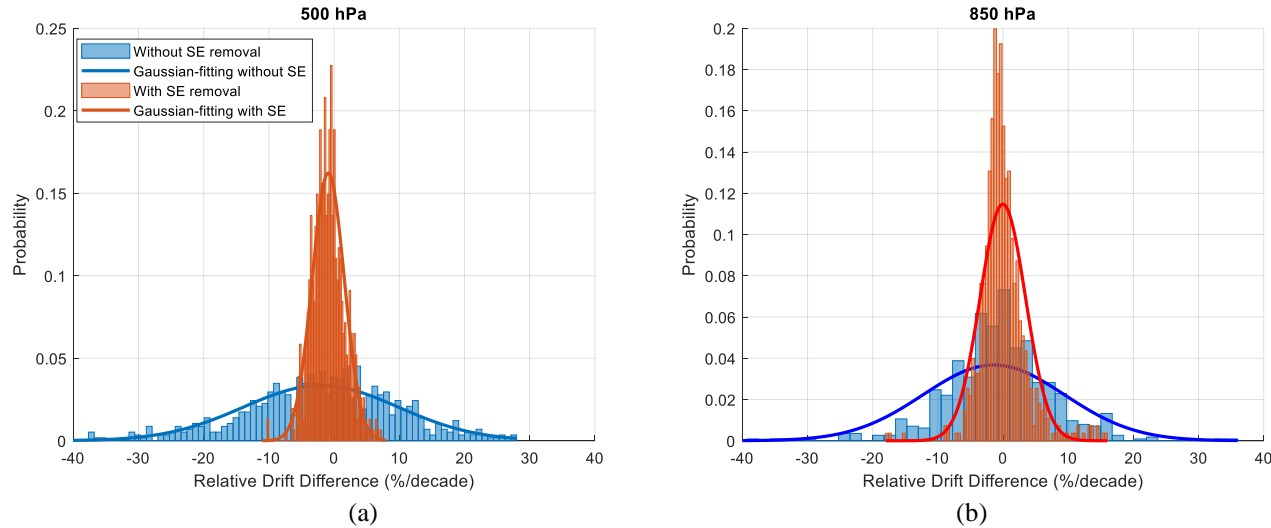

(a)                                                                                      (b)

**Figure A.6: (a, b) The histogram of relative water vapor trend difference (%/decade) between COSMIC and ERA5 water vapor at**
955 **500 and 850 hPa, respectively. In both panels, blue and orange bar charts are the distribution of the COSMIC water vapor trend difference relative to ERA5 before and after the sampling error removal was applied, respectively. The blue and red lines are the Gaussian-fitted distribution of the relative water vapor trend difference for the FWHM calculation.**

**A.4 Comparison of COSMIC and ERA5 over sites with notable increasing and decreasing water vapor trends**

Although the global and latitudinal water vapor trends presented in Section 4 exhibit an overall upward trend, Figures 7a-d in Section 5.1 highlight that within tropical and subtropical regions, the regional distribution of water vapor trends displays significant local variations with a blend of pronounced increases and decreases in trends. Such variations in regional water vapor trends in general agree with the past studies, e.g., Ross and Elliott, 2001; Dai, 2006; Mieruch et al., 2008, 2014; Zhang et al., 2018. In this section, we quantitatively evaluate the regional variability of water vapor trends by selecting a few sites with notable increasing (Sites #8-#12 in Fig. 8) and decreasing (Sites #13-#17 in Fig. 8) water vapor trends and compare with past studies. To identify these sites, we first searched the $10^{\circ}\times10^{\circ}$ global grids and identified the regions with the largest increasing and decreasing water vapor trends. Within these regions, we selected one representative $10^{\circ}\times10^{\circ}$ grid in each region as the site of interest, and the water vapor trends of these sites estimated from COSMIC and ERA5 data are listed and compared in Table A.1 and A.2.

Both COSMIC and ERA5 trend data show increasing water vapor trends at 500 hPa and 850 hPa for the five selected sites (Table A.1). Site #8, #9, and #12 are located in the ocean, and sites #10 and #11 are located on the land. The sites #8, #9, and #11 have high mean water vapor (> 7.5 g/kg at 850 hPa and > ~1.5 g/kg at 500 hPa). At 850 hPa, the mean water vapor from COSMIC is lower than ERA5 for all five sites in Table A.1. The trends between COSMIC and ERA5 are consistent with $|ND_{Q,COSMIC} - ND_{Q,ERA5}| < 2.7\,\%/\text{Decade}$ at 850 hPa for these five sites. Site #10 (latitude: $30^{\circ}$ to $40^{\circ}$; longitude: $270^{\circ}$ to $280^{\circ}$) over land in the United States has the strongest increasing water vapor trend: > 18 %/Decade at 850 hPa and > 39 %/Decade at 500 hPa among all of the $10^{\circ}$ by $10^{\circ}$ grids over land. Site #10 in the United States and Site #11 in Southeast China are representative land sites becoming moister. Among the sites situated over the ocean, Sites #8 and #12 stand out with substantial increasing water vapor trends (> 17 %/Decade and > 23 %/Decade at 850 hPa, respectively). Many previous studies have explored the trends in surface temperature (e.g., Gu and Adler, 2022 and references therein). Global surface keeps warming up, though with rich spatial structures of temperature change. Higher surface temperatures are closely linked to higher levels of water vapor in the atmosphere through the relationship governed by the Clausius-Clapeyron equation. The saturation vapor pressure of water vapor increases with temperature. The close relations between higher temperature and higher water vapor have been shown in observations and models (Wentz and Schabel, 2000; Trenberth et al., 2005; Held and Soden, 2006; Allan et al., 2014). From the study by Gu and Adler, 2022, ocean surface warming can readily be seen in the Indian and tropical Pacific oceans, roughly corresponding to the strong increasing tropospheric water vapor trends for Sites #8, #9, and #12 we observed.

Table A.2 lists the water vapor trends for five sites with notable decreasing trends. Sites #13 and #14 are located over the ocean, and Sites #15, #16, and #17 are located over land. For the two ocean sites, water vapor trends at 500 and 850 hPa from COSMIC and ERA5 are strongly negative (mostly < -10 %/Decade). These two ocean sites accompany the regions with strong

positive water vapor trends over the equatorial Pacific Ocean and the Laccadive Sea, respectively (Fig. 7). The long-term negative water vapor trend at 850 hPa for Site #15 in southern Africa can cause a regional drier atmosphere. Site #16 in Brazil has a mild decreasing water vapor trend at 850 hPa and a strong decreasing water vapor trend (< -10%/Decade) at 500 hPa from COSMIC data. Site #17 in Australia has the lowest mean water vapor, i.e., driest, among the five sites and a strong decreasing trend (< -10%/Decade at 850 hPa), which can result in a long-term drier atmosphere in this region (Dai et al., 2006; Zhang et al., 2018). Sites #17 is a representative dry region over land which becomes drier at 850 hPa.

**Table A.1: Water vapor trends over selected sites with notable increasing trends.**

| | | At 500 hPa | | | At 850 hPa | | |
|---|---|---|---|---|---|---|---|
| | Center (Lat., Long.) Region | $(\overline{Q_{COSMIC}}, \overline{Q_{ERA5}})$ (g/Kg) | $(D_{Q,COSMIC}, D_{Q,ERA5})$ (g/kg/Decade) | $(ND_{Q,COSMIC}, ND_{Q,ERA5})$ (%/Decade) | $(\overline{Q_{COSMIC}}, \overline{Q_{ERA5}})$ (g/Kg) | $(D_{Q,COSMIC}, D_{Q,ERA5})$ (g/kg/Decade) | $(ND_{Q,COSMIC}, ND_{Q,ERA5})$ (%/Decade) |
| Site #8 | (15°, 235°) West of Baja coast | 1.48±0.53, 1.53±0.81 | 0.44±0.34, 0.39±0.34 | 29.56±22.99, 25.46±22.08 | 7.68±1.87, 8.49±2.47 | 1.36±0.79, 1.51±0.78 | 17.71±10.22, 17.73±9.17 |
| Site #9 | (5°, 85°) Laccadive Sea | 2.86±0.66, 2.83±1.00 | 0.58±0.39, 0.48±0.40 | 20.28±13.79, 16.99±13.97 | 10.96±1.13, 11.11±1.29 | 1.08±0.51, 1.06±0.50 | 9.83±4.67, 9.55±4.51 |
| Site #10 | (35°, 275°) United States | 1.04±0.46, 1.10±0.61 | 0.41±0.24, 0.47±0.23 | 39.48±22.71, 42.67±21.34 | 5.93±2.85, 6.61±3.17 | 1.18±0.82, 1.22±0.81 | 19.91±13.78, 18.40±12.20 |
| Site #11 | (25°, 115°) Southeast China | 2.00±1.13, 1.96±1.27 | 0.13±0.30, 0.11±0.29 | 6.44±14.80, 5.65±14.99 | 9.31±3.15, 9.34±3.57 | 0.70±0.85, 0.86±0.85 | 7.52±9.18, 9.21±9.08 |
| Site #12 | (-45°, 165°) Near New Zealand | 0.62±0.20, 0.70±0.33 | 0.10±0.13, 0.10±0.13 | 15.60±21.43, 14.13±18.75 | 3.67±0.64, 3.98±1.34 | 0.95±0.54, 0.92±0.54 | 25.77±14.73, 23.10±13.56 |

**Table A.2: Water vapor trends over selected sites with notable decreasing trends.**

| | | At 500 hPa | | | At 850 hPa | | |
|---|---|---|---|---|---|---|---|
| | Center (Lat., Long.) Region | $(\overline{Q_{COSMIC}}, \overline{Q_{ERA5}})$ (g/Kg) | $(D_{Q,COSMIC}, D_{Q,ERA5})$ (g/kg/Decade) | $(ND_{Q,COSMIC}, ND_{Q,ERA5})$ (%/Decade) | $(\overline{Q_{COSMIC}}, \overline{Q_{ERA5}})$ (g/Kg) | $(D_{Q,COSMIC}, D_{Q,ERA5})$ (g/kg/Decade) | $(ND_{Q,COSMIC}, ND_{Q,ERA5})$ (%/Decade) |
| Site #13 | (25°, 175°) North Pacific Ocean | 1.13±0.46, 1.27±0.73 | -0.15±0.29, -0.10±0.29 | -13.36±25.53, -7.55±22.76 | 7.85±1.71, 8.01±1.97 | -1.09±0.64, -0.85±0.64 | -13.93±8.20, -10.62±7.98 |
| Site #14 | (15°, 65°) Arabian Sea | 1.30±0.92, 1.39±1.17 | -0.16±0.36, -0.26±0.36 | -12.59±27.49, -18.40±25.95 | 7.23±2.65, 7.35±3.15 | -0.91±0.82, -0.74±0.79 | -12.55±11.30, -10.05±10.78 |
| Site #15 | (-25°, 25°) Ngwaketse, Botswana | 1.25±0.86, 1.11±0.95 | 0.01±0.29, 0.04±0.29 | 0.59±23.40, 3.92±26.29 | 6.72±2.67, 6.80±2.95 | -0.43±0.70, -0.34±0.72 | -6.33±10.40, -5.06±10.62 |
| Site #16 | (-15°, 315°) Brazil | 1.60±0.94, 1.59±1.24 | -0.17±0.42, -0.09±0.42 | -10.42±26.10, -5.66±26.49 | 9.88±1.78, 10.28±1.95 | -0.29±0.47, -0.12±0.47 | -2.96±4.79, -1.13±4.59 |
| Site #17 | (-35°, 145°) Australia | 0.65±0.28, 0.73±0.48 | 0.17±0.20, 0.18±0.19 | 25.79±30.55, 23.97±26.53 | 4.27±1.05, 4.58±1.57 | -0.56±0.57, -0.49±0.58 | -13.09±13.41, -10.74±12.69 |

Data availability. The ECMWF ReAnalysis Model 5 (ERA5) data are publicly available at
1005 https://www.ecmwf.int/en/forecasts/dataset/ecmwf-reanalysis-v5. The UCAR COSMIC water vapor data are available at
https://cdaac-www.cosmic.ucar.edu/cdaac/products.html.

Author contribution. Conceptualization by SH and XS. XS, SH, XZ, and YC defined the validation methodology. XJ, TL, XS,
and BZ wrote the scripts used for the analysis. XJ, TL, XS, BZ, and JD performed the data analysis and validation. XZ, BZ,
and JD provided the satellite data. XS and SH wrote the manuscript. XS, SH, XZ, YC, TL, BZ, and JD reviewed and edited
the manuscript. Project administration by XS and BZ. Funding acquisition by SH and YC. All authors have read and agreed
to the published version of the manuscript.

Competing interests. The authors declare that they have no conflict of interest.

Acknowledgments. The author would like to thank Guojun Gu, Yun Zhou and Loknath Adhikari for their input during the
process of this manuscript. The manuscript contents are solely the opinions of the authors and do not constitute a statement of
policy, decision, or position on behalf of NOAA or the U.S. government.

Financial support. This study was supported by NOAA grant NA19NES4320002 (Cooperative Institute for Satellite Earth
System Studies-CISESS) at the University of Maryland/ESSIC.

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
