# Peer review of "Characterizing the tropospheric water vapor spatial variation and trend using 2007-2018 COSMIC radio occultation and ECMWF reanalysis data"

_Atmospheric Chemistry and Physics, 2022_

## Author Comment (AC1)

We thank the reviewers for the helpful comments and suggestions. We have made revision to the manuscript and believe that we have adequately addressed the reviewer's comments. In the following, we summarize our reply to the reviewers.

**Reviewers' comments:**

In this study COSMIC radio occultation and ECMWF reanalyses data is used to estimate trends in tropospheric water vapour. The trend analyses is performed globally for specific latitude bands, separating the globe into latitude bins of 20° and on three pressure levels, namely 300, 500 and 850 hPa. Before the trends are estimated the data sets are inter-compared.

**General comments:**

The study itself is worth to be published, however needs major revisions before publication in ACP. Since in this study first a detailed inter-comparison is performed followed by a quite detailed trend analyses, the paper becomes very long and hard to follow. The current version of the manuscript gives the impression that actually two manuscripts have been combined. I would suggest to significantly shorten the paper, especially the inter-comparison part. The most important results of the inter-comparison should, however, be provided in an appendix or supplement to this manuscript because knowing the differences between data sets is important for interpreting the derived trends.

Following the reviewer's suggestion, we have moved several sections (Seasonal variability analysis, Method of removing the COSMIC sampling errors, and Effects of sampling error removal on the uncertainty of the regional water vapor trend analysis) to the Appendix. The sections moved to the Appendix accumulate to about 10 pages. The main text of the manuscript has been substantially shortened.

A motivation for why a separation into 20° latitude bins has been performed. Accordingly, a motivation for why the three pressure levels 300, 500 and 850 hPa have been used is missing. Why do you look at the seasonal cycle before calculating the trends?

We have added the motivation for selecting three pressure levels in this study in Section 2.2 (L173) as "For RO data, the fine vertical resolution COSMIC RO water vapor profiles are interpolated onto three pressure levels, e.g., 300, 500, and 850 hPa, selected to characterize water vapor variations at representative heights around 9 km, 5.5 km, and 1.5 km, respectively. In particular, the pressure level at 850 hPa is close to the surface, and the COSMIC water vapor retrieval is strongly affected by super-refraction in the moisture-rich regions (Ho et al., 2010). The retrieved water vapor at 850 hPa from COSMIC data could differ from ERA5, making it worth evaluating the relative biases and consistency in the trends between these two datasets. Starting from the pressure level at 500 hPa, the RO-water vapor retrieval uncertainty increases as height decreases. The 300 hPa pressure level represents the water vapor with less horizontal variations at higher heights."

We also added the motivation for separating 20° latitude bins in the latitudinal dependence analysis in Section 3.2 (L251) as "The 20-degree wide latitude bins over northern and southern hemispheres are selected to characterize water vapor latitude-dependence in different reprehensive latitudinal

zones such as 0°-20° for tropical, 20°-40° for sub-tropical, 40°-60° for mid-latitude, and 60°-80° for high-latitude regions.".

We have moved the section on "Seasonal variability of COSMIC and ERA5 water vapor distribution" to Appendix 1 as supplementary information. The seasonal cycles in the monthly-mean water vapor time series need to be removed before calculating the trends (see Appendix 2).

Further, the manuscripts need significant improvements in writing and presentation of the results. Most of the figures and all tables need to be improved. Some figures are not really concise and use to small fonts.

We have made substantial changes to the figures such as replotting with larger fonts, rearranging labels, adding SH/NH separation lines, and rearranging/replotting panels for seasonal variability analysis. The tables (1-6) have all been revised to make them more comprehensible. The quality of the figures (1, 2, 3, 4, 5, 7, A.1, A.2, A.3, A.4) and tables (1-6) have been improved.

Instead of water vapour variation you should clearly state "seasonal cycle and trend". The time period considered could also be mentioned in the title.

We have revised the paper title to "Characterizing the Tropospheric Water Vapor Spatial Variation and Trend using 2007-2018 COSMIC Radio Occultation and ECMWF Reanalysis Data".

The reason of adding "spatial variation' in the title is to emphasize that this paper studies the water vapor variation in different spatial scales such as global, latitudinal and regional, and over three pressure levels.

Since the study of seasonal variabilities of water vapor has been deemphasized in the revised manuscript, we added "Trend" in the title by following the reviewer's suggestion.

**Specific comments:**

P2, L38-41: This paragraph is too general and too broad and thus a bit out of the context of the study and thus not useful at all. The whole paragraph should be removed.

This paragraph has been removed.

P2, L46-47: I would suggest to put the references at the end of sentence.

The references have been moved to the end of the sentence.

P2, L58: Please rephrase the sentence. "such as" is not correct here. It should rather read "and".

 'such as' has been changed to 'and'.

P6, L196: Which selected months? Do you mean January and July? Why have these two months been selected?

We modified the phrase to "two selected months (January and July of 2007)" to make it clear. These two months are selected as two reprehensive months (winter and summer of northern hemisphere) to show the relative seasonal consistency in the comparisons of collocated COSMIC water vapor retrieval versus ERA5 and ERA-Interim water vapor data. We also revised L187 to explain the motivation as "**Error! Reference source not found.** depicts the monthly (using January and July of 2007 as representative winter and summer months of the northern hemisphere) scatter plots of the collocated COSMIC global water vapor versus ERA5 and ERA-Interim water vapor data at three pressure levels." to explain the motivation.

P7, Figure 1: Why is the comparison done for ERA-5 and ERA-interim? Why not only ERA-5?

The UCAR's 1DVAR retrieval algorithm for COSMIC WETPrf (water vapor and humidity) uses ERA-Interim background profiles as the *a priori* input (Wee et al., 2022). In addition, the UCAR water vapor/temperature retrieval also enforces a retrieval constraint to the residual refractivity (refractivity computed from the final temperature and moisture minus the observed refractivity). Such constraint can determine the influence of ERA-Interim on the final water vapor retrieval at different pressure levels. On the other hand, the ERA5 provides a more comprehensive and reliable reanalysis by using improved weather forecast and data assimilation models with various ground, in-situ, and satellite measurements compared to ERA-Interim. To understand the impacts of ERA-Interim on the UCAR 1DVAR COSMIC water vapor retrieval, we use the comparison of the COSMIC retrieval with ERA5 as the reference. We have added these explanations to the paragraph above Figure 1 (L182 to L188).

P9, Figure 2: Put the labels of the panels at the top left of each panel.

The labels in Figure 2 has been placed on the top left of each panel.

P9, Figure 2: I am surprised by the good agreement between the two data sets and was wondering if COSMIC data is assimilated into ERA-5. If yes, this needs to be considered in the interpretation and discussion of the results.

Yes, COSMIC bending angles are assimilated into ERA5, which especially improve the upper-troposphere and lower-stratosphere temperatures (Hersbach et al. 2020). However, the COSMIC 1DVAR retrieval has more independence from its *a priori* (ERA-Interim) for water vapor within the lower/middle troposphere and major information is retrieved from the RO observations at these altitudes, which our study is focused on. As we can see from the quantitative comparisons in Section 3.1, the mean differences between COSMIC and ERA5 global water vapor are 5.67%, -1.86%, and -2.30% for pressure levels at 300, 500, and 850 hPa, respectively. These global and latitude-dependent (Fig. 3) water vapor bias evaluations between COSMIC and ERA5 help understand the extent and regional dependence of the assimilation of COSMIC RO water vapor data in ERA5. We have noted these points at L221 in the revised manuscript.

Hersbash, H., and Coauthors, 2020: The ERA5 global reanalysis. Quart. J. Roy. Meteor. Soc. 146, 1999-2049, https://doi.org/10/1002/qj.3803.

Figure 1-2 and according text could be moved to an appendix/supplement.

We moved three sections with ~10-page figures and texts to the Appendix (see our reply to general comments No.1). We kept Figure 1 and 2 in the main text since these figures and texts are more relevant in the main texts.

P10, L255ff: Is this shift in the NH/SH due to the ITCZ? If yes, then I assume this figure would look different for other months? Which month actually is shown here?

Fig. 3 shows the mean water vapor over 20-degree latitude bins for the collocated COSMIC and ERA5 data averaged over all months of 12 years (2007-2018). This has been noted in the text.

P11, Figure 3: The time period that has been considered should be added. Has an average over the 2007-2018 period been considered? For which month is shown in this figure? Or is here an average over all months/years shown?

For Figure 3, collocated COSMIC and ERA5 water vapor data over all months in 12 years (2007-2018) have been used to calculate the mean water vapor in the corresponding latitude bins. We have added the note to Figure 3 caption to make this clear.

Figure 4-6: I am not happy with these figures. In my opinion these are two overloaded and hardly readable. I am not yet convinced the figure 4b, 5b, 6b. These could be moved in an appendix/supplement. Or consider rearranging the results presented. See the following comment.

Following the reviewer's suggestion, Section 3.1 on "Seasonal variability of COSMIC and ERA5 water vapor distribution", including Figures 4-7, have been moved to Appendix 1. Also, see the reply to the following comment on the revision of Fig. 4-6 (Fig. A. 1 and A.2 in the revised manuscript).

P12-14, Figure 4-6: My suggestion would be to completely change the way of presenting the results for the seasonal cycle. Wouldn't it be better to show the NH and SH separately and then use one figure for each hemisphere showing the results for the three pressure levels. You then could have additionally one figure showing the differences for the three pressure levels (and as now with differences for both hemispheres).

The section on seasonal variability of water vapor distribution with Figures 4-6 has been moved to Appendix 1. In addition, we followed the reviewer's suggestion and revised the figures by showing seasonal variation in NH and SH separately into two figures (Fig. A.1 and A.2) with panels for the three pressure levels in the same figure. The texts have also been revised.

P15, Figure 7: I am also not happy with this figure. Is it really worth to who three pressure levels? The results are quite similar and thus there is no need to show in all figures all three levels. Also I would suggest to improve the figure so that the hemispheres can be better compared. One way of doing this would be to add a vertical line in the middle of the plot separating the NH and SH bars.

Section 3.1, including Figure 7 (Fig. A.3 in the revised manuscript), has been moved to Appendix 1. Also, following the reviewer's suggestion, we added a vertical line in the middle of each panel to separate the NH and SH bars. The texts in the figure caption has also been revised to reflect this change.

P18, L451: Add references.

Reference has been listed.

P19, L472: Which latitude bin? 0-20°, thus tropics? What trend do you derive for the other regions?

Figure 9 (or Figure 4 in the revised manuscript) and related texts (in Section 4.1) discuss the global water trends at three pressure levels, i.e., "COSMIC and ERA5 water vapor trending data in Fig. 4 show that the global water vapor trends at three pressure levels are all positive, suggesting the increase of global water vapor concentration, i.e., becoming globally wetter, over time at these pressure levels." The latitude-dependent water vapor trends are shown and discussed in Section 4.2.

P22, Figure 10: Add a vertical line at 0° to visually separate NH and SH. You could also write in the plot SH and NH, respectively.

Vertical green lines have been added to panels of Figure 5 in the revised manuscript. We also added in the figure caption "The green line in each panel separates the southern (to its left) and northern (to its right) hemispheres" To make it clear.

P23, Table 2: The table should be improved. In it's current form it is hard to read and thus not really useful.

We have revised and reformatted Table 2 to make it more readable.

P26, Figure 12: I don't understand this figure. What exactly has been done? Why is this kind of analyses useful? I think this analyses does not need to be shown in the main part of the paper and could be moved to the appendix.

Following reviewer's suggestions, the Section on "Effects of sampling error removal on the uncertainty of the regional water vapor trend analysis" (Figure 12 and relevant texts in previous draft) has been moved to Appendix 3. This section is to provide supplemental information on the impacts of the sampling error removal on the uncertainties in the regional water vapor trend estimation.

P28, Figure 13: Why is here only 500 and 850 hPa shown and not 300 hPa?

We noted in the paragraph "The distributions of COSMIC and ERA5 water vapor trends at 300 hPa have smaller regional variations. They are not shown in Fig. 7." at L464 in the revised manuscript.

P29, Figure 14 and corresponding text: It should be motivated how these sites have been selected.

We have added the motivation at L509 in the revised manuscript "In the following sections, we selected a few representative sites, such as stratocumulus cloud-rich sites (section 5.2), sites with notable increasing (wetter) and decreasing (drier) water vapor trends (Section 5.3), and sites with a notable difference between ERA5 and COSMIC trends (Section 5.4) to understand the spatial variability of water vapor trends. Their center locations are shown in Fig. 8. These established sites are in 10° by 10° latitude/longitude grids.".

P30-31, Tables 3-5: These tables are hard to read. You should find a way to present the results in a readable way. All additional information could be put into the appendix. E.g. in table 3 columns 2 and 3 could be combined.

Following reviewer's suggestion, we have revised Tables 3-6 by combining columns 2 and 3 into one column. We also removed the brackets in the table cell and reformatted these tables to make them more comprehensible.

P33, L762: ERA-5 is the latest version of reanalyses and has been significantly improved compared to ERA-interim. Thus, it is not astonishing that the agreement between COSMIC and ERA-5 is better than the agreement between COSMIC and ERA-interim. It would be maybe useful to check and discuss the results from the SPARC reananlyses comparison project (https://www.sparc-climate.org/activities/reanalysis-intercomparison/).

We have added in the summary Section "It is noted that the coordinated efforts from Stratosphere–troposphere Processes And their Role in Climate (SPARC) Reanalysis Inter-comparison Project (S-RIP) plays an important role in comparing reanalysis datasets using a variety of key diagnostics and particularly confirmed the significant improvements of the latest version of reanalyses in ERA5 compared to ERA-interim (Fujiwara et al., 2017).".

**Technical corrections:**

P1, L24: sub-tropics → sub-tropical

The term 'sub-tropics' has been changed to 'sub-tropical'.

P1, L28: are around → "are found around" or "are observed around"

The 'are around' has been changed to 'are found around'.

P1, L28: delete "at sites"

"at sites" has been deleted.

P3, L75: in → of

The term 'in' has been changed to 'of'.

P5, L139: Put "in this study" at the begin of the sentence.

"in this study" has been moved to the begin of the sentence.

P5, L141: delete "for global environment and weather studies".

"for global environment and weather studies" has been deleted.

P7, Figure 1: Write "ECMWF" in the x-label instead of "ERA".

The term 'ERA' has been changed to 'ECMWF' in Figure 1.

P8, L220: tropics → tropical

The term 'tropics' has been changed to 'tropical'.

P11, Figure 3: Put labels at the top left of each panel.

Label has been moved to the top left of each panel.

P12, L294-195: same month …..same latitude zones → each month and each latitude zone

'the same month and in the same latitude zones' has been changed to 'each month and each latitude zone'.

P15, L373: "RO" obsolete → delete

The term "RO" has been deleted.

P17, Figure 8: Put labels at the top left of each panel (should be aligned).

Label has been moved to the top left of each panel.

P18, L425: "~ "should be "- "

'~' has been changed to '-'.

P18, L433: Same here.

'~' has been changed to '-'.

P19, Figure 9: Put labels at the top left of each panel.

Label has been moved to the top left of each panel.

P22, Figure 10: Put labels at the top left of each panel.

Label has been moved to the top left of each panel.

P22, Figure 10 caption: What do you mean with "zone—mean"? Zonal mean or do you mean the latitude bins?

We have revised Figure 10 caption to "latitude bin-mean".

P22, L535: add "tropical" after "northern" so that it reads "northern tropical 0° to 20° latitude bins".

‘tropical’ has been added after ‘northern’.

P24, L69: Write either "no data points" or "missing data points".

‘no missing monthly data’ has been changed to ‘no data points’.

P24, L584: [2020] → (2021)

‘[2020]’ has been changed to ‘(2020)’.

P25, L589: [2020] → (2021)

‘[2020]’ has been changed to ‘(2020)’.

P25, Figure 11 caption: either "no monthly data" or " missing data".

‘no missing monthly data’ has been changed to ‘no data points’ in Figure 11.

P26, Figure 12: Put labels at the top left of each panel.

Label has been moved to the top left of each panel.

P27, Figure 13: Put labels at the top left of each panel.

Labels in Figure 13 have been moved to the top left pf each panel.

P32, L717: low latitude → tropical regions

‘low latitude’ has been changed to ‘tropical regions’.

P32, L720 and L727: trending → trend

We changed "trending → trend".

P32, L739: tropics → tropical,  subtropics → subtropical

‘tropics’ and ‘subtropics’ have been changed to ‘tropical’ and ‘subtropical’, respectively.

P33, L751: trending → trends

‘trending’ has been changed to ‘trends’.

P34, L778: from trending → from estimating the trend for 2007-2018 from

‘from trending of 2007-2018’ has been changed to ‘from estimating the trend for 2007-2018 from’.

P34, L792: tropics and sub-tropics region → tropical and subtropical regions

'tropics and sub-tropics region' has been changed to 'tropical and subtropical regions'.

P35, L803 and 810: degree sign misplaced.

Degree sign has been modified.

P35, L805: difficulty → difficult

'difficulty' has been changed to 'difficult'.

P35, L813: having → have

We didn't find the word 'having' in this line.

P35, L814: trending → trend

We changed "trending → trend".

P36, L829: trending → trend estimation

'trending' has been changed to 'trend estimation'.

P36, L846: [2021] → (2021)

'[2020]' has been changed to '(2020)'.

P38, L912: Co-authors are missing.

Co-authors have been listed.

---

## Author Comment (AC2)

**Review of Characterizing the Tropospheric Water Vapor Variation using COSMIC Radio Occultation and ECMWF Reanalysis Data: Shao et al.**

We thank the reviewers for the helpful comments and suggestions. We have previously replied and revised the manuscript according to the first reviewer's comments. Therefore, our further revision of the manuscript and reply to the second reviewer's comment are based on the revised manuscript from the first round. In the following, we summarize our reply to the reviewer.

Title indicates the characterization of water vapour variability using measurements and reanalysis data, but the results and discussion are mostly on the comparison of COSMIC water vapour data with reanalysis at different spatial and temporal scales.

We have revised the title to "Characterizing the tropospheric water vapor spatial variation and trend using 2007-2018 COSMIC radio occultation and ECMWF reanalysis data" following the suggestion from Reviewer #1. This paper focuses on quantifying the consistency and differences of the spatial variability and trend of water vapor from COSMIC RO retrievals and ERA5 reanalysis data. The COSMIC and ERA5 water vapor data at lower (850 hPa), mid- (500 hPa), and upper troposphere (300 hPa) from 2007 to 2018 are compared in terms biases and trends over spatial scales ranging from global, latitudinal to regional. The general consistency between the two datasets demonstrated in this study help cross-validate the water vapor variation and trend at different scales and assure the quality of these datasets for climate studies. We also showed the differences such as the biases at different pressure levels and the differences in water vapor trend estimation between COSMIC and ERA5 over regions in the ITCZ. We provided explanation for the negative water vapor biases in lower troposphere and the need to address and resolve super refraction or ducting in RO 1DVAR retrieval. We agree with the reviewer and point out that further studies with other long-term water vapor data will be needed to resolve the COSMIC vs. ERA5 water vapor trending differences.

**Major:**

1. Its global comparison, why cannot use ground-based observations like radiosonde, GNSS, GPS, which are commonly used for validation and comparisons. This is very important as reanalysis data can have a relatively large bias in some regions (e.g. tropics, what we have found in our studies).

The ground-based radiosonde, GNSS or GPS observations cover mostly over land. The radiosonde water vapor measurements can differ by different type of radiosondes, e.g., RS41 vs. RS92, and the calibration or correction implemented at the stations (Ho et al., 2010; Sun et al., 2019; Ho et al., 2020a; Shao et al., 2021b). Radiosonde measurement is more suitable to study regional water vapor variability and trends over land. Inter-calibration among different radiosonde stations will be needed to assess global water vapor trend over land. For ground-based GNSS stations, there have been ongoing work of deriving global total column water vapor time series from their observations and check its consistency with other type of observations. Fig. 1 (from Ho et al., 2020a) shows example of global water vapor trend comparison over ocean and land with multiple datasets. The global water vapor trend from GNSS station over land is in general consistent with COSMIC and other reanalysis model results. We agree with the reviewer that the comparison with

independent ground-based observations will help address the biases and trend differences between RO and reanalysis model over land.

We cited the paper by Mears et al., 2017 and Patel and Kuttippurath, 2022 and added in Section 6 about the importance of using ground-based GNSS and GPS data for global and regional validation and using radiosonde for regional validation of water vapor data from RO and reanalysis.

[Figure]

(a) Oceans

(b) Land

Figure 1: Global mean total precipitable water vapor annual anomalies for (a) ocean only and (b) land only for observations and reanalysis averaged over 60°S to 60°N. The shorter time series have been adjusted so that there is zero mean difference relative to the mean of the three reanalyses over the 2006–14 period (constructed from same data as in Mears et al. 2017, their Fig. 2.16).  (from Ho et al. (2020a))

Mears, C., S. P. Ho, J. Wang, H. Huelsing, and L. Peng, 2017: Total column water vapor [in "States of the Climate in 2016"]. *Bull. Amer. Meteor. Soc.*, **98** (8), S24–S25, https://doi.org/10.1175/2017BAMSStateoftheClimate.1.

Patel, V. and Kuttippurath, J.: Significant increase in water vapour over India and Indian Ocean: Implications for tropospheric warming and regional climate forcing, Science of Total Environment, https://doi.org/10.1016/j.scitotenv.2022.155885, 08 May 2022.

2. You have shown the bias and differences, but no valid reasons are given. Please discuss the reasons for the differences

We added in Section 6 to discuss about the relative water vapor biases between COSMIC and ERA5 at three pressure levels.

[revised manuscript text omitted]

**Specific Comments:**

**Line 110-117:** Please move these sentences to Data section, where COSMIC water vapour description is given.

L110-117 have been moved to Section 2.2 COSMIC WETPrf water vapor retrieval.

**Line 147:** The ERA5 water vapour…….pressure levels. This sentence is about the availability of ERA5 water vapour at different pressure levels, so please move this to lines 140-145.

This sentence has been moved to the suggested location.

**Line 182:** Why only three pressure levels? 850, 500 and 300 hPa, and why these particular altitudes?

We have added the motivation for selecting three pressure levels in this study in Section 2.2 (L173) as "For RO data, the fine vertical resolution COSMIC RO water vapor profiles are interpolated onto three pressure levels, e.g., 300, 500, and 850 hPa, selected to characterize water vapor variations at representative heights around 9 km, 5.5 km, and 1.5 km, respectively. In particular, the pressure level at 850 hPa is close to the surface, and the COSMIC water vapor retrieval is strongly affected by super-refraction in the moisture-rich regions (Ho et al., 2010). The retrieved water vapor at 850 hPa from COSMIC data could differ from ERA5, making it worth evaluating the relative biases and consistency in the trends between these two datasets. Starting from the pressure level at 500 hPa, the RO-water vapor retrieval uncertainty increases as height decreases. The 300 hPa pressure level represents the water vapor with less horizontal variations at higher heights."

**Line 186:** Give references

We added the reference "Fujiwara et al., 2017 and Hersbach et al., 2020".

**Figure 1 and Line 185-197:** What is the need of comparing COSMIC water vapour with both ERA5 and ERA-Interim, if it is already stated in Line 185-186 that the ERA5 water vapour retrieval is better than that of ERA-Interim? Also, why only January and July are considered here"?

The UCAR's 1DVAR retrieval algorithm for COSMIC WETPrf (water vapor and humidity) uses ERA-Interim background profiles as the *a priori* input (Wee et al., 2022). In addition, the UCAR water vapor/temperature retrieval also enforces a retrieval constraint to the residual refractivity (refractivity computed from the final temperature and moisture minus the observed refractivity). Such constraint can determine the influence of ERA-Interim on the final water vapor retrieval at different pressure levels. On the other hand, the ERA5 provides a more comprehensive and reliable reanalysis by using improved weather forecast and data assimilation models with various ground, in-situ, and satellite measurements compared to ERA-Interim (Fujiwara et al., 2017 and Hersbach et al., 2020). To understand the impacts of ERA-Interim on the UCAR 1DVAR COSMIC water vapor retrieval, we use the comparison of the COSMIC retrieval with ERA5 as the reference. We have added these explanations to the paragraph above Figure 1 (L182 to L188).

The two months (January and July) are selected as two representative months (winter and summer of northern hemisphere) to show the relative seasonal consistency in the comparisons of collocated COSMIC water vapor retrieval versus ERA5 and ERA-Interim water vapor data. We also revised L187 to explain the motivation as "**Error! Reference source not found.** depicts the monthly (using January and July of 2007 as representative winter and summer months of the northern hemisphere) scatter plots of the collocated COSMIC global water vapor versus ERA5 and ERA-Interim water vapor data at three pressure levels." to explain the motivation.

**Section 3.1** Global distribution of water vpour:

Why authors have shown the distribution of water vapour at 10-degree latitude and longitude grid not in the original resolution of COSMIC and ERA5? If bias is computed at a coarser spatial resolution, there might be a chance of large uncertainty and the regional variability will not be reflected in the bias estimates.

The 10×10 degree latitude and longitude grids were chosen to match the later discussion of regional trends over the same 10×10 degree resolution grids. While ERA5 can have finer uniform longitude and latitude grids, the COSMIC profile locations are non-uniform. We have to specify grids with finite latitude and longitude bins to organize the COSMIC data. As we showed in Figure 6 (The percentage of missing monthly data over the 2007 to 2018 interval on the global 10°×10° grids), there can be locations with missing monthly data on the global 10°×10° grids due to the varying number of COSMIC data. Our analysis excludes grids with > 2-month missing monthly data from the trend calculation. We agree with the reviewer that it is a tradeoff of choosing finer grids to reduce regional variability and uncertainty while keeping sufficient samples in the grid. The 10×10 degree grids seem to be optimal for both COSMIC bias and trend estimation.

**Line 232:** Why COSMIC water vapour overestimates ERA5 in the upper troposphere?

The main cause for higher water vapor retrieved by COSMIC than ERA5 at 300 hPa can be due to the low concentration of water vapor in the upper troposphere and the large uncertainty in retrieving water vapor in the reanalysis model. Johnston et al. [2021] analyzed COSMIC-2 and reanalysis models (ERA5 and MERRA2) water vapor difference in different latitude zones and their results are shown in the figure below. The UCAR COSMIC-2 water vapor retrieval is consistently lower than both ERA5 and MERRA2 water vapor data in the lower troposphere (below 2 km). However, COSMIC-2 water vapor retrieval is higher than ERA5 data and lower than MERRA2 data at heights above 4 km. The magnitude of COSMIC-2 vs. ERA5 water vapor difference is smaller than that of COSMIC-2 vs. MERRA2. The opposite sign and large magnitude of the ERA5 and MERRA2 model water vapor differences relative to COSMIC-2 in the upper troposphere suggest the large uncertainties in retrieving water vapor in reanalysis model over this height region.

[Figure]

Figure 2: COSMIC-2 and Reanalysis Model (ERA5 and MERRA2) mean water vapor and water vapor difference comparison in different latitude zones (from Johnston et al. (2021)).

Johnston, B.R., Randel, W.J., Sjoberg J.P.: Evaluation of Tropospheric Moisture Characteristics Among COSMIC-2, ERA5 and MERRA-2 in the Tropics and Subtropics. *Remote Sensing*. 13(5), 880, DOI: 10.3390/rs13050880, 2021.

**Line 232-233:** Since the water vapour concentration at 300 hPa is very small, its contribution to the total precipitable water would also be very small.

We have revised this sentence according to the suggestion.

**Section 3.3** Seasonal variability of COSMIC and ERA5 water vapour distribution: If you want to discuss the seasonal variability, discuss the seasonal changes and then present the bias. Also, why authors have divided the latitude in 20-degree interval here? Why not tropics, mid-latitude and Polar Regions then?

This section on seasonal variability analysis has been moved to Appendix 1 following the first reviewer's comments. We followed the reviewer's suggestion and moved the section on seasonal changes to where before discussing the biases between COSMIC and ERA5.

The 20-degree wide latitude bins over northern and southern hemispheres are selected to characterize water vapor latitude-dependence in different reprehensive latitudinal zones such as 0º-20º for tropical, 20º-40º for sub-tropical, 40º-60º for mid-latitude, and 60º-80º for high-latitude regions. In this way, we can also study the differences in the northern and southern hemisphere. When discussing the seasonal variability in tropical region, we do combine the -20 to 20 latitude bins.

**Line 339-341:** It is already mentioned in the previous section "Decline in water vapour in southern hemisphere is faster than the northern hemisphere"

This section have been moved to Appendix 1 and we removed the sentence "In Fig. A.3, the decrease of $Q_{max,COSMIC}$ as |latitude| increases from 20º are more rapid in the southern hemisphere than in the northern hemisphere,….." in the revised manuscript.

**Line 364:** This sampling error does not affect the bias discussed in the previous section? If it is, then how authors have addressed this issue?

This sampling error affects the trend estimation and does not affect the relative biases between COSIC and ERA5 discussed in the previous section. The relative biases are estimated from the collocated COSMIC and ERA5 data.

**Line 386-387:** Sampling error for COSMIC ? Also, for ERA 5?

Thanks for catching this. It is a bit misleading. The sampling error removal is only for COSMIC. As noted on L, for ERA5 data, the application of sampling error $Q_{SE}$ removal to $\overline{Q_{ERA5\_sample}}$ essentially recovers $\overline{Q_{ERA5\_RoI}}$. So, when calculating ERA5 trend, we only need to calculate $\overline{Q_{ERA5\_RoI}}$ and don't need to apply sampling error removal. We made the correction "we need to apply sampling error removal to COSMIC data".

**Line 408-409:** "which is mainly due to the difference between the orbital-specific distribution of COSMIC RO observations and uniformly-distributed global ERA5 data". Give references for this statement. Also, how orbital-specific distribution of COSMIC RO observations cause oscillations in the sampling error?

We added a reference: Ho et al. (2020). Example of monthly local time and latitudinal distribution of RO profiles retrieved from COSMIC-1 data are shown in Fig. 3 (see below). We can see the nonuniform distribution of the COSMIC-1 profiles in both local time and latitudes, which is due to the limited local time and latitude coverage of the orbits of the small satellites in the COSMIC-1 constellation. The non-uniform local time and latitude distribution of COSMIC-1 profiles coupled with the annual variation of the Sun's declination contribute to the seasonal oscillation in the sampling error time series.

[Figure]

Figure 3: Example of monthly local time and latitudinal distribution of RO profiles retrieved from COSMIC-1 data.

**Line 411:** Why COSMIC sampling decreases significantly after 2017?

This is due to that three of the six small satellites in the COSMIC-1 constellation stopped working during the time interval from 2015 to 2017. After 2017, there were only two small satellites in COSMIC-1 that were still in operation. See Figure X and answer to the next question for more details.

**Figure 8:** Sampling is very small in 2011 as compared to that in 2007-2009. Its almost constant in 2011-2014, and then decreases until 2019. Why these disparities in the sample numbers?

There are six small satellites (C1E1 to C1E6) in COSMIC-1 constellation. The service interval and performance of these six satellites vary over time. Figure below shows the variation of the monthly profile numbers of these six small satellites in COSMIC-1. C1E3 is the first satellite that stopped producing data in the middle of 2010. C1E2, C1E3 and C1E4 ended their operations over the time interval from 2015 to 2017. C1E1 and C1E6 continued in operation until the middle of 2019 and early 2020, respectively. Due to the varying performances and availabilities of C1E1 to C1E6, the time series of the combined valid profile numbers from these six satellites thus show the pattern shown in Figure A.4 (Figure 8 in previous draft).

[Figure]

Figure 4: Time series of monthly profile number of the six small satellites (C1E1 to C1E6) of COSMIC-1.

**Figure 9:** Water vapour is increasing from 2008 to 2010, almost constant from 2011 to 2014, then again increased during the period 2014-2017, and finally it shows constant (i.e., no trends) at all three pressure levels. Why these particular distributions? Discuss

The two intervals of water vapor increase over 2009-2010 and 2015-2016 are associated with the two large El Nino events during these two periods. These warm events can enhance surface evaporation, increase tropospheric water vapor, and warm the entire tropical troposphere (e.g., Zveryaev and Allan 2005; Trenberth et al. 2005). However, even without the ENSO impact, global mean tropospheric water vapor especially in the tropics still shows evident upward trend following global surface warming (e.g., Allan et al. 2022).

Trenberth K.E., J. Fasullo, and L. Smith, 2005: Trends and variability in column-integrated atmospheric water vapor. Clim. Dyn., doi:10.1007/s00382-005-0017-4.

Zveryaev, I.I. and R.P. Allan, 2005: Water vapor variability in the tropics and its links to dynamics and precipitation. J. Geophys. Res.-Atmos., 110, D21112, doi:10.1029/2005JD006033.

Allan et al. (2022): Global changes in water vapor 1979-2020. JGR-Atmos, 127, e2022JD036728. https://doi.org/10.1029/2022JD036728.

**Line 480:** How these results can be consistent or even comparable with Chen and Liu (2016)? They have computed the PWV trends (entire column of water vapour). Here only three pressure levels are taken. Please cite some other references, in which tropospheric water vapour trends are computed.

We added a reference to Allan et al. (2022). The revised texts read "Allan et al. (2022) studied the global-scale changes in water vapor and responses to surface temperature variability since 1979 using coupled and atmosphere-only CMIP6 climate model simulations. In the water vapor trend estimation over the 1988 to 2014 period, Allan et al. (2022) showed positive increase of global water vapor at near surface, at 400 hPa and Column Integrated Water Vapor from ensemble of climate model simulations with the CMIP6 historical and amip experiments. The period of COSMIC RO data studied in this paper partially (2007 to 2014) overlaps with the simulations from Allan et al. (2022). The increasing trend in the global atmospheric water vapor concentration at three pressure levels from our trend analysis is generally consistent with the results from Allan et al. (2022)".

**Line 488-491:** Again, Chen and Liu (2016) is used here for the comparison.

We removed the citation to Chen and Liu (2016) and added citation to Allan et al., (2022). The sentence has been revised as "It was also shown in Allan et al. (2022) that in the ensembled historical experimental model simulations, the water vapor increases by 1.53 and 3.52 %/Decade at surface and at 400 hPa, respectively. Our study shows that the increasing global water vapor trends estimated with 2007-2018 COSMIC data are 2.03±0.65, 3.25±1.25, 3.47±1.47 %/Decade at 850, 500 and 300 hPa, respectively, which are in general agreement with the results from in Allan et al. (2022), considering that the two work cover two distinct periods with 8 overlapped years. In Allan et al. (2022), there is an increase of water vapor trend from surface to at 400 hPa by ~2 %/Decade. Our work shows an increase of positive water vapor trend by 1.44 %/ Decade when height varies from near surface (at 850 hPa) to 300 hPa, which is also in general consistent.".

**Line 523-527:** It can't be directly attributed to the dry atmosphere.

We revised the sentence as "The only latitude bin with a small negative water vapor trend with large uncertainty is in the -80º to -60º southern high latitude bin at 500 hPa. From the global surface temperature trend analysis by Gu and Adler, 2022, there is a mixture of weak decreasing trend in the surface temperature at the Southern Ocean around Antarctic and an increasing trend of over Antarctic in the 60 to 80 degree southern latitude zone. However, the uncertainties of estimating both the temperature and water vapor trends in this latitude zone are large."

See Figure 5 below for the global surface temperature trend map from Gu and Adler, 2022.

[Figure]

*Figure 5: Linear trend in global surface temperature (K/decade) during 1998-2020.*

Gu, G., and Adler, R. F.: Observed Variability and Trends in Global Precipitation During 1979-2020. *Clim. Dyn.*, https://doi.org/10.1007/s00382-022-06567-9, 2022.

**Line 531:** What do you mean by the most stable water vapour trend?

"most stable" is not correct. It should be "the lowest water vapor trend".

**Line 622-623:** This sentence about sea surface temperature has no meaning here. Better to write the trends in sea surface temperature, which can influence the water vapour trends.

This sentence has been changed to: "Sea surface temperature has been increasing in the western Pacific during the recent decades (e.g., Gu and Adler 2022)".

Gu, G., and R. F. Adler, 2022: Observed Variability and Trends in Global Precipitation During 1979-2020. *Clim. Dyn.*, https://doi.org/10.1007/s00382-022-06567-9.

**Line 625:** "Indo-Pacific warm pool region and increase in the equatorial region of the Pacific Ocean is what we here observe." I do not see any analysis here for making this statement.

We have revised the sentence to "Sea surface temperature has been increasing in the western Pacific during the recent decades (e.g., Gu and Adler 2022). There is a high correspondence with regards to the trends in sea surface temperature and tropospheric water vapor in the western Pacific during the recent decades (e.g., Gu and Adler 2013). It was shown by Chen and Liu (2016) that the moderate increase in surface temperature over the Pacific Ocean could cause the PWV to increase in the equatorial region of the Pacific Ocean and decrease in this Indo-Pacific warm pool region, which is what we observe here. Further quantitative analysis of trends at selected locations

in the Pacific Ocean (Sites # 4, 16 in Fig. 8) and in Indo-Pacific warm pool region (Site #14 in Fig. 8) will be performed in the following sections.".

Gu, G., and R. F. Adler, 2013: Interdecadal Variability/Long-Term Changes in Global Precipitation Patterns during the Past Three Decades: Global Warming and/or Pacific Decadal Variability? *Clim. Dyn.*, 40, 3009-3022. doi: 10.1007/s00382-012-1443-8.

**Line 626-630:** How monsoon climate and precipitation affect the trends in water vapour in these regions? Precipitation is known for the sink of water vapour. Discuss this.

We added the discussion as "This region is affected by the monsoon climate over the south of the Himalayas, resulting in a sizeable regional change in precipitation at different seasons. Indian Ocean is the essential part of the coupled Indian monsoon system because it feeds the moist convection over both land and ocean. Convection, precipitation, and water vapor are also a fully coupled process. It is shown that the Indian Ocean has been warming up during the recent decades (see Figure 5 from Gu and Adler, 2022), which is the driver for positive water vapor trend in this region."

**Figure 14:** How these sites are selected?

We have added the motivation at L509 in the revised manuscript "In the following sections, we selected a few representative sites, such as stratocumulus cloud-rich sites (section 5.2), sites with notable increasing (wetter) and decreasing (drier) water vapor trends (Section 5.3), and sites with a notable difference between ERA5 and COSMIC trends (Section 5.4) to understand the spatial variability of water vapor trends. Their center locations are shown in Fig. 8. These established sites are in 10$^o$ by 10$^o$ latitude/longitude grids."

In Section 5.2, 5.3 and 5.4, we explained the selection of the specific group of the representative sites.

**Section 5.2:** Without analysing cloud data how authors identified the regions of Stratocumulus clouds?

The three regions rich of Stratocumulus clouds are selected according to the regions identified in Wood et al., 2011; Wood, 2012 and Ho et al., 2015. We have added these references.

Ho, S.-P., L. Peng, R. A. Anthes, Y.-H. Kuo, and H.-C. Lin 2015: Marine boundary layer heights and their longitudinal, diurnal, and interseasonal variability in the southeastern Pacific using COSMIC, CALIOP, and radiosonde data. *J. Climate*, **28**, 2856–2872, https://doi.org/10.1175/JCLI-D-14-00238.1.

Wood, R., Mechoso, C. R., Bretherton, C. S., Weller, R. A., Huebert, B., Straneo, F., Albrecht, B. A., Coe, H., Allen, G., Vaughan, G., Daum, P., Fairall, C., Chand, D., Gallardo Klenner, L., Garreaud, R., Grados, C., Covert, D. S., Bates, T. S., Krejci, R., Russell, L. M., de Szoeke, S., Brewer, A., Yuter, S. E., Springston, S. R., Chaigneau, A., Toniazzo, T., Minnis, P., Palikonda, R., Abel, S. J., Brown, W. O. J., Williams, S., Fochesatto, J., Brioude, J., and Bower, K. N.: The VAMOS Ocean-Cloud-Atmosphere-Land Study Regional Experiment (VOCALS-REx): goals,

platforms, and field operations, Atmos. Chem. Phys., 11, 627–654, https://doi.org/10.5194/acp-11-627-2011, 2011.

Wood, R., 2012: Stratocumulus clouds. Mon. Wea. Rev., 140, 2373– 2423, doi:10.1175/MWR-D-11-00121.1.

**Line 661:** "RO data can penetrate the cloud, and the water vapour retrieval from RO data is not affected by the stratocumulus cloud." Reference for this statement.

RO signal can penetrate the cloud because the wavelengths for L1 and L2 frequency of radio occultation signals are around 19 cm and 24.2 cm, respectively, which are much larger than the size of cloud water droplets and ice crystals (Kursinski et al., 1997).

Kursinski, E. R., , G. A. Hajj, J. T. Schofield, R. P. Linfield, and K. R. Hardy, 1997: Observing Earth's atmosphere with radio occultation measurements using the Global Positioning System. J. Geophys. Res., 102, 23 429–23 465, https://doi.org/10.1029/97JD01569.

**Line 675:** "The possible cause of smaller trends from ERA5 water vapour data over stratocumulus cloud-rich regions could be difficulty in accurately estimating water vapour at low height in ERA5 reanalysis data compared with COSMIC RO measurements". Can you provide the reference for the statement?

We added the reference to Lonitz and Geer 2017.

Lonitz, K., and Geer, A.: Effect of assimilating microwave imager observations in the presence of a model bias in marine stratocumulus, EUMETSAT/ECMWF Fellowship Programme Research Reports, https://www.ecmwf.int/node/17164, 2017.

**Section 5.3:** What is the basis for the selection of these sites?

To select sites with notable increasing and decreasing water vapor trends shown in Section 5.3, we searched the 10×10 degree global grids and identified the regions with the largest increasing and decreasing water vapor trends. Within these regions, we selected the representative sites and these sites are listed in Table 4 and Table 5.

**Line 695:** Where is the analysis of trends in ocean surface temperature?

We have added the citation to the ocean surface temperature studies and revised the sentence as "Many previous studies have explored the trends in surface temperature (e.g., Gu and Adler, 2022 and references therein). Global surface keeps warming up, though with rich spatial structures of temperature change. From the study by Gu and Adler, 2022, ocean surface warming can readily be seen in the Indian Ocean and tropical Pacific Ocean, roughly corresponding to the strong increasing tropospheric water vapor trends for Site#4, #5, and #8 we observed." See Figure 5 in this reply (from Gu and Adler, 2022) on the global trend in NASA GISS surface temperature during 1998-2020, roughly corresponding to the period focused in our study.

**Line 726-729:** For site#17 ………..Pacific Ocean is on the west. The reasons stated for the decline in water vapour at site#17 are not convincing.

We did more research on this and attribute the decline in water vapour at site#17 to the regional sea surface temperature decrease in this region. From the above figure "Linear trend in global surface temperature (K/decade) during 1998-2020" from Gu and Adler 2022 in our reply to Line 695, we can see an overall temperature decrease in this region at Site #17. We have revised the corresponding text "From the study of linear trend in global surface temperature during 1998-2020 by Gu and Adler, 2022, there is a trend of decreasing ocean surface temperature (~-0.1 K/Decade) near Site #17, which matches the decrease of water vapor observed by COSMIC.".

**Line 729-730:** Water vapour at 850 hPa is not a precipitable water vapour. Also, there is no "near-surface precipitable water vapour".

We have removed the word "precipitable".

**Line 731**: Again, precipitable water vapour, it just water vapour at 850 hPa.

We have removed the word "precipitable".

**Line 732:** Earlier it is mentioned that COSMIC measurements are not affected by stratocumulus cloud, then how it becomes more challenging here?

The original sentence says "which makes it more challenging to accurately estimate $D_{Q,ERA5}$ than $D_{Q,COSMIC}$" and it intends to state that it is more challenging to accurately estimate water trends from ERA5 data than from COSMIC data. This is consistent with the earlier statement that COSMIC measurements are not affected by stratocumulus cloud.

We have revised the sentence to "which makes it more challenging to accurately estimate water trends from ERA5 data than from COSMIC data".

**Section 6:** Most of the results and discussion are repeated here with the same references. Please rewrite this section and draw a solid conclusion.

We made the following changes to Section 6.

- We only listed the key findings from this study in this section.
- We added new citation to Fujiwara et al., 2017.
- We added new discussion about the COSMIC vs. ERA5 biases at three pressure levels and cited new references (Sokolovskiy, 2003; Ao et al., 2003; Xie et al., 2006; Ao, 2007; Xie et al., 2010).
- We removed summary about seasonal variability.
- We also added "In particular, the comparison with long-term ground-based GNSS and GPS data (Mears et al., 2017) and radiosonde data (Patel and Kuttippurath, 2022) can help address the biases and trend differences between RO and reanalysis model over land."

Also, please crosscheck the citation Liu et al. (2016) in Line 838.

We have changed the reference to Allan et al., 2022.

---

## Referee Report (RR1)

**Review on Manuscript No. ACP-2022-660**

Shao et al.: Characterizing the tropospheric water vapor spatial variation and trend using 2007-2ß18 COSMIC radio occultation and reanalysis data

The manuscript has significantly improved, but still lacks in the quality of the language and a clear writing. There are lot of small errors. It would have been good if the authors would have spent more effort in a careful check of the manuscripts before submission. To list all this small errors is for me as referee really time consuming and distracts me from really focusing on the contents of your study.

**General comments:**

The abstract is a bit too long and detailed on the results. I would suggest to skip the explicit numbers of the trend and rather qualitatively state if the trend is positive or negative (and give the order, i.e a few percent). Further, it should be more clearly stated which comparisons have been made instead of writing in detail which differences where derived in which area. I also think mentioning that the El Nino water vapor increases are visible in the data is rather obsolete in the abstract. Write down some key points and make out of these a clear structured abstract.

I don't understand the difference between slope and trend. Isn't the slope of the linear fit the trend?

The result session is still too descriptive and lacks explanations.

The term trending should be replaced by trend, wetter should be replaced by moister, heights by altitudes.

There is still too much data analyses shown in the result section. It is not clearly motivated what the intention is of picking all these sites and looking at them. The paper is in my opinion somewhat overloaded and hard to follow. Especially since you have trouble expressing yourself it makes following the discussion really difficult. In this case a shorter more concise study would be more beneficial.

Section 5 should either significantly improved or omitted. At the moment it makes only the paper longer, but does not provide any knowledge gain.

For example Section 5.3. Why is this studied? It's just a listing of trend estimates without explaining or discussing the cause or consequence of these increasing trends.

Section 5.4 is quite confusing. Here you refer to Figure 7, but this Figure has already been described in Section 5.1. Are you using this figure for both sections? Is the reference really correct? You should more clearly state here what is shown in this figure and refer to Section 5.1.

Further, I am not satisfied with your answer why you selected the altitudes 500 and 300 hPa. You should provide a motivation with respect to atmospheric processes or a certain atmospheric region, e.g. stating that you with these levels cover the entire troposphere (lower, middle, higher troposphere) would be fine with me. However, how can you know what differences and uncertainties to expect? Isn't that what you derive from this study? I think you should improve this paragraph (Sect. 2). Further, you should provide an explanation why you expect this differences.

I think also the conclusion and discussion should be improved. You just squeezed in there some answers to my comments, which however feel there rather lost and out of context.

I have listed my specific and technical comments below.

**Specific and technical comments:**

P2, L25: qualities -> quality

P2, L59: skip "and" before microwave

P2, L63: skip "and others". I would say that this is obsolete since you already write in the beginning of the sentence "mainly"

P3, L65: in -> for, so that it reads "for long-term……."

P3, L66: I would not use the term "monitoring" for reanalysis data. This is a term that should be rather used with measurements. Thus, I would suggest to write: These atmospheric reanalysis data have been used for understanding (or investigating) long-term atmospheric water vapor variability and trends (or more general changes).

P3, L82: the rise -> the increase

P3, L88: data -> datasets

P3, L90: add "ERA" before Interim

P3, L90: better than reanalysis from -> better than the reanalysis data (add "the" and "data")

P3, L92: What exactly do you mean with "other sensor data"? Please clarify and rephrase.

P3, L94: assure the climate community with -> provide to the climate community (replace assure by provide to and delete with)

P4, L101: microwaves -> microwave

P4, L102: Add "Further," before "RO-derived".

P4, L108: skip comma and add "for" and "and", so that it reads "for climate and meteorological research"

P4, L110: distribution -> distributions

P4, L111: from 2007 to 2018 -> for the time period from 2007 to 2018

P4, L121: replace "As supplementary" by "Additionally"

P4, L122: add "the" -> in the Appendix

P4, l122: ……with introduction to estimating the water vapor trend -> you mean with introducing (or rather describing) the estimation of the water vapor trend? Please check and rephrase (correct English grammar)

P5, L138: occulted by the Earth's atmosphere -> not correct, please rephrase.

P5, L140: Start the sentence with "From" replace "data" by observations and write "are derived", thus "From the retrievals …… first the bending angle……are derived"

P5, L144: profile -> profiles

P5, L146: 2007 to 1028 _> 2007 to 2018

P6, L166: before the study? Please rephrase? Do you mean before the analyses has been performed?

P6, L169: add "the" -> For the RO

P6, L173: could -> may

P8, L231: troposphere -> tropospheric

P8, L233: small -> low

P8, L233: skip "also" and rephrase the next sentence as follows: "The main cause that at 300 hPa higher water vapor from ERA5 than from COSMIC is derived is due….. estimating water vapor….." You can retrieve data, but from a model the data is simulated or estimated.

P9, L247: delete "model"

P9, L248: in consistently -> is consistently

P9, L249: not the "retrieval", but the "retrieval data", thus change "retrieval" -to "retrieval data"

P10, L262: 20-degree-latitude-bin-averaged -> please rephrase and write averaged over 20 degree bins, thus you could write "averaged over 20 degree latitude bins at the three selected pressure levels (300, 500 and 850 hPa).

P10, L266: in Fig. 3 -> shown in Fig. 3 or just write Fig. 3 in parentheses -> (Fig. 3)

P10, L272: a wetter what? A wetter atmosphere? Skip? Rephrase?

P10, L278: wetter -> moister

P10, L279: Sentence not clear. Please rephrase.

P10, L281: Which factors? Clearly state what you are referring to.

P11, L282-283: What do you mean with "warmer NH" and "colder SH"? The winter and summer hemispheres?

P11, L286: grow -> increase

P11, L286-287: Sentence is not clear. Something is missing here.

P11, L289: These factors? Do you mean these conditions? Not clear what you exactly mean.

P11, L289: How or why? How can temperature differences affect water vapor? You just described temperature differences. How is temperature connected to water vapor?

P11, Figure 3 caption: skip "20-degree-latitudinal……" Just write in the first sentence what is compared and then in the second sentence how the data has been treated.

P11, L291: retrieval -> observation or retrieved data

P11, L295: over all months in 12 years -> for all months of the considered 12 year period

P12, L297: add "shown" before "in the middle……."

P12, L310: put Fig 3h and 3f in parenthesis

P12, L311: Fig 3h shows -> From Fig 3h it can be seen…..

P12, L314: Same here: From Fig 3j it can be seen…..

P12, L316: add "differences" before being

P12, L3198-319: Skip "after sampling……". You don't need this in the section title. It is enough to mention this in the text part.

P12, L323: Sampling removed water vapor -> rephrase. Rather write COSMIC water vapor data where sampling errors have been removed.

P12, L323: in the rest of this paper -> in the remainder of this paper, but better would be to write in the following.

P12, L325: This section compares. Not the section is doing the comparison, but you. Correctly it should read "In this section………are compared".

P13, L335: Change "It is noted" to "It becomes visible" or "It can be seen"

P13, L341. What is NINO3.4 and NIN4? An explanation should be given in the text.

P13, L342: add "the" before April

P13, L346: Add "the" before seasonal

P13, L346: change "as seen" to "as visible" or even better skip this and put Fig 4a in parenthesis at the end of the sentence.

P14, Figure 4 caption: Needs to be improved. Too much repetition. First sentence obsolete? Trending should be replaced by trend (throughout the manuscript).

P1, L366: trending -> trend and put Fig 4a in parenthesis

P15, L366: Add time period after water vapor concentration.

P15, L369: with -> using, delete of ECMWF data or write ECMWF ERA-40 data

P15, L369, 370 and 371: delete "in" and put references in parenthesis.

P15, L373: were trended -> were used to derive the trend

P15, L378: move the time period behind paper

P15, L379: at three -> at the three and replace from by considered in

P15, L384: add "the" -> the three

P15, L386: trends -> trend, with -> for the and move the time period data

P15, L387: is -> are

P15, L388: overlapped -> overlapping

P15, L390: Just write "a positive water vapor trend of " or " a positive water vapor trend of 1.44%". If you write positive or negative you do not need to give a number. Vice versa, If you give the number then you do not need to write positive or negative.

P16, L404: trending -> trend

P16, L408 and 411: What do you mean with trending slope? Is not the trend the slope of the linear fit?

P16, L409: in -> at and delete "latitude range" at the end of the sentence.

P16, L416: in should be replaced by at or for

P17, L426: "mixed with" should rather read "composed of" or "consisting of".

P17, Figure 5 caption: Comparing -> Comparison of (two occasions)

P17, Figure 5 caption L437: skip retrieval.

P17-18: Figure 5: The sentence starting with "The bar ....." is not clear and needs to be rephrased.

P18, L445ff: In most occasions you can skip writing northern and southern. IF you provide the coordinates with plus and minus signs this is enough.

P18, $49: within -> of

P18, L453: Put Fig 5f in parenthesis and delete "in"

P18, L454: Why -60 to 80? Is that correct? Or should it be -60 to -80?

P19, L457: Put "Figure 5g and Table 2" in parenthesis and delete "show that"

P19, L460: high -> higher

P19, L461: estimations -> estimated

P19, $63: with latitude bins needs to be rephrased.

P19, L460ff: I thought you made a separation into 20 degree bins, Why are in this section larger bins discussed?

P19, L466-467: Sentence not clear. Needs to be rephrased.

P19, L475: interval -> period

P19, L476: the period -> this time period

P19, L476: delete distributed and write "mostly fond over the ......"

P19, L377: above -> greater than

P19, L478: Sentence not clear. Please correct.

P19, L478:trending -> trend

P20, Figure 6 caption: no monthly data missing? What do you mean here? With no data or with missing data? I guess the grids with missing data are shown as white blanks.

P20: I do not understand what you mean here with interface.

P21, Figure 7 caption: In two occasions trending should be replaced by trend.

P22, General: Move the latitudes und longitudes behind the respective region, e.g Laccadive Sea

P22, L533: Here you write this region, but two different latitude and longitudes are given. Do you mean "these regions"? And which regions are you talking here about? The text would be much easier to read if you would put the coordinates at the end and not always after "region".

P22, L544: "These established sites are in 10 by 10 longitude/latitude grids". Not clear what you mean since sentence not grammatically correct. What do you mean with established?

P23, L550 and 552_ heights -> altitudes

P23, L550: delete "being driven by"

P23, L553: cloud -> cloud layer

P23, L564: show -> show that

P23, L565: Increasing with what? With time? With space?

P24, L574: heights I -> altitudes from

P24, L575: cloud -> clouds

P24, L576: water trends -> water vapor trends

P24, L580 and 581: add "located" -> are located in the ocean, are located on the land

P24, L580: Delete "In" and move Table 4 in parenthesis at the end of the sentence and start sentence with "Both".

P24, L581-582: What do you mean with substantial water vapor? High concentrations?

P24, L590: Still the connection between water vapor and temperature has not been explained. Why should or does higher temperature cause higher water vapor?

P26, L523: add located (twice) before over the ocean and over land, respectively.

P28, L630: What do you mean with "on the west"? The West Pacific?

P26; L531: from 2007 to 2018 -> for the timer period 2007 to 2018

P26, L632: Change to "From the linear trend study of global……"

P26; L634 to 635: singular or plural? A nearby cloud or nearby clouds? Why nearby? How do you know that a cloud was nearby?

P26, L636: in -> at

P26, L639: delete "area"

P26, L646: paper -> study

P26, L649: move "´better" before "resolve", so that it reads "better resolve……"

P27, L654: The section "Conclusions and Discussions" should be renamed to "Discussion and Conclusion".

P27, L664: I don't agree. Only because the COSMIC data agrees better to ERA5 it does not mean that it is closer to the true state of the atmosphere. The reanalysis is, although data is assimilated, still a

model. I also do not understand why the assimilation impacts are negligible. This discussion should not be squeezed to the major conclusion bulltest, but rather discussed beforehand.

P27; L696: Also mentioning here SPARC feels a bit lost. The mentioning of the efforts of the SPARC community would rather fir into the first paragraph of this section to highlight why such intercomparisons are important.

P27; L675: this paper -> here

P28; L694: estimating from 2007-2018 -> estimates for the time period from 2007 to 2018

P28; L704: have substantial variabilities -> show substantial variability

P28, L707: slopes?

P29, L711: with -> between

P29, L717: move the latitude/longitude coordinates behind the respective areas

P29; L729: What can be better characterized?

P29, L737: Not clear, if you mean here in general or in specific areas. Before good quality of RO data mentioned, here now deficiencies discussed, but is not made clear that this is a correction of the data.

P29, L738: Here it could be stated that ERA5 data has significantly improved compared to ERA-interim.

P29, L739: trending -> trend

P30, L746: times of what. Please be more precise.

P38, L921: trending -> trend

P38, L933, Figure A6 caption: Delete distributions

---

## Author Response (AR2)

**Review of Shao et al.: Characterizing the tropospheric water vapor spatial variation and trend using 2007-2018 COSMIC radio occultation and reanalysis data**

We thank the reviewers for the helpful comments and suggestions. We have revised the manuscript and addressed the reviewer's comments. The manuscript has been largely improved. In the following, we summarize our reply to the reviewer.

**Editor's Comment:**

1. Check the spatial resolution of ERA5. Is it 0.2 or 0.25?
We checked the spatial resolution of the ERA5 reanalysis data we downloaded. It is 0.25 degree. Thank the reviewer for pointing this out. The texts at L131 and L171 have been revised.

2. How the sampling error in COSMIC can affects trend estimation, but not bias?
The relative water vapor biases are estimated from the collocated COSMIC and ERA5 data. The sampling error correction is a correction factor derived from the difference between mean of ERA5 data collocated with COSMIC and the mean of all ERA5 data in the Region of Interest (RoI). The trend estimation needs to keep the time-dependent sampling error correction in the time series of COSMIC and ERA5 data in order to separately estimate the trends.

3. "About the discussion on monsoon climate and water vapour trend"
Precipitation is generally known as a sink rather than a source of water vapour. The monsoon climate influences water vapour variability and trends through the moisture transport. Please modify the discussion to include a link between the water vapour trend and moisture transport, not precipitation.
We have revised the paragraph as "This region is affected by the monsoon climate over the south of the Himalayas. The monsoon climate influences water vapor variability and trends through moisture transport (An et al., 2015; Turner and Annamalai, 2012). The variability in water vapor trends in a region experiencing a monsoon climate is closely tied to the alternating wet and dry phases. Factors such as the strength and duration of the monsoon, the temperature of the ocean waters, and atmospheric circulation patterns all play a role in determining the extent of moisture transport and its impact on water vapor levels. Changes in sea surface temperatures due to global warming can affect the intensity and timing of monsoon patterns, leading to shifts in moisture transport and potentially altering the variability of water vapor content in affected regions. Indian Ocean is an essential part of the coupled Indian monsoon system because it feeds the moist convection over both land and ocean."

4. "Representative months January and July for winter and summer in northern hemisphere, respectively"
I understand that these months can be representative of the seasons, but any significant changes in winter and summer cannot be directly linked to these months; as they could occur in other months of the respective seasons as well. In other words, any changes in these two months, such as trends and bias, may not represent the seasonal change. It should be specified in Methods.

Yes, we agree with the reviewer. Only in Fig. 1, we chose January and July months for the scatter plots of collocated COSMIC water vapor retrieval versus ERA5 and ERA-Interim water vapor data comparison. We also studied the scatter plots for other months (not shown in the

paper), the conclusion that COSMIC water vapor is more consistent with ERA-5 than ERA-Interim holds for these other months as well. We added this note in the paper.

In the later part of paper, the study of the seasonal variability of the of COSMIC and ERA5 water vapor distribution in Section A.1 was performed over 12 months.

5. Reanalysis data can be used as reference for the validation of satellite measurements? To support this argument, cite some references on the validation of ERA5 (to check the consistency of ERA5) and write about the errors, if any.

We added one paragraph in Section 2.1 to address this "Many studies have been conducted to validate the ERA5 atmospheric products using satellite measurements (Chen and Liu, 2016; Lei et al., 2020; Tang et al., 2021; Campos et al., 2022). Overall, the results of these studies show that ERA5 is in good agreement with satellite measurements (or retrieved products). For example, Tang et al. (2021) compared the Atmospheric downward longwave radiation (DLR) from Clouds and Earth's Radiant Energy System (CERES) satellite retrievals and ERA5 data with observations at Baseline Surface Radiation Network (BSRN) stations over land surfaces. The ERA5 atmospheric reanalysis performed better than satellite retrievals in estimating DLR over the land surface. According to Chen and Liu (2016), the global water vapor trend over 1992–2014 from the data of the ECMWF reanalysis model agrees well with the microwave satellite data. These studies provide confidence in the accuracy of the ERA5 products for comparison with COSMIC retrievals."

6. In reply to my previous comment "Why COSMIC water vapour overestimates ERA5 in the upper troposphere?" authors state that there is large uncertainty in retrieving water vapour in the reanalysis model in the upper troposphere. It should be clearly mentioned in the methodology, why there is a bias in ERA5 and how much, because ERA5 is used as the reference data for the validation of COSMIC here. It is better to check the consistency of ERA5 in other pressure levels also.

The main cause that at 300 hPa, water vapor from COSMIC is higher than from ERA5 stems from the distinctive cloud-penetration capability of the RO signal, whereas the water vapor from the reanalysis data is assumed from the cloud-free scenes. COSMIC RO water vapor is retrieved over both cloudy and cloud-free scene, while water vapor from the ERA5 reanalysis model is from cloud-free scene water vapor profile. It is expected that the water vapor concentration derived from COSMIC will be higher than ERA5 at 300 hPa when cloudy scene is accounted in the RO retrieval. Our evaluation of the bias of ERA5 at 300 hPa indicates that the ERA5 may underestimate the global water vapor by about 5.67% when assuming cloud-free in the ERA5. Such assessment is consistent with the water vapor biases between COSMIC-2 and ERA5 presented in Johnston et al., 2021. Figure 1 below shows the height-dependent COSMIC versus ERA5 water vapor biases. The biases between COSMIC and ERA5 grows from nearly 0% at 450 hPa to about 8% at 260 hPa (10 km) in the upper troposphere.

We note that there are large uncertainties in estimating the upper troposphere water vapor by the reanalysis model, which are due to the combined effects of complex atmospheric dynamics (jet streams, convection, and mixing) at high altitudes, sparse observations and difficulties in validation, errors in extrapolating from lower altitude measurements, and accurate accounting of radiative effects at high altitudes. There are ongoing efforts to quantify the ERA5 biases in the upper troposphere through comparison with other measurements such as using multi-campaign data set on research aircraft (Krüger et al., 2022). But, the results are not conclusive due to limited regional, height and temporal coverage of the comparison. In this regard, the comparisons presented in this paper help assess the biases in the reanalysis model. Further

comparisons with collocated radiosonde measurements can also help assess the biases in ERA5 in the upper troposphere.

[Figure]

Figure 1: COSMIC vs. ERA5 water vapor biases (%) as a function of pressure or height.

**Reviewer #1's Comments:**

The manuscript has significantly improved, but still lacks in the quality of the language and a clear writing. There are lot of small errors. It would have been good if the authors would have spent more effort in a careful check of the manuscripts before submission. To list all this small errors is for me as referee really time consuming and distracts me from really focusing on the contents of your study.

We thank the reviewer for the helpful comments and have substantially revised the manuscript.

**General comments:**

The abstract is a bit too long and detailed on the results. I would suggest to skip the explicit numbers of the trend and rather qualitatively state if the trend is positive or negative (and give the order, i.e a few percent). Further, it should be more clearly stated which comparisons have been made instead of writing in detail which differences where derived in which area. I also think mentioning that the El Nino water vapor increases are visible in the data is rather obsolete in the abstract. Write down some key points and make out of these a clear structured abstract.

Following the reviewer's suggestion, we have revised and significantly shortened the abstract. The sentence on the El Nino-related water vapor increases has been removed.

I don't understand the difference between slope and trend. Isn't the slope of the linear fit the

trend?

Yes. the slope is the linear fit of the trend. We went through the paper to limit the use of term "slope" only for fitting.

The result session is still too descriptive and lacks explanations.

We made improvements to the manuscript to address this. Details can also be found in our replies to the specific comments.
1. Added detailed motivations of choosing three pressure layers in this study, their characteristics, and uncertainties, and past studies.
2. Revised and expanded the explanation inter-hemispheric water vapor differences.
3. Added the explanation of the connection between higher temperatures and higher levels of water vapor in the atmosphere
4. Added explanations of Niño3.4 3.4 and Niño4 indices, and the identification of El Niño event.
5. Added the motivations and methods for selecting the regional sites presented in this paper.
6. Expand the discussion on the Monsoon climate effects on regional water vapor trends,

The term trending should be replaced by trend, wetter should be replaced by moister, heights by altitudes.

We went through the manuscript and made corresponding corrections.

There is still too much data analyses shown in the result section. It is not clearly motivated what the intention is of picking all these sites and looking at them.

The representative regional sites selected in Section 5 help quantitatively understand the regional variability, consistency and differences between COSMIC and ERA5 water vapor trends. Section 5.3 in previous draft has been moved to Appendix A.4. Sections 5.2, 5.3 and Section A.4 provide quantitative information of the water vapor trends over the selected sites and complements the color map of Fig. 7 shown in Section 5.1. We added the explanation of the motivations and selection processes for these regional sites.

The first set of sites (Sites #1-3 in Fig. 8) discussed in Section 5.2 is over stratocumulus cloud-rich regions. Section 5.1 helps to answer the question if there are differences between COSMIC and ERA5 water vapor trends over these stratocumulus cloud-rich regions and quantify the difference. The selection of these stratocumulus cloud-rich sites follows the study of Wood 2012 (see the figure below).

[Figure]

Figure 2: Selected stratocumulus cloud-rich regions (Site #1-3 in our paper) overlaid on the annual mean coverage of stratocumulus cloud map from Wood, 2012.

The sites studied in Section 5.3 (Section 5.4 in previous draft) all have substantial water vapor trend differences between COSMIC and ERA5. We added the explanation of the motivation and process for selecting these sites "Comparing regional water vapor trends between COSMIC and ERA5 data and quantifying their differences contribute to validating both datasets. In particular, it can identify regions where the reanalysis model could exhibit constraints. In this section, we select a few sites with a notable trend difference between COSMIC and ERA5 to quantitatively understand the magnitude of the differences and the distribution of these sites. To identify these sites, we first searched the 10º×10º global map of the water vapor trend difference between COSMIC and ERA5 (Fig. 7e and 7f shown in Section 5.1). We identified the regions with the largest positive or negative water vapor trend difference between COSMIC and ERA5. Within these regions, we selected one representative 10º×10º grid in each region as the site of interest. The estimated water vapor trends for COSMIC and ERA5 over these sites, with notable trend differences are listed and compared in Table 4.".

We have moved Section 5.3 in previous draft to Appendix A.4. The motivation and details of the selection process for sites with strong increasing and decreasing water vapor trends are explained in the first paragraph in Section A.4. The explanation can also be found in our response to the question that immediately follows the next one.

since you have trouble expressing yourself it makes following the discussion really difficult. In this case a shorter more concise study would be more beneficial. Section 5 should either significantly improved or omitted. At the moment it makes only the paper longer, but does not provide any knowledge gain.

Section 5 has been improved with our revision. We have shortened Section 5 by moving Section 5.3 to Appendix A.4 "Comparison of COSMIC and ERA5 over sites with notable increasing and decreasing water vapor trends".

We also revised Section 5 following both reviewers' specific comments. For example, we expand the discussion on the Monsoon climate effects on regional water vapor trends, added the explanation of the connection between higher temperatures and higher levels of water vapor in the atmosphere, and added the motivations and methods for selecting the regional sites presented in this paper. Section 5 is a critical part of this paper which complements previous sections on the global and latitudinal water vapor trend comparisons between COSMIC and ERA5. In Section 5, the emphasis on evaluating the consistency and differences of regional water vapor trends between COSMIC and ERA5, and provide quantitative information of the water vapor trends over the selected sites.

For example Section 5.3. Why is this studied? It's just a listing of trend estimates without explaining or discussing the cause or consequence of these increasing trends.
Following the reviewer's suggestion, we have revised and moved Section 5.3 to Appendix A.4 "Comparison of COSMIC and ERA5 over sites with notable increasing and decreasing water vapor trends".

In Section A.4, we added the motivation and the selection process of these regional sites "Although the global and latitudinal water vapor trends presented in Section 4 exhibit an overall upward trend, Figures 7a-d in Section 5.1 highlight that within tropical and subtropical regions, the regional distribution of water vapor trends displays significant local variations with a blend of pronounced increases and decreases in trends. Such variations in regional water vapor trends in general agree with the past studies, e.g., Ross and Elliott, 2001; Dai, 2006; Mieruch et al., 2008, 2014; Zhang et al., 2018. In this section, we quantitatively evaluate the regional variability of water vapor trends by selecting a few sites with notable increasing (Sites #8-#12 in Fig. 8) and decreasing (Sites #13-#17 in Fig. 8) water vapor trends and compare with past studies. To identify these sites, we first searched the $10^\circ \times 10^\circ$ global grids and identified the regions with the largest increasing and decreasing water vapor trends. Within these regions, we selected one representative $10^\circ \times 10^\circ$ grid in each region as the site of interest, and the water vapor trends of these sites estimated from COSMIC and ERA5 data are listed and compared in Table A.1 and A.2. ".

Section 5.4 is quite confusing. Here you refer to Figure 7, but this Figure has already been described in Section 5.1. Are you using this figure for both sections? Is the reference really correct? You should more clearly state here what is shown in this figure and refer to Section 5.1.

We agree with the reviewer and have moved the paragraph on Figure 7e-7f (map of trend difference between COSMIC and ERA5) to Section 5.1 to avoid confusion about the discussion of Fig. 7e and f.

Further, I am not satisfied with your answer why you selected the altitudes 500 and 300 hPa. You should provide a motivation with respect to atmospheric processes or a certain atmospheric region, e.g. stating that you with these levels cover the entire troposphere (lower, middle, higher troposphere) would be fine with me. However, how can you know what differences and uncertainties to expect? Isn't that what you derive from this study? I think you should improve this paragraph (Sect. 2). Further, you should provide an explanation why you expect this differences.

We have revised Section 2 on the motivation of choosing the three pressure levels studied in the paper to address this.

"The pressure level at 850 hPa studied in this paper is close to the surface and within the boundary layer. Its water vapor can vary based on factors such as humidity levels near the surface, regional water vapor sources, and weather patterns. From previous studies (Ho et al., 2009, 2020a; Shao et al., 2021a; Johnston et al., 2021) of comparing RO water vapor data with collocated reanalysis model data or radiosonde measurements, it was found that RO water vapor retrievals have a negative bias in the lower troposphere. The COSMIC water vapor retrieval is strongly affected by super-refraction at this pressure level in the moisture-rich regions (Ho et al., 2010). It is worth evaluating the relative biases and consistency in the trends on various spatial scales between COSMIC and ERA5 water vapor datasets at this 850 hPa pressure level.

The water vapor at 500 hPa can vary widely depending on local weather conditions and atmospheric patterns. Water vapor at 500 hPa is crucial for understanding the development of weather patterns, including mid-latitude cyclones, ridges, and troughs. This pressure level also contributes to the upper-level atmospheric circulation patterns through convection, which carries moist air upward from the lower troposphere and plays a role in redistributing heat and moisture. It was learned from the earlier comparison of RO data with radiosonde measurements that starting from the pressure level at 500 hPa, the RO-water vapor retrieval uncertainty increases as altitude decreases. Therefore, we chose 500 hPa as the representative middle troposphere of interest to study in this paper.

The 300 hPa pressure level represents the water vapor layer with fewer horizontal variations at higher altitudes. Water vapor in the upper troposphere plays a critical role in the Earth's radiative balance and climate system. It affects the absorption and emission of radiation, contributing to warming (absorbing and trapping infrared radiation, i.e., greenhouse effect) and cooling (emitting heat energy) effects. Johnston et al. (2021) showed large discrepancies in the ERA5 and MERRA2 reanalysis model water vapor profiles compared to COSMIC-2 in the upper troposphere. There are large uncertainties for the reanalysis model to estimate the upper troposphere water vapor due to the combined effects of complex atmospheric dynamics (jet streams, convection, and mixing) at high altitudes, sparse observations and difficulties in validation, errors in extrapolating from lower altitude measurements, and accurate accounting of radiative effects at high altitudes. Therefore, we chose 300 hPa as the representative upper troposphere level to compare spatial and temporal variabilities of water vapor between COSMIC and ERA5."

I think also the conclusion and discussion should be improved. You just squeezed in there some answers to my comments, which however feel there rather lost and out of context.
We have revised and improved conclusion and discussion section by incorporating the reviewer's specific comments.

I have listed my specific and technical comments below.
We have addressed the specific and technical comments one by one. See our replies below.

**Specific and technical comments:**

P2, L25: qualities -> quality
Corrected.

P2, L59: skip "and" before microwave
Corrected.

P2, L63: skip "and others". I would say that this is obsolete since you already write in the beginning of the sentence "mainly"
Corrected.

P3, L65: in -> for, so that it reads "for long-term......."
Corrected.

P3, L66: I would not use the term "monitoring" for reanalysis data. This is a term that should be rather used with measurements. Thus, I would suggest to write: These atmospheric reanalysis data have been used for understanding (or investigating) long-term atmospheric water vapor variability and trends (or more general changes).
Rephrased the sentence to "These atmospheric reanalysis data have been used for investigating long-term atmospheric water vapor variability and trends".

P3, L82:the rise -> the increase
Corrected.

P3, L88: data -> datasets
Corrected.

P3, L90: add "ERA" before Interim
Corrected.

P3, L90: better than reanalysis from -> better than the reanalysis data (add "the" and "data")
Corrected.

P3, L92: What exactly do you mean with "other sensor data"? Please clarify and rephrase.
We have rephrased the sentence as "The ERA-interim overestimates the PWV over the ocean for the period before 1992 compared to microwave satellite data."

P3, L94: assure the climate community with -> provide to the climate community (replace assure by provide to and delete with)
Corrected.

P4, L101: microwaves -> microwave
Corrected.

P4, L102: Add "Further," before "RO-derived".
Corrected.

P4, L108: skip comma and add "for" and "and", so that it reads "for climate and meteorological research"
Corrected.

P4, L110: distribution -> distributions
Corrected.

P4, L111: from 2007 to 2018 -> for the time period from 2007 to 2018
Corrected.

P4, L121: replace "As supplementary" by "Additionally"
Corrected.

P4, L122: add "the" -> in the Appendix
Corrected.

P4, L122: ……with introduction to estimating the water vapor trend -> you mean with introducing (or rather describing) the estimation of the water vapor trend? Please check and rephrase (correct English grammar)
Rephrased the sentence to "Appendix A.2 and A.3 describe the estimation of the water vapor trend with sampling error removal and its associated uncertainties for a given region of interest (RoI)."

P5, L138: occulted by the Earth's atmosphere -> not correct, please rephrase.
The sentence has been revised to "bent by atmospheric refraction".

P5, L140: Start the sentence with "From" replace "data" by observations and write "are derived", thus "From the retrievals …… first the bending angle……are derived"
Corrected.

P5, L144: profile -> profiles
Corrected.

P5, L146: 2007 to 1028 _> 2007 to 2018
Corrected.

P6, L166: before the study? Please rephrase? Do you mean before the analyses has been performed?
Corrected.

P6, L169: add "the" -> For the RO
Corrected.

P6, L173: could -> may
Corrected.

P8, L231: troposphere -> tropospheric
Corrected.

P8, L233: small -> low
Corrected.

P8, L233: skip "also" and rephrase the next sentence as follows: "The main cause that at 300 hPa higher water vapor from ERA5 than from COSMIC is derived is due….. estimating water vapor….." You can retrieve data, but from a model the data is simulated or estimated.
Corrected.

P9, L247: delete "model"
Corrected.

P9, L248: in consistently -> is consistently
Corrected.

P9, L249: not the "retrieval", but the "retrieval data", thus change "retrieval" -to "retrieval data" P10, L262: 20-degree-latitude-bin-averaged -> please rephrase and write averaged over 20 degree bins, thus you could write "averaged over 20 degree latitude bins at the three selected pressure levels (300, 500 and 850 hPa).
Corrected.

P10, L266: in Fig. 3 -> shown in Fig. 3 or just write Fig. 3 in parentheses -> (Fig. 3)
Corrected.

P10, L272: a wetter what? A wetter atmosphere? Skip? Rephrase?
We have removed "wetter".

P10, L278: wetter -> moister
Corrected.

P10, L279: Sentence not clear. Please rephrase.
The sentence has been rephrased to "Feulner et al. (2013) examined climatological data, Earth's energy budget, and model simulations for factors that could lead to interhemispheric temperature differences.".

P10, L281: Which factors? Clearly state what you are referring to.
We revised the sentence as "The study of Feulner et al. (2013) compared various factors, including seasonal differences in solar radiation, the tropical land area difference, the difference in albedo and temperature between Antarctic and Arctic polar regions, as well as cross-equatorial ocean heat transport from the southern hemisphere to the northern hemisphere." to make it clear.

P11, L282-283: What do you mean with "warmer NH" and "colder SH"? The winter and summer hemispheres?
We should remove "warmer" and "colder" here. The sentence has been revised to "cross-equatorial ocean heat transport from the southern hemisphere to the northern hemisphere".
The warmer NH and colder SH refer to the air temperature in the northern hemisphere being 1–2°C warmer than in the southern hemisphere according to the annual values for hemispheric average of surface air temperature listed in Table 1 of Feulner et al. (2013).

P11, L286: grow -> increase
Corrected.

P11, L286-287: Sentence is not clear. Something is missing here.

We rephrased the sentence as "As greenhouse gas emissions continued to rise throughout the industrial era, interhemispheric temperature disparities became larger. This is attributed to the intensified warming of land areas compared to oceans and the significant reduction of Arctic sea ice and snow cover in the northern hemisphere.".

P11, L289: These factors? Do you mean these conditions? Not clear what you exactly mean.
We revised the sentence as "These factors, e.g., cross-equatorial ocean heat transport, albedo difference in polar regions, intensified warming of land areas, and reduction of Arctic ice/snow cover, affecting interhemispheric temperature difference, can affect the interhemispheric water vapor difference.".

P11, L289: How or why? How can temperature differences affect water vapor? You just described temperature differences. How is temperature connected to water vapor?
We added explanation "The close relationship between temperature and the capacity of the atmosphere to hold water vapor is governed by the Clausius-Clapeyron equation (Held and Soden, 2006). The equation states that for every 1-degree Celsius increase in temperature, the saturation vapor pressure increases by about 7%. As temperature increases, this will lead to the potential for more water vapor to be held in the air. In other words, warmer air has a higher capacity to hold water vapor. This relationship is crucial for understanding how temperature changes can impact atmospheric humidity. The observed and modeled evidences presented by Wentz and Schabel (2000), Trenberth et al. (2005), Held and Soden (2006), and Allan et al. (2014), supports the notion that higher atmospheric water vapor contents are, in general, associated with higher temperatures."

P11, Figure 3 caption: skip "20-degree-latitudinal......" Just write in the first sentence what is compared and then in the second sentence how the data has been treated.
Corrected.

P11, L291: retrieval -> observation or retrieved data
Corrected.

P11, L295: over all months in 12 years -> for all months of the considered 12 year period
Corrected.

P12, L297: add "shown" before "in the middle......."
Corrected.

P12, L310: put Fig 3h and 3f in parenthesis
Corrected.

P12, L311: Fig 3h shows -> From Fig 3h it can be seen.....
Corrected.

P12, L314: Same here: From Fig 3j it can be seen..... P12, L316: add "differences" before being
Corrected.

P12, L3198-319: Skip "after sampling......". You don't need this in the section title. It is enough to mention this in the text part.
Corrected.

P12, L323: Sampling removed water vapor -> rephrase. Rather write COSMIC water vapor data where sampling errors have been removed.
Corrected.

P12, L323: in the rest of this paper -> in the remainder of this paper, but better would be to write in the following.
Corrected.

P12, L325: This section compares. Not the section is doing the comparison, but you. Correctly it should read "In this section………are compared".
Corrected.

P13, L335: Change "It is noted" to "It becomes visible" or "It can be seen"
Corrected.

P13, L341. What is NINO3.4 and NIN4? An explanation should be given in the text.
The sentence has been revised as "The recent 2015-2016 El Niño event broke warming records in the central Pacific according to Niño3.4 (sea surface temperature (SST) anomalies averaged over the equatorial region (Latitude: -5º to 5º; Longitude: -150º to 160º) of the Pacific Ocean) and Niño4 indices (SST anomalies over the region (Latitude: -5º to 5º; Longitude: -150º to 160º))." to explain the two indices.

P13, L342: add "the" before April
Corrected.

P13, L346: Add "the" before
seasonal
Corrected.

P13, L346: change "as seen" to "as visible" or even better skip this and put Fig 4a in parenthesis at the end of the sentence.
Corrected.

P14, Figure 4 caption: Needs to be improved. Too much repetition. First sentence obsolete? Trending should be replaced by trend (throughout the manuscript).
Corrected.

P14, L366: trending -> trend and put Fig 4a in
parenthesis P15, L366: Add time period after water
vapor concentration.
Corrected.

P15, L369: with -> using, delete of ECMWF data or write ECMWF ERA-40 data
Corrected.

P15, L369, 370 and 371: delete "in" and put references in parenthesis.
Corrected.

P15, L373: were trended -> were used to derive the
trend P15,
Corrected.

L378: move the time period behind paper
Corrected.

P15, L379: at three -> at the three and replace from by considered in
Corrected.

P15, L384: add "the" -> the three
Corrected.

P15, L386: trends -> trend, with -> for the and move the time period data
Corrected.

P15, L387: is -> are
Corrected

P15, L388: overlapped -> overlapping
Corrected.

P15, L390: Just write "a positive water vapor trend of " or " a positive water vapor trend of 1.44%". If you write positive or negative you do not need to give a number. Vice versa, If you give the number then you do not need to write positive or negative.
Corrected.

P16, L404: trending -> trend
Corrected.

P16, L408 and 411: What do you mean with trending slope? Is not the trend the slope of the linear fit?
Corrected.

P16, L409: in -> at and delete "latitude range" at the end of the sentence.
Corrected.

P16, L416: in should be replaced by at or for
Corrected.

P17, L426: "mixed with" should rather read "composed of" or "consisting of".
Corrected.

P17, Figure 5 caption: Comparing -> Comparison of (two occasions)
Corrected.

P17, Figure 5 caption L437: skip retrieval.
Corrected.

P17-18: Figure 5: The sentence starting with "The bar ….." is not clear and needs to be rephrased.
Corrected.

P18, L445ff: In most occasions you can skip writing northern and southern. IF you provide the coordinates with plus and minus signs this is enough.
Corrected.

P18, $49: within -> of
Corrected.

P18, L453: Put Fig 5f in parenthesis and delete "in"
Corrected.

P18, L454: Why -60 to 80? Is that correct? Or should it be -60 to -80?
Corrected.

P19, L457: Put "Figure 5g and Table 2" in parenthesis and delete "show that"
Corrected.

P19, L460: high -> higher
Corrected.

P19, L461: estimations -> estimated
Corrected.

P19, 463: with latitude bins needs to be rephrased.
Corrected.

P19, L460ff: I thought you made a separation into 20 degree bins, Why are in this section larger bins discussed?
Certain latitude bins exhibit similar or closely aligned trends. As a result, the characteristics of these latitude bins are collectively discussed within combined latitude ranges.

P19, L466-467: Sentence not clear. Needs to be rephrased.
We have rephrased the sentence as "This indicates that the main reason for the relatively lower global water vapor trends estimated from COSMIC data compared to ERA5 data at the 850 hPa level (as presented in Table 1) is the lower values of COSMIC trends within the middle and low latitude bins."

P19, L475: interval -> period
Corrected.

P19, L476: the period -> this time period
Corrected.

P19, L476: delete distributed and write "mostly fond over the ……"
Corrected.

P19, L477: above -> greater than
Corrected.

P19, L478: Sentence not clear. Please correct.

We have rephrased the sentence to "Missing COSMIC RO data is prominent over the regions covering the Tibetan Plateau, specifically at pressure levels of 500 and 850 hPa. The absence of RO data in these regions can be attributed to the lower atmospheric pressure prevailing over areas at an average altitude of around 4 km."

P19, L478:trending -> trend
Corrected.

P20, Figure 6 caption: no monthly data missing? What do you mean here? With no data or with missing data? I guess the grids with missing data are shown as white blanks.
Corrected. Figure 6 caption has been revised as "The percentage of missing monthly data over the 2007 to 2018 interval on the global 10°×10° grids. The percentage of missing data is shown as color-coded. Grids with complete monthly data and without gaps, i.e., covering all months, are represented as white blank spaces."

P20: I do not understand what you mean here with interface.
Corrected.

P21, Figure 7 caption: In two occasions trending should be replaced by trend.
Corrected.

P22, General: Move the latitudes und longitudes behind the respective region, e.g Laccadive Sea
Corrected.

P22, L533: Here you write this region, but two different latitude and longitudes are given. Do you mean "these regions"? And which regions are you talking here about? The text would be much easier to read if you would put the coordinates at the end and not always after "region".
Corrected.

P22, L544: "These established sites are in 10 by 10 longitude/latitude grids". Not clear what you mean since sentence not grammatically correct. What do you mean with established?
We have merged this sentence and the previous sentence to make it clearer. It is revised as "The center locations of these selected 10°×10° grids are shown in Fig. 8.".

P23, L550 and 552_ heights -> altitudes
Corrected.

P23, L550: delete "being driven by"
Corrected.

P23, L553: cloud -> cloud layer
Corrected.

P23, L564: show -> show that
Corrected.

P23, L565: Increasing with what? With time? With space?
Corrected.

P24, L574: heights I -> altitudes from
Corrected.

P24, L575: cloud -> clouds
Corrected.

P24, L576: water trends -> water vapor trends
Corrected.

P24, L580 and 581: add "located" -> are located in the ocean, are located on the land
Corrected.

P24, L580: Delete "In" and move Table 4 in parenthesis at the end of the sentence and start sentence with "Both".
Corrected.

P24, L581-582: What do you mean with substantial water vapor? High concentrations?
Corrected.

P24, L590: Still the connection between water vapor and temperature has not been explained. Why should or does higher temperature cause higher water vapor?
We added the explanation in Appendix A.4 "Many previous studies have explored the trends in surface temperature (e.g., Gu and Adler, 2022 and references therein). Global surface keeps warming up, though with rich spatial structures of temperature change. Higher surface temperatures are closely linked to higher levels of water vapor in the atmosphere through the relationship governed by the Clausius-Clapeyron equation. The saturation vapor pressure of water vapor increases with temperature. The close relations between higher temperature and higher water vapor have been shown in observations and models (Wentz and Schabel, 2000; Trenberth et al., 2005; Held and Soden, 2006; Allan et al., 2014). From the study by Gu and Adler, 2022, ocean surface warming can readily be seen in the Indian and tropical Pacific oceans, roughly corresponding to the strong increasing tropospheric water vapor trends for Sites #8, #9, and #12 we observed."

P26, L623: add located (twice) before over the ocean and over land, respectively.
Corrected.

P28, L630: What do you mean with "on the west"? The West Pacific?
We revised the sentence to "There are no 850 hPa RO data over the Andes Mountains (over 6 km in altitude) area. The RO water vapor trend data mainly come from the Pacific Ocean in the 10°×10° grid of Site #7."

P26; L631: from 2007 to 2018 -> for the timer period 2007 to 2018
Corrected.

P26, L632: Change to "From the linear trend study of global……"
Corrected.

P26; L634 to 635: singular or plural? A nearby cloud or nearby clouds? Why nearby? How do you know that a cloud was nearby?

We have revised the sentence as "Site #7 is situated in close proximity to Site #3 and falls within an area where there is a frequent presence of low-height stratocumulus clouds (Wood, 2012)"

P26, L636: in -> at
Corrected.

P26, L639: delete "area"
Corrected.

P26, L646: paper -> study
Corrected.

P26, L649: move "´better" before "resolve", so that it reads "better resolve……"
Corrected.

P27, L654: The section "Conclusions and Discussions" should be renamed to "Discussion and Conclusion".
Corrected.

P27, L664: I don't agree. Only because the COSMIC data agrees better to ERA5 it does not mean that it is closer to the true state of the atmosphere. The reanalysis is, although data is assimilated, still a model. I also do not understand why the assimilation impacts are negligible. This discussion should not be squeezed to the major conclusion bulltest, but rather discussed beforehand.
We agree with the reviewer and removed the sentence "COSMIC water vapor is closer to the true state of the atmosphere, i.e.," to avoid calling ERA5 as the true state of the atmosphere. We also rephrase the sentence as "). Our study shows that COSMIC water vapor retrievals are more consistent with ERA5 reanalysis data than ERA-Interim. From the data assimilation point of view, this suggests that although UCAR COSMIC 1DVAR retrieval used ERA-Interim as the background model (see Section 2.2), but the impacts from ERA-Interim in the UCAR 1DVAR retrieval processing is minimum.". The impacts of the ERA-Interim on UCAR COSMIC 1DVAR retrieval was discussed in Section 2.2 and referenced here.

P27; L666: Also mentioning here SPARC feels a bit lost. The mentioning of the efforts of the SPARC community would rather fir into the first paragraph of this section to highlight why such intercomparisons are important.
This is a good suggestion. We moved the mentioning of "SPARC" to the beginning of this paragraph. The revised sentence reads "There have been coordinated efforts from Stratosphere–troposphere Processes And their Role in Climate (SPARC) Reanalysis Inter-comparison Project (S-RIP) to compare reanalysis datasets such as ERA5 and ERA-Interim using a variety of key diagnostics. The SPARC S-RIP confirmed the significant improvements of the latest version of reanalyses in ERA5 compared to ERA-interim (Fujiwara et al., 2017)."

P27; L675: this paper -> here
Corrected.

P28; L694: estimating from 2007-2018 -> estimates for the time period from 2007 to 2018
Corrected.

P28; L704: have substantial variabilities -> show substantial variability
Corrected.

P28, L707: slopes?
We have corrected it to "trends".

P29, L711: with -> between
Corrected.

P29, L717: move the latitude/longitude coordinates behind the respective
areas
Corrected.

P29; L729: What can be better characterized?
We revised the sentence to "the height and temporal distribution of water vapor can be better characterized in RO retrievals than ERA5 in the presence of convection, such as deep clouds.".

P29, L737: Not clear, if you mean here in general or in specific areas. Before good quality of RO data mentioned, here now deficiencies discussed, but is not made clear that this is a correction of the data.
Our intention was to discuss the sampling error removal and that the comparison of water vapor trend of COSMIC with ERA5 can be extended to compare with other reanalysis models such as MERRA and NCEP. We have combined the two paragraphs to "In analyzing long-term water vapor trends from RO data, it is important to remove sampling errors to correct the biases due to RO data's limited time and location coverage. ……This paper's overall global water vapor trends are close to the trend results from Allan et al. (2022). We postulate that using other global reanalysis models, such as NCEP and MERRA-2, may have compatible global trends but differ in regional trends from our results, which will need further evaluation.".

P29, L738: Here it could be stated that ERA5 data has significantly improved compared to ERA- interim.
We revised the sentence to "confirms that ERA5 has significantly improved quality than ERA-Interim."

P29, L739: trending -> trend
Corrected.

P30, L746: times of what. Please be more precise.
We revised the sentence as "After applying sampling error removal, our estimations indicate a reduction in uncertainty by approximately 4.8 times at 500 hPa and 3.1 times at 850 hPa." to make it clear.

P38, L921: trending -> trend
Corrected.

P38, L933, Figure A6 caption: Delete distributions
Corrected.

---

## Author Response (AR3)

**Review of Shao et al.: Characterizing the tropospheric water vapor spatial variation and trend using 2007-2018 COSMIC radio occultation and reanalysis data**

We thank the reviewers for the helpful comments and suggestions on the minor revision. We have revised the manuscript and addressed the reviewer's comments. The manuscript has been largely improved. In the following, we summarize our reply to the reviewer.

**Reviewer #1's Comments:**
I really appreciate the effort the authors have put in the revision of their manuscript. However, there are still some technical/minor issues that should be considered before publication.

**General comments:**
P5, Section 2.2: This section is too long is not entirely describing the data. Here you have a mixture between data set description (L141-L167), method description (L169 to L202) and results (L204-227). The method part should be put into an extra subsection and the results part either there in a extra subsection or moved in an extra subsection to the results section.

Following the reviewer's suggestion, we have inserted two section titles "2.3 Method of comparing COSMIC and ERA5 water vapor data" (L168) and "2.4 Impact of ERA-Interim as *a priori* on COSMIC water vapor retrieval" (L203) to separate Section 2.2 into three subsections.

P9, L264: Not clear why you consider reanalysis data for cloud-free scenes if there should be data available for cloud and cloud-free scenes.
ERA5 provides atmospheric water vapor data for both cloudy and cloud-free scenes. However, there are two aspects to consider in this context. On one hand, it poses a challenge for reanalysis models to accurately represent atmospheric conditions over cirrus or thin clouds, often leading to misclassification as cloud-free scenes. This misclassification introduces uncertainties in the water vapor data obtained from ERA5 for the cloud-free scenes. On the other hand, the water vapor concentration derived from COSMIC in the RO retrieval system may include some effects from the thin or cirrus clouds, resulting in a slightly higher reported value. We have revised the sentence as "In contrast, there are uncertainties in the water vapor from the reanalysis data over the cloud-free scenes since these scenes can be over thin or cirrus cloud due to the difficulty in the data assimilation system over these types of clouds. The water vapor concentration derived from COSMIC is expected to be higher than ERA5 at 300 hPa when the thin or cirrus cloud are present. Our evaluation of water vapor at 300 hPa indicates that the difference between RO and ERA5 about 5.7% is likely due to the uncertainty in classifying cloud-free scenes in the data assimilation and in the RO retrieval system. Such assessment is consistent with the water vapor biases between COSMIC-2 and ERA5 presented in Johnston et al., 2021.".

P11, L292-293: "…….large uncertainties in retrieving water vapor in the reanalyses model……… " Please rephrase. This is simply not correct. A model does not retrieve any data. In a model data is calculated.
We agree with the reviewer and have changed "retrieving" to "calculating".

P11, L295: You should formulate this more carefully. How can you be sure that if there is a bias this is a bias of the ERA5 data? What about the COSMIC RO data? Nowhere in the paper the quality of the RO data is discussed.
We agree with the reviewer. This is still an open question and will need further study. We revised the sentence as "Further comparisons of reanalysis model data with collocated

radiosonde measurements and RO retrievals can help assess and understand the uncertainties in estimating the upper troposphere water vapor.".

P12, L340-341:"……..affecting interhemispheric temperature difference, can affect the interhemispheric water vapor difference". Sentence not clear. It seems in the latter part of this sentence something is missing. Please correct/rephrase the sentence.
We revised the sentence as "These factors, including cross-equatorial ocean heat transport, albedo difference in polar regions, intensified warming of land areas, and reduction of Arctic ice/snow cover, which affect interhemispheric temperature difference, can also be the primary driving factors of the interhemispheric water vapor difference.".

**Technical corrections:**
P1, L23: upward -> increasing
Corrected.

P1, L24: downward -> decreasing
Corrected.

P1, L26: higher than ERA5 data -> higher than the ones derived from ERA5 data
Corrected.

P17, L453: trending -> trend
Corrected.

P17, L454: delete "than" and move "300 and 500 hPa at the end of the sentence and add "at" before so that the sentence reads: …….is lower by 1.44 and 1.22%/Decade at 300 and 500 hPa, respectively.
Corrected.

P20, L514: is -> are? Consider rephrasing sentence. It is difficult to understand what you want to say.
We have rephrased the sentence to "This indicates that the relatively lower global water vapor trends estimated from COSMIC data compared to ERA5 data at the 850 hPa level (as presented in Table 1) are mainly due to the lower values of COSMIC trends within the middle and low latitude bins.".

P21, L546: slope -> trend
Corrected.

P22, Figure 7 caption: replace 2 times "slope" with "trend"
Corrected.

P23, L584: add "the" -> in the affected regions
Corrected.

P23, L599: replace "degree" by the degree sign "°"
Corrected.